# ADAFISHER: ADAPTIVE SECOND ORDER OPTIMIZATION VIA FISHER INFORMATION

**Damien Martins Gomes**[*]
Concordia University and IPSA Toulouse
`damien.martinsgomes@mail.concordia.ca`

**Yanlei Zhang**
Université de Montréal and Mila
`yanlei.zhang@mila.quebec`

**Eugene Belilovsky**
Concordia University and Mila
`eugene.belilovsky@concordia.ca`

**Guy Wolf**
Université de Montréal and Mila
`wolfguy@mila.quebec`

**Mahdi S. Hosseini**[†]
Concordia University and Mila
`mahdi.hosseini@concordia.ca`

## ABSTRACT

First-order optimization methods are currently the mainstream in training deep neural networks (DNNs). Optimizers like Adam incorporate limited curvature information by employing the diagonal matrix preconditioning of the stochastic gradient during the training. Despite their widespread, second-order optimization algorithms exhibit superior convergence properties compared to their first-order counterparts e.g. Adam and SGD. However, their practicality in training DNNs is still limited due to increased per-iteration computations compared to the first-order methods. We present *AdaFisher*–an adaptive second-order optimizer that leverages a *diagonal block-Kronecker* approximation of the Fisher information matrix for adaptive gradient preconditioning. AdaFisher aims to bridge the gap between enhanced *convergence/generalization* capabilities and computational efficiency in second-order optimization framework for training DNNs. Despite the slow pace of second-order optimizers, we showcase that AdaFisher can be reliably adopted for image classification, language modeling and stands out for its stability and robustness in hyper-parameter tuning. We demonstrate that AdaFisher **outperforms the SOTA optimizers** in terms of both accuracy and convergence speed. Code is available from https://github.com/AtlasAnalyticsLab/AdaFisher.

## 1 INTRODUCTION

Deep Neural Network (DNN) optimization often struggles with the challenge of generalizing across varied architectures and complex data distributions. Current methods such as Adam optimizer (Kingma & Ba, 2015) and its variants (AdamP (Heo et al., 2021), AdaInject (Dubey et al., 2022), AdaBelief (Zhuang et al., 2020) and YOGI Zaheer et al. (2018)) require extensive Hyper-Parameter (HP) tuning and often fail to generalize efficiently. DNN training typically minimizes a highly non-convex loss function $\mathcal{L}(\theta)$, updating parameters $\theta$ using the expression $\theta^{(t+1)} = \theta^{(t)} - \alpha(\mathcal{G}^{(t)})^{-1}\nabla\mathcal{L}(\theta^{(t)})$ at time step $t$, where $\mathcal{G}^{(t)}$ represents the curvature information. Here, $\mathcal{G}$ is the identity matrix for first-order optimizations such as SGD (Kiefer & Wolfowitz, 1952), and Hessian or Fisher Information Matrix (FIM) for the second-order case (Amari & Nagaoka, 2000). The Hessian matrix interfaces with the deterministic Newton-Raphson method (Holmgren, 1996), whereas the FIM harmonizes with the statistical measure of the Natural Gradient Descent (NGD) approach (Amari & Nagaoka, 2000). This curvature information crucially optimizes the gradient's preconditioning by accurately rescaling and orienting it. This adjustment significantly

---

[*]To my father and grandmother, whose strength and love continue to inspire me, this work is dedicated.
[†]Corresponding Author

accelerates convergence by ensuring more direct progress towards minima, thus enhancing training efficiency and reducing the number of required iterations (Kashyap, 2022).

As mentioned, the second-order methods employ curvature matrices for $\mathcal{G}$ to enhance the optimization process. Although these matrices accelerate convergence by effectively navigating through saddle points and swiftly moving towards minima (Foret et al., 2021), they require higher computational resources for the inverse computation. In fact, when the number of learnable parameters increases, the curse of dimensionality associated with curvature matrix $\mathcal{G}$ makes the entire training process completely intractable by a commodity hardware platform.

Noteworthy approaches, such as Adagrad (Duchi et al., 2011), Adadelta (Zeiler, 2012), RMSProp (Hinton et al., 2012), and Adam family utilize a simple diagonal approximation of the empirical FIM, which often results in convergence to suboptimal local minima and poor generalization (Wilson et al., 2017; Luo et al., 2019). Advanced methods like AdaHessian (Yao et al., 2021) and Shampoo (Gupta et al., 2018) improve this by integrating structured matrices such as the diagonal Hessian or tensor-based preconditioners to enhance optimization. However, these second-order approaches, including K-FAC (Martens & Grosse, 2020; Eschenhagen et al., 2024), still face challenges of high computational demands, lacking generalization and needing extensive HP tuning when applied to large-scale models (Ma et al., 2020).

To address these challenges, we present AdaFisher, an adaptive second-order optimizer, as an innovative solution to address the generalization challenges raised in training DNNs. Substituting the second moment of Adam by a novel *diagonal block-Kronecker* approximation of the FIM, AdaFisher strikes a balance between simplicity and generalization by introducing an extra HP compared to Adam but fewer than K-FAC, AdaHessian or Shampoo. With memory and time requirements on par with first-order methods, AdaFisher remains a practical choice for achieving effective generalization in DNN optimization. As illustrated in Figure 1, AdaFisher not only converges more rapidly but also reaches a superior local minimum by effectively navigating through saddle points compared to its counterparts. Further details regarding the visualization can be found in Appendix C.

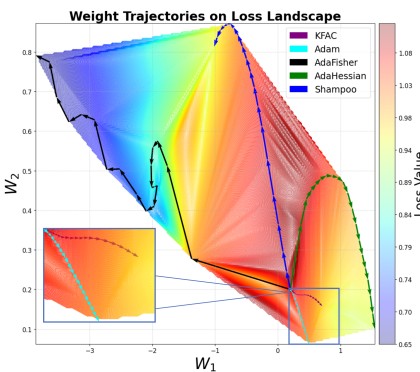

Figure 1: Visualizing optimization trajectories for various optimizers overlaid a loss landscape.

In summary, our contributions build upon these findings as follows: **[C1]** We empirically showcase the energy of the Kronecker Factors (KF) is mainly concentrated along the diagonal and provide fresh insights of FIM in optimization; **[C2]** We introduce a diagonal block-Kronecker approximation of the FIM applicable to various layers, including normalization layers, enhancing model adaptability; **[C3]** We demonstrate AdaFisher's robustness and stability across diverse settings, proving its effectiveness; **[C4]** We showcase AdaFisher's empirical performance against SOTA optimizers in image classification and language modeling, highlighting its superior performance; **[C5]** We develop a new technique that visualizes trajectories across different optimizers for better understanding of model behavior in the loss landscape. Additionally, we introduce an explainable FIM measure from AdaFisher, enabling comparative analysis of optimizer behavior.

## 2 BACKGROUND

We consider a supervised learning framework with a dataset $\mathbf{D}$ containing $N$ i.i.d samples, $\mathbf{D} \coloneqq \{x_n, y_n\}_{n=1}^N$ where $x_n \in \mathbb{R}^d$ and $y_n \in \mathbb{R}^C$. Let $f_\theta : \mathbb{R}^d \to \mathbb{R}^C$ be a L-layer neural network parametrized by $\theta$ where $\theta_i = \text{concat}(W_i, b_i) \in \mathbb{R}^{P_i}$, and $P_i = P_i^{out} \times (P_i^{in} + 1)$. Let $\mathcal{L} : \mathbb{R}^C \times \mathbb{R}^C \to \mathbb{R}$ be the loss function defined by the negative log-likelihood, i.e. $\mathcal{L}(y, f_\theta(x)) \coloneqq -\log p_\theta(y|x)$ where $p_\theta(y|x)$ is the likelihood of the neural network $f_\theta$. The network computes its output $h_L = f_\theta(x)$ according to

$$a_i = \theta_i \bar{h}_{i-1}, \; h_i = \phi_i(a_i), \; \forall i \in \{1, \dots, L\} \mid h_0 = x_n,$$

where $\bar{h}_{i-1} = [h_{i-1}^\top, 1]^\top \in \mathbb{R}^{P_{i-1}^{in}+1}$, and $\phi_i$ is an element-wise nonlinearity applied at layer $i$. For a given input target pair $(x, y)$, the gradient of the loss $\mathcal{L}(y, f_\theta(x))$ concerning the weights are

computed by the backpropagation algorithm (Lecun, 2001). For convenience, we adopt the special symbol $s_i = \nabla_{a_i}\mathcal{L}$ for the pre-activation derivative. Starting from $\nabla_{h_L}\mathcal{L} = \partial_{h_L}\mathcal{L}(y, h_L)$, we perform

$$s_i := \nabla_{a_i}\mathcal{L} = \nabla_{h_i}\mathcal{L} \odot \phi_i'(a_i), \ \nabla_{\theta_i}\mathcal{L} = s_i \bar{h}_{i-1}^\top, \ \nabla_{\bar{h}_{i-1}}\mathcal{L} = \theta_i^\top s_i \quad | \ \forall i \in \{L, \ldots, 1\},$$

where $\odot$ denotes the element-wise product. Finally, the gradient $\nabla_\theta\mathcal{L}$ is retrieved by: $\nabla_\theta\mathcal{L} = [\text{vec}(\nabla_{\theta_1}\mathcal{L})^\top, \text{vec}(\nabla_{\theta_2}\mathcal{L})^\top, \ldots, \text{vec}(\nabla_{\theta_L}\mathcal{L})^\top]^\top$; $\text{vec}(\cdot)$ denotes the Kronecker vectorization operator which stacks the columns of a matrix into a vector. Optimization of a DNN can be recast as a problem of finding the parameter set $\theta$ that maximizes the likelihood, or equivalently, minimizes the negative log-likelihood of the observed data. This Maximum Likelihood Estimation can be expressed as an unconstrained composite optimization problem: $\min_\theta J(\theta) = \sum_{n=1}^N \mathcal{L}(y_n, f_\theta(x_n))$, where $J(\theta)$ denotes the objective function, corresponding to the negative log-likelihood of the data. For notation convenience, we define $g_i = \nabla_{\theta_i} J(\theta)$. The FIM, utilized in lieu of the Hessian for Newton-Raphson's method, approximates the curvature of the log-likelihood function (Amari, 1998),

$$F = \sum_{n=1}^N \mathbb{E}_{y \sim p(y|f_\theta(x_n))}\left[\nabla_\theta \log p_\theta(y|x_n)\nabla_\theta \log p_\theta(y|x_n)^\top\right] = \mathbb{E}\left[\nabla_\theta J(\nabla_\theta J)^\top\right] = \mathbb{E}[gg^\top], \quad (1)$$

where $F$ measures the expected information that an observable $y$ conveys about the parameter $\theta$. For brevity, we write $\mathbb{E}$ instead of $\mathbb{E}_{y \sim p(y|f_\theta(x_n))}$. The K-FAC approach further simplifies FIM calculation using a block-diagonal approximation in DNNs, known as Empirical FIM (EFIM), denoted by $\hat{F}$. In Eq. (1), $F$ is construed as a block matrix with dimensions $L \times L$, where each $(i, j)$th block $F_{i,j}$ is articulated by $F_{i,j} = \mathbb{E}[\text{vec}(g_i)\text{vec}(g_j)^\top]$. From the Kronecker-vectorization equality $\text{vec}(uv^\top) = v \otimes u$, we express $\text{vec}(g_i)$ as $\bar{h}_{i-1} \otimes s_i$ (Petersen & Pedersen, 2008), where $g_i$ is defined as $s_i \bar{h}_{i-1}^\top$. By segmenting the FIM into discrete layer-specific blocks, a systematic factorization of each block yields as

$$\hat{F}_{i,j} = \mathbb{E}[\text{vec}(g_i)\text{vec}(g_j)^\top] = \mathbb{E}[\bar{h}_{i-1}\bar{h}_{j-1}^\top \otimes s_i s_j^\top] \approx \mathbb{E}[\bar{h}_{i-1}\bar{h}_{j-1}^\top] \otimes \mathbb{E}[s_i s_j^\top],$$

where $i, j$ span the layer indices from 1 to $L$. Given the dimensionality of $\bar{h}_{i-1} \in \mathbb{R}^{P_i^{in}+1}$ and $s_i \in \mathbb{R}^{P_i^{out}}$, the Fisher matrix dimension will be $\hat{F}_{i,j} \in \mathbb{R}^{P_i \times P_i}$. Initially, K-FAC estimates the expectation of the Kronecker product under the presumption that activations and pre-activation derivatives are mutually independent. This can be represented as the Kronecker product of the individual expectations $\hat{F}_{i,j} = \mathcal{H}_{i-1,j-1} \otimes \mathcal{S}_{i,j}$, where $\mathcal{H}_{i-1,j-1} = \mathbb{E}[\bar{h}_{i-1}\bar{h}_{j-1}^\top]$ and $\mathcal{S}_{i,j} = \mathbb{E}[s_i s_j^\top]$, denoting the KFs. The assumption for the block-diagonal structure posits that weight derivatives across distinct layers are uncorrelated, expressed by $\hat{F} = \text{diag}(\hat{F}_{1,1}, \ldots, \hat{F}_{L,L}) = \text{diag}(\hat{F}_1, \ldots, \hat{F}_L)$.

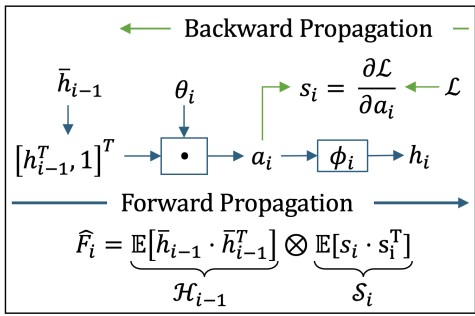

Figure 2: Illustration of EFIM computation using K-FAC for a given layer $i$.

Figure 2 illustrates the EFIM computation via K-FAC for a given layer $i$.

## 3 METHODOLOGY

Our methodology consists of four primary components: **(i)** Analyzing the KFs' structure in Section 3.1 and showcase their diagonal dominance; **(ii)** Introducing a novel approximation of the FIM that retains only the diagonals of the KFs, detailed in Section 3.2; **(iii)** Incorporating the diagonal FIM approximation within the adaptive optimization framework as an alternative to the conventional second moment used in Adam, described in Section 3.3; **(iv)** Providing a theoretical proof of AdaFisher's convergence under both convex and non-convex conditions in Section 3.4.

### 3.1 DIAGONAL CONCENTRATION OF KFS

Inspired by Gershgorin circle theorem (Horn & Johnson, 2012), we empirically conclude the KFs are diagonally concentrated by studying their eigenvalue distribution and perturbation under Gaussian noise. For demonstration, we focus on the eigenvalue spectrum of weight matrices from the

37th layer of ResNet-18 (He et al., 2016) after training for 50 epochs on CIFAR-10 (Krizhevsky et al., 2009). As illustrated in Figure 3, the eigenvalues (denoted as red crosses) predominantly cluster within the Gershgorin discs, which are centered along the matrix's diagonal elements (denoted as black circles), signifying substantial diagonal dominance. This phenomenon is quantitatively supported by the Gershgorin circle theorem, which posits that every eigenvalue $\lambda$ of a complex square matrix $\mathcal{A}$ lies within at least one of the Gershgorin discs $D(a_{ii}, R_i)$, where $R_i = \sum_{j \neq i} |a_{ij}|$ represents the radius computed as the sum of the absolute values of the off-diagonal entries of the $i$th row. Next, we add Gaussian noise $\mathcal{N}(0, \sigma^2)$, $\sigma = 10^{-3}$ on off-diagonal elements. The perturbed matrix $\hat{\mathcal{M}}$ is then expressed as $\hat{\mathcal{M}} = \mathcal{A} + \mathcal{E}$, where $\mathcal{E} = [e_{ij}]$ and $e_{ij} \sim \mathcal{N}(0, \sigma^2)$ for $i \neq j$.

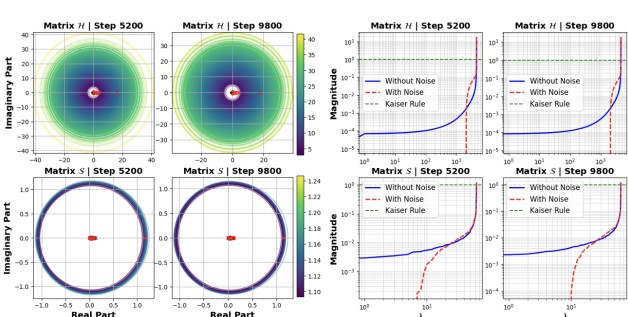

The noise perturbation on KF eigenvalues is critical to comprehend the dynamics and stability of the matrix. This demonstrates the introduction of noise to the off-diagonal elements yields minimal eigenvalue perturbations, particularly those surpassing the Kaiser criterion (i.e. most dominant eigenvalues), which remain virtually unchanged (Braeken & Van Assen, 2017). Both the above understandings from Gershgorin disc analysis and eigenvalue perturbation corroborate the robustness of the matrix's **diagonal dominance**. Extensive discussions of these analyses, including the KFs Fourier analysis, are available in Appendix A.1.

Figure 3: Gershgorin disks and eigenvalue perturbations from the 37th Convolutional Layer of ResNet-18 at steps 5200 (middle of training) and 9800 (end of training). Left: Gershgorin circles; Right: Eigenvalue spectrum w/w-o noise.

## 3.2 EFFICIENT COMPUTATION OF THE FIM

In the realm of optimization, NGD offers a geometrically nuanced adaptation of the classical steepest descent approach (in Euclidean space), transitioning the focus from parameter space to the model's distribution space underpinned by the adoption of a *Riemannian metric*, Amari & Nagaoka (2000). The formulation of the *preconditioned* gradient $\bar{g}^{(t)}$ given the NGD method is articulated as

$$\bar{g}^{(t)} = (F^{(t)})^{-1} g^{(t)}, \tag{2}$$

where $F^{(t)}$ denotes the FIM at time step $t$ distinguished from $F_i$, the FIM at layer $i$. One of the distinguishing features of NGD within this framework is its re-parametrization invariance, a direct consequence of leveraging the model's distribution properties rather than its parameters. Nevertheless, the direct FIM computation is highly demanding, and we solve this by adopting the diagonal approximation of the KFs as supported by our analyses from Section 3.1. In addition, a critical component of modern DNNs is known to be the normalization of the layers by introducing scale and shift parameters (e.g. batch-normalization (Ioffe, 2015), layer-normalization (Lei Ba et al., 2016)). This is to adjust the network's dynamics (e.g., reducing covariance shift) in a non-trivial way Huang et al. (2023) where the lack of FIM approximation on such normalization layers can lead to suboptimal preconditioning. Therefore, we introduce a method for calculating the KFs for normalization layers detailed in Proposition 3.1.

**Proposition 3.1** (EFIM for normalization layer). *Let $(\nu_i, \beta_i) \in \mathbb{R}^{C_i}$ be the scale and shift parameters of a normalization layer $i$. The empirical KFs for the FIM approximation are*

$$\mathcal{H}_{i-1}\Big|_{\nu_i} = \frac{1}{|\mathcal{T}_i|} \sum_{x \in \mathcal{T}_i} h_{i-1,x} h_{i-1,x}^\top, \quad \mathcal{H}_{i-1}\Big|_{\beta_i} = \mathbf{1}\mathbf{1}^\top, \quad \mathcal{S}_i = \frac{1}{|\mathcal{T}_i|} \sum_{x \in \mathcal{T}_i} s_{i,x} s_{i,x}^\top,$$

*where $h_{i-1}, s_i \in \mathbb{R}^{C_i \times |\mathcal{T}_i|}$ represent the pre-normalized activations and gradients, respectively. Here, $\mathcal{T}_i$ is the set of dimensions over which normalization statistics are computed, and $C_i$ is the channels/features size.*

The proof of this proposition and the extended computation of other type of layers are given in Appendix A.2 and Section A.3, respectively. Note that in the context of online and stochastic optimization, the KFs for a given layer $i$ can be estimated using an Exponentially Moving Average

(EMA) scheme across batches defined by

$$\mathcal{H}_{i-1}^{(t)} = \gamma \mathcal{H}_{i-1}^{(t)} + (1-\gamma)\mathcal{H}_{i-1}^{(t-1)}, \; \mathcal{S}_i^{(t)} = \gamma \mathcal{S}_i^{(t)} + (1-\gamma)\mathcal{S}_i^{(t-1)}, \tag{3}$$

where $0 < \gamma \le 1$ is the exponential decay factor at step time $t$, $\mathcal{H}_{i-1}^{(t)}$ and $\mathcal{S}_i^{(t)}$ in the right-hand side are new KFs calculated during each forward and backward pass computation. This EMA scheme is commonly used in methods involving diagonal or block-diagonal approximations to the curvature matrix (e.g. LeCun et al. (2012); Park et al. (2000); Schaul et al. (2013)). Such schemes have the desirable property that they allow the curvature estimation to depend on much more data than what can be reasonably processed in a single mini-batch.

Our study from Section 3.1 suggests that the FIM's critical information predominantly resides along its diagonal. Building upon this, we propose a novel approximation for the FIM, described in Proposition 3.2, that conceptualizes the KFs as diagonal matrices denoted as $\tilde{F}_{D_i}$ for layer $i$.

**Proposition 3.2** (Efficient EFIM). *Assume that $\mathcal{H}_{i-1}$ and $\mathcal{S}_i$ can be closely approximated by diagonal matrices, denoted by $\mathcal{H}_{D_{i-1}}$ and $\mathcal{S}_{D_i}$ respectively at layer $i$, such that $\mathcal{H}_{D_{i-1}} = Diag(\mathcal{H}_{i-1})$, $\mathcal{S}_{D_i} = Diag(\mathcal{S}_i)$ where Diag denote the diagonal of a matrix. Accordingly, the Empirical FIM is defined by*

$$\tilde{F}_{D_i} \triangleq \mathcal{H}'_{D_{i-1}} \otimes \mathcal{S}'_{D_i} + \lambda \mathbf{I}, \tag{4}$$

*where $\mathcal{H}'_{D_{i-1}}$ and $\mathcal{S}'_{D_i}$ denote the Min-Max normalization of $\mathcal{H}_{D_{i-1}}$ and $\mathcal{S}_{D_i}$ (Patro & Sahu, 2015) and $\lambda$ is a regularization parameter.*

The proof of this proposition is given in Appendix A.2. This approximation strikes a balance between computational time and space complexity and the accuracy of performance, as discussed in Section 4. We set the regularization parameter $\lambda = 0.001$, which acts as a damping factor following the Tikhonov regularization principle Martens & Grosse (2015), enhancing computational stability and conditioning of the FIM. The closed-form solution for the preconditioned gradient $\bar{g}^{(t)}$ is derived from the diagonal approximation of the FIM, given by $\bar{g}^{(t)} = (\tilde{F}_D^{(t)})^{-1} g^{(t)}$, for time step $t$. This represents the AdaFisher augmented gradient and incorporates local loss *curvature information*. It focuses on the diagonal elements to reduce computational overhead while maintaining a reasonable FIM approximation. This simplification enhances the efficiency of the optimization process, which is crucial for training DNNs where computational resources are limited.

Table 1: Summary of the first, second moments, regret bound and Applicability used in Adam Kingma & Ba (2015), AdaHessian Yao et al. (2021), K-FAC Martens & Grosse (2020), Shampoo Gupta et al. (2018), and AdaFisher for updating model parameters $\theta^{(t+1)} = \theta^{(t)} - \alpha m^{(t)}/\sqrt{v^{(t)}}$. Here $\beta_1$ and $\beta_2$ are first and second moment HPs. $L^{(t)}$ and $R^{(t)}$ refer to the preconditioning method used by Shampoo Gupta et al. (2018), $g^{(t)} = \text{vec}(G^{(t)})$, and $T$ denotes the total number of steps. Note that Transf. denotes Transformers.

| Optimizer | $m^{(t)}$ | $v^{(t)}$ | Regret Bound | Applicability | |
| --- | --- | --- | --- | --- | --- |
| | | | | CNNs | Transf. |
| Adam | $\frac{(1-\beta_1)\sum_{i=1}^t \beta_1^{t-i} g_i}{1-\beta_1^t}$ | $\left(\frac{(1-\beta_2)\sum_{i=1}^t \beta_2^{t-i} g_i g_i}{1-\beta_2^t}\right)^{1/2}$ | $O(\log T \sqrt{T})$ | ✓ | ✓ |
| AdaHessian | $\frac{(1-\beta_1)\sum_{i=1}^t \beta_1^{t-i} g_i}{1-\beta_1^t}$ | $\left(\frac{(1-\beta_2)\sum_{i=1}^t \beta_2^{t-i} D_i^{(s)} D_i^{(s)}}{1-\beta_2^t}\right)^{1/2}$ | $O(\log T \sqrt{T})$ | ✓ | ✓ |
| K-FAC | $(\hat{F}^{(t)})^{-1} g^{(t)}$ | 1 | $O(\sqrt{T})$ | ✓ | ✗ |
| Shampoo | $(L^{(t)})^{\frac{-1}{4}} G^{(t)} (R^{(t)})^{\frac{-1}{4}}$ | 1 | $O(\sqrt{T})$ | ✓ | ✗ |
| AdaFisher | $\frac{(1-\beta_1)\sum_{i=1}^t \beta_1^{t-i} g_i}{1-\beta_1^t}$ | $\tilde{F}_D^{(t)}$ | $O(\log T \sqrt{T})$ | ✓ | ✓ |

### 3.3 Integrating FIM into Adaptive Optimization Framework

Following the spirit of the adaptive optimization framework from Adam, which combines momentum and RMSProp principles (Sutskever et al., 2013), the parameters are updated via $\theta^{(t+1)} = \theta^{(t)} - \alpha \frac{m^{(t)}}{v^{(t)}}$. Here, $\alpha$ represents the learning rate, while $m^{(t)}$ and $v^{(t)}$ denote the first and second moment estimates, respectively, for time step $t$. Although Adam is widely used, its approximation of the second moment using simple diagonal elements of second-order statistics through squared gradients can mirror stability challenges (Kunstner et al., 2019).

We overcome these challenges by utilizing a more refined diagonal block-Kronecker approximation of the FIM introduced in Section 3.2, a more precise approximation of the Hessian from Taylor series expansion viewpoint. This structured alternative to Adam's diagonal approximation enables precise curvature estimation, mitigating stability issues and improving convergence in non-convex settings. AdaFisher distinguishes itself from Adam by incorporating a higher fidelity approximation of the FIM, enhancing both optimization efficiency and model robustness in complex scenarios. As demonstrated in Figure 4, AdaFisher's FIM values exhibit narrow variations and lower mean values during training, suggesting a convergence towards **flatter minima** and an **implicit regularization** effect. In contrast, Adam shows broader variations, indicating less efficient generalization (Wang et al., 2024), for more details regarding the convergence behavior refer to Appendix B.1. Moreover, AdaFisher omits the square root and the traditional EMA applied over the second moment since the FIM naturally incorporates an EMA of its KFs (detailed in Eq. (3)). The exclusion of the square root aligns with the theoretical definition of second-order methods, as using a square root deviates from the second-order Taylor expansion approximation that these methods aim to follow. A comparative summary of different moment estimates, $m^{(t)}$ and $v^{(t)}$, along with their regret bounds and applicability across various optimizers, is presented in Table 1. Building on the principles of AdamW (Loshchilov & Hutter, 2019b), which modifies Adam by integrating weight decay directly into the weight update step to counteract suboptimal decay behaviors and boost optimization performance, we introduce AdaFisherW. This variant adapts the AdamW framework to further enhance the optimizer by leveraging curvature information from AdaFisher. Finally, AdaFisher is compatible with **multi-GPU environments**, with a distributed version detailed in Appendix A.4. The implementation for both AdaFisher variants is delineated in the pseudo-code presented in Algorithm 1.

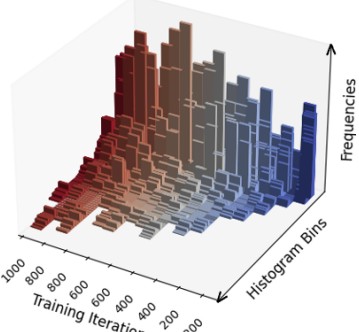

**Histogram of FIM Diagonal for Adam**

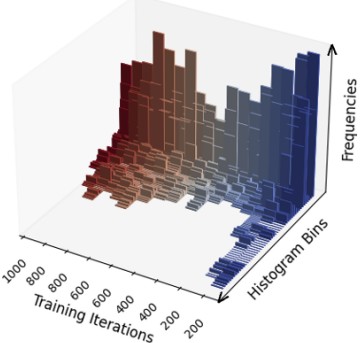

**Histogram of FIM Diagonal for AdaFisher**

Figure 4: Comparison of FIM diagonal histograms during ResNet18 training on CIFAR10: The figure displays the FIM diagonal elements for the first convolutional layer with Adam and AdaFisher over 1,000 training iterations.

### 3.4 CONVERGENCE ANALYSIS

In this section, we provide a theoretical analysis of AdaFisher's convergence in both convex optimization and non-convex stochastic optimization. We first present a standard convergence behavior of Eq. (2) for a simple strongly convex and strictly smooth function $f(J)$.

**Proposition 3.3** (Convergence in convex optimization). *For FIM defined in Eq. (4), the updating scheme $\theta^{(t+1)} = \theta^{(t)} - \alpha(\tilde{F}^{(t)})^{-1}\nabla J(\theta^{(t)})$ converges. Further, if $\nabla J$ is Lipschitz, the convergence rate is bounded.*

For non-convex cases, we adopt the similar derivations of Chen et al. (2019) since AdaFisher belongs to the family of generalized Adam-type methods.

**Proposition 3.4** (Convergence in non-convex stochastic optimization). *Under the assumptions: (i) $J$ is lower bounded and differentiable; $||\nabla J(\theta) - \nabla J(\theta')||_2 \leq L||\theta - \theta'||_2$, $||\tilde{F}_D^{(t)}||_\infty < L$, $\forall t, \theta, \theta'$, (ii) Both the true and stochastic gradient are bounded, i.e. $||\nabla J(\theta^{(t)})||_2 \leq \lambda$ and $||g^{(t)}||_2 \leq \lambda$, $\forall t$ for some $\lambda > 0$, (iii) Unbiased and independent noise in $g^{(t)}$, i.e. $g^{(t)} = \nabla J(\theta^{(t)}) + \zeta^{(t)}$, $\mathbb{E}[\zeta^{(t)}] = 0$, and $\zeta^{(i)} \perp \zeta^{(j)}$, $\forall i \neq j$. Assume $\eta^{(t)} = \frac{\eta}{\sqrt{t}}$, $\beta^{(t)} \leq \beta \leq 1$ is non-increasing, $\frac{\tilde{F}_D^{(t-1)}[j]}{\eta^{(t-1)}} \leq \frac{\tilde{F}_D^{(t)}[j]}{\eta^{(t)}}$, $\forall t \in [T], j \in [d]$, we then have*

$$\min_{t\in[T]} \mathbb{E}[||\nabla J(\theta^{(t)})||_2^2] \leq \frac{L}{\sqrt{T}}(C_1\eta^2\lambda^2(1+\log T) + C_2 d\eta + C_3 d\eta^2 + C_4)$$

where $C_1, C_2, C_3$ are constants independent of $d$ and $T$, $C_4$ is a constant independent of $T$, the expectation is taken with respect to all the randomness corresponding to $\{g^{(t)}\}$.

The proofs of propositions 3.3 and 3.4 are given in Appendix A.2. Proposition 3.4 implies the convergence rate for AdaFisher in the non-convex case is at $O(\log T/\sqrt{T})$, which is similar to Adam-type optimizers. While DNNs often include non-smooth components like ReLU and max pooling, which create non-differentiable points in the loss landscape, optimizers like AdaFisher handle these cases effectively, as shown by our results in Section 4.

---

**Algorithm 1** AdaFisher optimization algorithm. Good default settings for the tested machine learning problems are $\alpha = 0.001$ (learning rate), $\lambda = 0.001$ (Tikhonov damping parameter),$\gamma = 0.8$ (Exponentially decaying factor). [Default parameters are: $\beta = 0.9$ (Exponentially decaying factor of Adam), $\kappa$ (weight decay) (Kingma & Ba (2015), Loshchilov & Hutter (2019b))].

---
**Require:** Step size $\alpha$; Exponential decay rate for KFs $\gamma \in [0, 1)$; Tikhonov damping parameter $\lambda$; Exponential decay rate for first moments $\beta$ in $[0, 1)$; Initial parameters $\theta$
    **Initialize** 1st moment variable $m = 0$; FIM $\tilde{F}_{D_i} = \mathbf{I}$; time step $t = 0$
1: **while** stopping criterion not met **do**
2:     Sample a minibatch of $M$ examples from the training set $\{(x_n, y_n)\}_{n=1}^{M}$
3:     Compute $\mathcal{H}_{D_{i-1}}, \mathcal{S}_{D_i}$ for $i \in \{1, \ldots, L\}$ using Section A.3 (notice that: $\mathcal{H}_{D_0} = x$)
4:     Compute EMAs of $\mathcal{H}_{D_{i-1}}$ and $\mathcal{S}_{D_i}$ using Eq. (3)
5:     Compute $\tilde{F}_{D_i}$ for $i \in \{1, \ldots, L\}$ using Eq. (4)
6:     $g^{(t)} \leftarrow \frac{1}{M} \sum_n \nabla_{\theta^{(t)}} \mathcal{L}(f(x_n; \theta^{(t)}), y_n)$ (Compute gradient)
7:     $m^{(t+1)} \leftarrow \frac{\beta m^{(t)} + (1-\beta)h^{(t)}}{1 - \beta^t}$ (Update and correct biased first moment)
8:     **Case AdaFisher:** $\Delta\theta^{(t)} = -\alpha(\tilde{F}_D^{(t)})^{-1}m^{(t)}$
       **Case AdaFisherW:** $\Delta\theta^{(t)} = -\alpha\left((\tilde{F}_D^{(t)})^{-1}m^{(t)} + \kappa\theta^{(t)}\right)$
9:     $\theta^{(t+1)} \leftarrow \theta^{(t)} + \Delta\theta^{(t)}$ (Apply update)
10:    $t \leftarrow t + 1$
11: **end while**

---

## 4 RESULTS

To evaluate AdaFisher, we conduct experiments on six benchmark datasets across Image Classification for Computer Vision (CV) and Language Modeling for Natural Language Processing (NLP) that are commonly used to evaluate optimization algorithms: CIFAR-10, CIFAR100 (Krizhevsky et al., 2009), Tiny ImageNet (Le & Yang, 2015), and ImageNet-1k (Deng et al., 2009) for image classification; Wikitext-2 (Merity et al., 2017) and Penn Treebank (PTB) (Marcus et al., 1993) for language modeling. The six baseline methods we compare with are SGD, Adam/AdamW, K-FAC, AdaHessian, and Shampoo. For CIFAR experiments, we report the average over five runs. We also perform a transfer learning task using the ImageNet-1k weights from Paszke et al. (2019). Detailed descriptions of the experimental setup (including HP tuning, datasets, and data augmentation), results, and analyses are provided in Appendix D.

Table 2: Performance metrics (mean, std) of different networks and optimizers on CIFAR10 and CIFAR100 using batch size 256 with a 200-epoch AdaFisher training cutoff.

| Network | CIFAR10 | | | | | | CIFAR100 | | | | | |
|---|---|---|---|---|---|---|---|---|---|---|---|---|
| | SGD | Adam | AdaHessian | K-FAC | Shampoo | AdaFisher | SGD | Adam | AdaHessian | K-FAC | Shampoo | AdaFisher |
| ResNet18 | $95.64_{0.1}$ | $94.85_{0.1}$ | $95.44_{0.1}$ | $95.17_{0.2}$ | $94.08_{0.2}$ | $\mathbf{96.25_{0.2}}$ | $76.56_{0.2}$ | $75.74_{0.1}$ | $71.79_{0.2}$ | $76.03_{0.3}$ | $76.78_{0.2}$ | $\mathbf{77.28_{0.2}}$ |
| ResNet50 | $95.71_{0.1}$ | $94.45_{0.2}$ | $95.54_{0.1}$ | $95.66_{0.1}$ | $94.59_{0.1}$ | $\mathbf{96.34_{0.2}}$ | $78.01_{0.1}$ | $74.65_{0.5}$ | $75.81_{0.3}$ | $77.40_{0.4}$ | $78.07_{0.4}$ | $\mathbf{79.77_{0.4}}$ |
| ResNet101 | $95.98_{0.2}$ | $94.57_{0.1}$ | $95.29_{0.6}$ | $96.01_{0.1}$ | $94.63_{0.1}$ | $\mathbf{96.39_{0.1}}$ | $78.89_{0.2}$ | $75.56_{0.3}$ | $73.38_{0.2}$ | $77.01_{0.4}$ | $78.83_{0.2}$ | $\mathbf{80.65_{0.4}}$ |
| DenseNet121 | $96.09_{0.1}$ | $94.86_{0.1}$ | $96.11_{0.1}$ | $96.12_{0.1}$ | $95.66_{0.1}$ | $\mathbf{96.72_{0.1}}$ | $80.13_{0.4}$ | $75.87_{0.4}$ | $74.80_{0.9}$ | $79.79_{0.2}$ | $80.24_{0.3}$ | $\mathbf{81.36_{0.3}}$ |
| MobileNetV3 | $94.43_{0.2}$ | $93.32_{0.1}$ | $92.86_{3.1}$ | $94.34_{0.1}$ | $93.81_{0.2}$ | $\mathbf{95.28_{0.1}}$ | $73.89_{0.3}$ | $70.62_{0.3}$ | $56.58_{4.5}$ | $73.75_{0.3}$ | $70.85_{0.3}$ | $\mathbf{77.56_{0.1}}$ |
| Tiny Swin | $82.34_{0.2}$ | $87.37_{0.6}$ | $84.15_{0.2}$ | $64.79_{0.5}$ | $63.91_{0.4}$ | $\mathbf{88.74_{0.4}}$ | $54.89_{0.4}$ | $60.21_{0.4}$ | $56.86_{0.5}$ | $34.45_{0.4}$ | $30.39_{1.2}$ | $\mathbf{66.05_{0.5}}$ |
| FocalNet | $82.03_{0.2}$ | $86.23_{0.1}$ | $64.18_{0.2}$ | $38.94_{0.8}$ | $37.96_{0.7}$ | $\mathbf{87.90_{0.1}}$ | $47.76_{03}$ | $52.71_{0.5}$ | $32.33_{0.3}$ | $9.98_{0.6}$ | $9.18_{0.1}$ | $\mathbf{53.69_{0.3}}$ |
| CCT-2/3×2 | $78.76_{0.3}$ | $83.89_{0.4}$ | — | $33.08_{2.3}$ | $35.16_{0.4}$ | $\mathbf{84.94_{0.3}}$ | $54.05_{0.4}$ | $59.78_{0.5}$ | — | $7.17_{0.2}$ | $8.60_{0.1}$ | $\mathbf{62.91_{0.5}}$ |

[*] Note that Adam and AdaFisher were used for all CNN architectures, while AdamW and AdaFisherW were applied for all ViT experiments.

## 4.1 IMAGE CLASSIFICATION

We commence our analysis by assessing the convergence and generalization capabilities of various models on image classification tasks. Specifically, we deploy ResNet architectures (ResNetX where $X \in \{18, 50, 101\}$), DenseNet121 (Huang et al., 2017), MobileNetV3 (Howard et al., 2019), Tiny Swin (Liu et al., 2021), FocalNet (Yang et al., 2022) and CCT-2/3×2 (Hassani et al., 2021) on CIFAR10 and CIFAR100, while utilizing standard ResNet50 for Tiny ImageNet and ImageNet-1k. The performance outcomes for CIFAR datasets are detailed in Table 2. Our empirical evaluation of AdaFisher optimizer across these models and datasets illustrates its efficiency in optimizing image classification, surpassing all SOTA optimizers. We employ the Wall-Clock-Time (WCT) method with a cutoff of 200 epochs for AdaFisher's training, except for ImageNet-1k, where we use a 90-epoch WCT for Adam, which surprisingly matched AdaFisher's training duration. Results confirm AdaFisher's superior classification accuracy on both CNNs and ViTs. Note that the results for Tiny ImageNet are described in Appendix D.2.4.

**ImageNet-1k Training.** Training on ImageNet-1k typically requires multiple GPUs and large batch sizes. Our study showcases that AdaFisher achieves superior validation accuracy on a single GPU than its counterparts in scenarios marked by the light blue region. This performance outstrips traditional approaches like SGD, LAMB (You et al., 2020), and LARS (You et al., 2017), which typically utilize batch sizes of 16K. While AdaFisher attains SOTA results on a single GPU, it further excels when scaled up in a distributed setting with larger batch sizes. The results, benchmarked using 256 batch size and a WCT of 90 Adam training epochs, are detailed in Table 3 and illustrated in Figure 5. Distributed AdaFisher curves are illustrated in Figure 15. The light blue highlights in the table represent our experiments with a batch size of 256 on a single GPU. The light green indicates results from a distributed version of AdaFisher employing larger batch sizes, whereas the orange reflects results from SOTA methods using a higher batch size of 16K, SGD with a batch size of 256 and AdamW with a batch size of 1024. It is important to note, however, that the training setups and augmentation techniques for the results highlighted in orange, taken from the literature, may differ from those in our study. These results are included to provide a broader context and intuition regarding AdaFisher's performance compared to other experiments. Overall, by balancing curvature-aware updates with parameter efficiency, AdaFisher maintains strong generalization even under constrained computational budgets, a critical advantage over methods like K-FAC that struggle with over-parameterized models.

Table 3: Validation of ImageNet-1k / ResNet50 by different optimizers reported on Top-1 and Top-5 accuracy.

| Optimizers | Batch size | Top-1 | Top-5 |
|---|---|---|---|
| Adam | 256 | 67.78 | 88.37 |
| K-FAC | 256 | 70.96 | 89.44 |
| Shampoo | 256 | 72.82 | 91.42 |
| AdaFisher | 256 | **76.95** | **93.39** |
| AdaFisher | 512 | **77.01** | **93.45** |
| AdaFisher | 1024 | **77.09** | **93.56** |
| SGD Goyal et al. (2017) | 256 | 76.40 | - |
| AdamW Chen et al. (2024) | 1024 | 76.34 | - |
| LAMB You et al. (2020) | 16K | 76.66 | 93.22 |
| SGD You et al. (2020) | 16K | 75.20 | - |
| LARS Huo et al. (2021) | 16K | 75.1 | - |

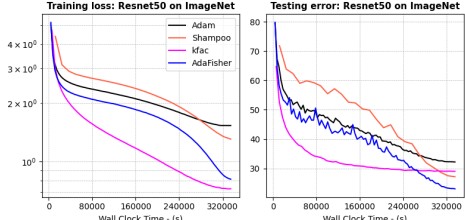

Figure 5: Training loss and validation error of ResNet-50 on ImageNet-1k. AdaFisher consistently achieves lower test error as compared to its counterparts.

Table 4: Performance comparison of different networks and optimizers on CIFAR10 and CIFAR100 using ImageNet-1k pretrained weights. Evaluation is based on wall clock time of 50 training epochs with AdaFisher.

| Network | \multicolumn{6}{c}{CIFAR10} | | | | | | \multicolumn{6}{c}{CIFAR100} | | | | | |
|---|---|---|---|---|---|---|---|---|---|---|---|---|
| | SGD | Adam | AdaHessian | K-FAC | Shampoo | AdaFisher | SGD | Adam | AdaHessian | K-FAC | Shampoo | AdaFisher |
| ResNet50 | $96.50_{0.2}$ | $96.45_{0.2}$ | $96.35_{0.3}$ | $96.45_{0.1}$ | $96.03_{0.4}$ | $\mathbf{97.13_{0.2}}$ | $82.12_{0.1}$ | $82.01_{0.4}$ | $80.64_{0.9}$ | $80.55_{0.4}$ | $81.70_{0.2}$ | $\mathbf{82.23_{0.2}}$ |
| ResNet101 | $97.07_{0.2}$ | $96.70_{0.1}$ | $96.65_{0.2}$ | $96.84_{0.1}$ | $96.63_{0.1}$ | $\mathbf{97.22_{0.1}}$ | $84.01_{0.1}$ | $82.43_{0.2}$ | $81.36_{0.8}$ | $82.26_{0.3}$ | $82.65_{0.2}$ | $\mathbf{84.47_{0.2}}$ |
| DenseNet121 | $94.80_{0.1}$ | $94.77_{0.1}$ | $93.08_{0.1}$ | $94.41_{0.2}$ | $94.76_{0.1}$ | $\mathbf{95.03_{0.1}}$ | $75.98_{0.2}$ | $75.65_{0.3}$ | $71.06_{0.9}$ | $76.10_{0.3}$ | $76.08_{0.2}$ | $\mathbf{76.92_{0.3}}$ |
| MobileNetV3 | $91.76_{0.3}$ | $90.92_{0.3}$ | $86.45_{2.5}$ | $91.72_{0.2}$ | $91.39_{0.3}$ | $\mathbf{92.78_{0.2}}$ | $71.86_{0.4}$ | $66.11_{0.8}$ | $59.69_{2.3}$ | $69.85_{0.4}$ | $68.87_{0.3}$ | $\mathbf{72.38_{0.4}}$ |

## 4.2 TRANSFER LEARNING

Following a more sustainable practice of training DNNs, we employ pretrained models from ImageNet-1k in PyTorch on datasets like CIFAR10 and CIFAR100 to showcase AdaFisher's gener-

alization capability for transfer learning. We applied these pretrained weights across various CNN architectures to train on these datasets. The results, presented in Table 4, highlight the significant advantages of using AdaFisher, consistently achieving top accuracy across both datasets. More details can be found in Appendix D.2.3.

## 4.3 LANGUAGE MODEL

We employ the WikiText-2 dataset, which encompasses approximately 100 million tokens derived from over 2 million words extracted from a curated set of "Good" and "Featured" articles on Wikipedia. Additionally, we utilize the PTB dataset, renowned for its extensive collection of English words with part-of-speech tags, which has been widely used in NLP tasks for training and benchmarking language models. Our experiments utilize a scaled-down version of GPT-1 (Radford et al., 2019), featuring four self-attention layers with masking capabilities with more than 28 million learnable parameters. More details about tuning HPs and models can be found in Appendix D.3. The perplexity (PPL) on the test set, corresponding to the best-performing model during validation, is

Table 5: Language Modeling performance (PPL) on Wikitext-2 and PTB test dataset (lower is better).

| Optimizer | Test PPL | |
|---|---|---|
| | WikiText-2 | PTB |
| AdamW | 175.06 | 44.70 |
| AdaHessian | 407.69 | 59.43 |
| Shampoo | 1727.75 | — |
| AdaFisherW | **152.72** | **41.15** |

documented in Table 5. Similar to approaches in image classification, we apply the WCT method with 50 epochs training time of AdaFisher as the cutoff period. Notice that Shampoo did not achieve convergence despite using optimal HPs, and the K-FAC was unable to train with ASDL library (Osawa et al., 2023).

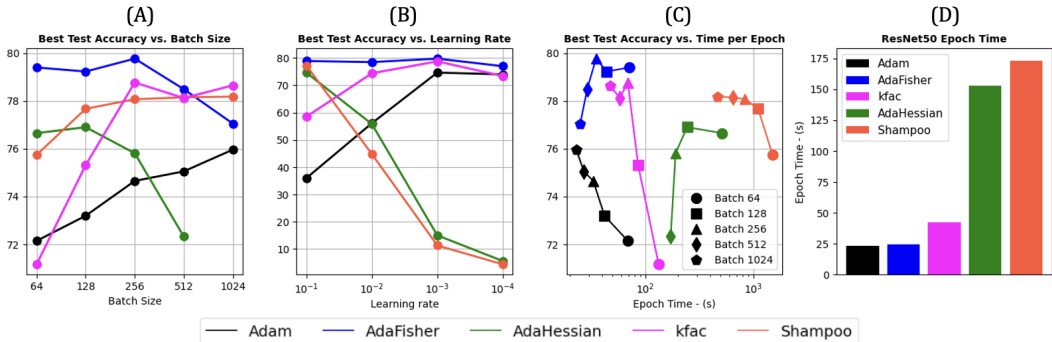

Figure 6: Performance comparison of AdaFisher and other optimizers using the ResNet50 network on the CIFAR100 dataset. (A) Test accuracy by batch size. (B) Accuracy vs. learning rates. (C) Accuracy related to epoch time across batch sizes. (D) Epoch time for different optimizers with a batch size of 256.

## 4.4 STABILITY ANALYSIS

In this section, we assess AdaFisher's stability under varying learning rates and batch sizes using ResNet50 on CIFAR100 and compare its performance to other optimizers. Improved stability indicates a reduced need for HP tuning while maintaining high performance. To ensure a fair comparison, all methods were evaluated using a consistent experimental setup, with parameters tailored to each optimizer's strengths. However, we exclude AdaHessian results for a batch size of 1024 due to its significant computation cost.

**Batch Size Analysis.** We examine the impact of batch size on AdaFisher's performance, as shown in Panels (A) and (C) of Figure 6. AdaFisher maintains high test accuracy across various batch sizes, excelling particularly at smaller sizes despite some sensitivity to larger ones. Panel (C) highlights AdaFisher's efficiency, achieving high accuracy with shorter epoch times compared to Adam, detailed further in Panel (D), where AdaFisher shows competitive epoch durations against other optimizers. These results, discussed in Appendix D.2.7, underscore AdaFisher's effective performance across batch size variations without adjusting other HPs.

**Learning Rate Stability.** This analysis evaluates the impact of learning rate variations on AdaFisher's performance, as depicted in Panel (B) of Figure 6. AdaFisher demonstrates superior stability, particularly at lower learning rates, maintaining consistent performance across a broad

spectrum. This stability alleviates the need for meticulous learning rate adjustments, thereby stream-lining model training in various computational environments. Additionally, AdaFisher's stability across various learning rates can be attributed to its effective approximation of the curvature matrix.

**Ablation Studies.** We further conduct extensive ablation studies on additional components of AdaFisher, including the convergence efficiency, our novel approximation of the FIM, the significance of EMA for Kronecker factors, the impact of the square root, the stability across learning rate schedulers and the updated computation of the FIM for normalization layers. These analyses are thoroughly detailed in Appendix B.

## 5 RELATED WORK

**Tractable Approximation of the FIM.** Efficient approximations of the FIM for neural network optimization have evolved significantly, beginning with block-diagonal strategies exemplified by TONGA (Roux et al., 2007) and extending to the Kronecker-factored approaches like K-FAC. Further innovations have emerged, such as SK-FAC (Tang et al., 2021), EVA (Zhang et al., 2023), which accelerates the computation of the FIM, and Eschenhagen et al. (2024), who propose a generalized framework for FIM computation that enhances preconditioning methods like Shampoo. More recently, Liu et al. (2024), Huang et al. (2024) and Duvvuri et al. (2024) have introduced tractable solutions for computing the FIM. AdaFisher distinguishes itself by integrating enhanced FIM computations with novel diagonal Kronecker factors, enriching the Adam optimization framework. This integration, outlined in Proposition 3.2 and detailed in Appendix A.3, advances the fusion of second-order optimization principles with first-order methods. This builds upon innovations like AdaHessian, which incorporates Hessian diagonals into the Adam framework.

**Adaptive First-Order Methods.** Building upon the diagonal approximation heritage of the FIM, AdaFisher extends traditional diagonally-scaled first-order methods such as AdaGrad (Duchi et al., 2011), AdamP, AdaInject, AdaBelief, and Adam. These methods have inspired advancements like AMSGrad (Reddi et al., 2018), AdaBound (Luo et al., 2019), RAdam (Liu et al., 2020), and enhanced AdaBelief, FOOF (Benzing, 2022), improving both theoretical rigor and practical effectiveness. A recent study by Jiang et al. (2024a) illustrates that first-order adaptive methods can bias training trajectories and effectively navigate through saddle points. In response, Mishchenko & Stich (2023) propose an empirical solution involving the addition of noise to mitigate these biases. Leplat et al. (2022) introduces a novel method accelerating convergence via Gauss-Seidel type discretization. AdaFisher differentiates itself by eliminating the conventional square root in the second moment calculation, with benefits underscored by Lin et al. (2024) and Malladi et al. (2022) in CNN architectures, and Zhang et al. (2024) demonstrates the critical role of Adam family optimizers in Transformer models. Its unique preconditioning, based on the Fisher Information, is elaborated in Algorithm 1.

## 6 CONCLUSION, LIMITATIONS AND FUTURE RESEARCH

In this work, we introduced AdaFisher–an adaptive optimizer that utilizes the Fisher Information Matrix (FIM) with a new diagonal block-Kronecker approximation to enhance gradient rescaling and improve descent directions. Integrated within the adaptive optimization framework, AdaFisher not only accelerates training but also minimizes the need for hyper-parameter tuning, thereby achieving higher accuracy and stability in tasks like image classification and language modeling. Empirical and theoretical analyses confirm its superiority over existing optimizers, and its ease of implementation, along with modest space and time requirements, allows it to adapt across various tasks. Notably, AdaFisher outperforms SOTA optimizers on ImageNet-1k when trained using both single and multi-GPU configurations. AdaFisher is optimized for statistical learning tasks where the deep neural network outputs parameterize distributions within the exponential family, and the loss function is the negative log-likelihood (or its derivatives). Outside these conditions, where the FIM no longer coincides with the Generalized Gauss-Newton matrix, the method may not capture the true curvature information of the loss landscape.

Looking ahead, extensive testing across a wider range of models and domains, including generative modeling and graph neural networks, will further validate AdaFisher's effectiveness. Additionally, implementing CUDA kernels for efficient computation of Kronecker factors could significantly enhance AdaFisher's scalability and performance.

ACKNOWLEDGMENT

This research is funded by Fonds de recherche du Québec (FRQNT) Masters (B1X) Scholarship [D.M.G]; Natural Sciences & Engineering Research Council (NSERC)-Discovery Grant RGPIN-2022-05378 [M.S.H.]; FRQNT-NSERC grant 2023-NOVA-329125 [E.B.& G.W.].

IMPACT STATEMENT

AdaFisher represents a significant advancement in training efficiency, achieving superior accuracy on the ImageNet-1K dataset using only a single GPU. This optimization is particularly beneficial for academia and students who may not have access to extensive computational resources. By enabling effective training with fewer GPUs, AdaFisher offers an accessible yet powerful solution, reducing hardware costs and making advanced machine learning more attainable for those with limited resources. This capability underscores AdaFisher's potential as a valuable tool in democratizing machine learning technology.

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

# APPENDIX

CONTENTS

# A  THEORY

## A.1  KFs: A STRUCTURAL EXAMINATION (CONTINUE)

In the realm of matrix theory, the Gershgorin circle theorem offers a principle for localizing the eigenvalues of a complex square matrix, asserting that each eigenvalue is situated within at least one Gershgorin disk. These disks are defined by the matrix's diagonal elements and the sum of the absolute values of the respective off-diagonal row entries. Formally, the theorem is stated as follows:

**Theorem A.1** ( Gershgorin Circle Theorem). *Let $\mathcal{A}$ be a complex square matrix with eigenvalues $\lambda$. For each $\lambda$, there exists an index $i$ such that*

$$|\lambda - \mathcal{A}_{ii}| \leq \sum_{\substack{j=1 \\ j \neq i}}^{n} |\mathcal{A}_{ij}|,$$

*where the summation excludes the diagonal entry $\mathcal{A}_{ii}$.*

For a detailed proof of Theorem A.1, the reader is referred to the seminal work by Horn and Johnson Horn & Johnson (2012). Extending the application of the Gershgorin circle theorem to the study of KFs within deep neural networks, we analyze these factors from both convolutional (37th) and linear (41st) layers of a ResNet-18 network through different training phases on CIFAR10 dataset. As elucidated in Section 3.1, leveraging Theorem A.1 demonstrates that the eigenvalues of the KFs from the convolutional layer are predominantly concentrated along the diagonal. This observation is analogously applicable to the linear layer. Figure 3 showcases the Gershgorin disks for the 41st (linear) layer, with the eigenvalues (red crosses) significantly clustered within these disks (centered at the black circles), underscoring a pronounced diagonal dominance. Moreover, upon introducing Gaussian noise to the off-diagonal elements following this scheme: $\hat{\mathcal{M}} = \mathcal{A} + \mathcal{E}$, where $\mathcal{E} = [e_{ij}]$ and $e_{ij} \sim \mathcal{N}(0, \sigma^2)$ for $i \neq j$, the perturbation analysis elucidates that such stochastic variations engender only marginal displacements in the eigenvalues. Notably, those eigenvalues fulfilling the Kaiser criterion are minimally affected, substantiating the resilience of the diagonal dominance against noise-induced perturbations. Our next analysis focus centers on

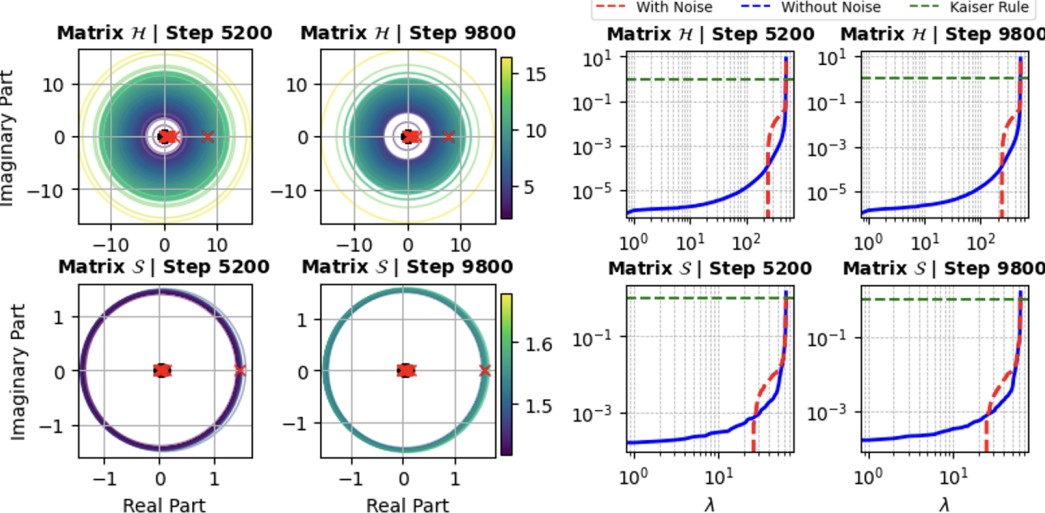

Figure 7: Gershgorin disks and eigenvalue perturbation analysis for matrices $\mathcal{H}$ and $\mathcal{S}$ at training steps 5200 (middle of training) and 9800 (end of training) in a ResNet-18 Network's Linear Layer (41st Layer). The left panel depicts Gershgorin's circles in the complex plane, while the right panel illustrates the magnitude spectrum of eigenvalues with and without the influence of Gaussian noise.

elucidating the behaviors of matrices through consecutive steps in the frequency domain, thereby highlighting the intricate patterns and transformations emergent from the training process. By deploying a Fast Fourier Transform (FFT) on $\mathcal{H}$ and $\mathcal{S}$, along with their noise-infused variants $\hat{\mathcal{H}}$ and $\hat{\mathcal{S}}$, we aim to dissect the spectral nuances of these factors. The deliberate addition of noise to the

off-diagonal serves as a probe to validate our hypothesis that the pivotal information of the KFs is predominantly concentrated along their diagonals. The minimal impact of such noise perturbations observed empirically underscores this diagonal dominance. Our analysis aims to juxtapose the frequency domain representations of both the uncontaminated and the noise-affected matrices at assorted iterative phases, thereby illuminating the inherent stability and tenacity of the Kronecker structures amidst stochastic disturbances.

Let $A$ be a two-dimensional $m \times n$ matrix. The FFT of $A$, denoted as $\mathcal{F}(A)$, is computed as

$$\mathcal{F}(A)_{kl} = \sum_{p=0}^{m-1} \sum_{q=0}^{n-1} A_{pq} \cdot e^{-2\pi i \left( \frac{pk}{m} + \frac{ql}{n} \right)}, \tag{5}$$

where $\mathcal{F}(A)_{kl}$ is the value of the FFT at the $k$th row and $l$th column of the transformed matrix, $A_{pq}$ is the value of the original matrix at the $p$th row and $q$th column, and $i$ is the imaginary unit (Oppenheim et al., 1999). Figure 8 demonstrates the Fourier spectral analysis of the KFs $\mathcal{H}$ and $\mathcal{S}$

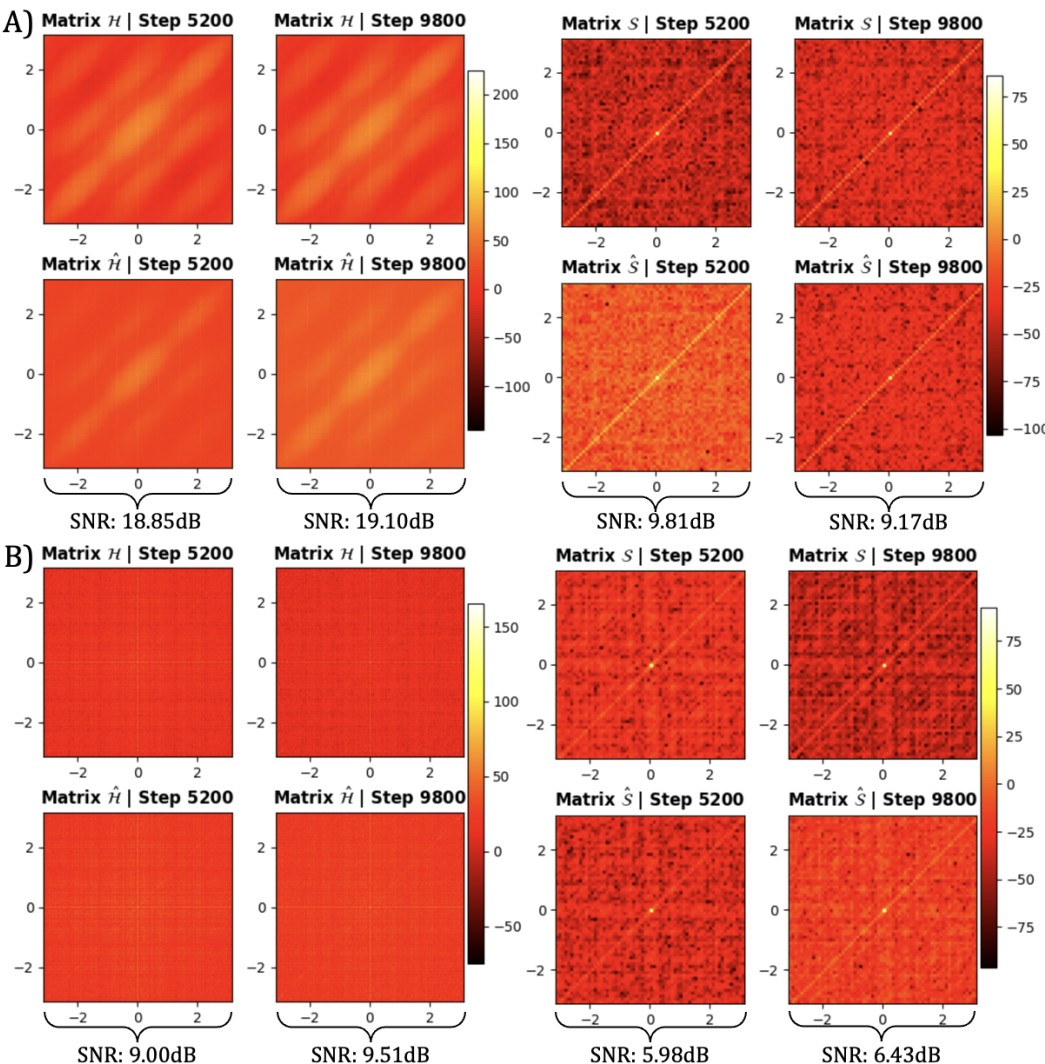

Figure 8: Comparative Visualization of FFT Outputs for KFs in a ResNet-18 Network's Convolutional and Linear Layers. (A) FFT results for KFs $\mathcal{H}$ and $\hat{\mathcal{H}}$ from the 37th convolutional layer under noise-free conditions (top) and with Gaussian noise (bottom) at iterations 5200 (middle of training) and 9800 (end of training). (B) Analogous FFT results for the KFs $\mathcal{S}$ and $\hat{\mathcal{S}}$ from the 41st linear layer, also contrasted between noise-free (top) and noisy conditions (bottom) at the same iterations.

over two distinct iterative stages of training—5200 and 9800 for a convolutional and a linear layers

(37th and 41st of a ResNet-18 network respectively). Each KF is analyzed via FFT in both a pristine, noise-free condition and a Gaussian noise-affected state, with the associated Signal-to-Noise Ratios (SNRs) detailed in Eq. (6). In the noise-free FFT spectra, a pronounced diagonal energy concentration is manifest in the $\mathcal{H}$ and $\mathcal{S}$ factors of the convolutional layer, indicative of significant informational preservation along the diagonal. In contrast, the linear layer exhibits a less pronounced but still discernible diagonal energy distribution, suggesting a more diffuse yet still noteworthy diagonal information structure. With the addition of noise, the matrices $\hat{\mathcal{H}}$ and $\hat{\mathcal{S}}$ still display a notable diagonal pattern, indicating minimal SNR deterioration. This observation supports the proposition that the KFs primarily encode their information along the diagonal, and the introduction of noise into the off-diagonal elements has a limited impact. The SNR between a matrix $\mathcal{M}$ and $\hat{\mathcal{M}}$ is computed using the formula:

$$\text{SNR} = 10 \cdot \log_{10}\left(\frac{\sum_{i=1}^{N}|\mathcal{M}_{ii}|^2}{\sum_{j>i}^{N}|\hat{\mathcal{M}}_{ij}|^2}\right), \tag{6}$$

where $\mathcal{M}_{ii}$ denotes the diagonal elements of $\mathcal{M}$, and $\hat{\mathcal{M}}_{ij}$ represents the upper triangular elements of $\hat{\mathcal{M}}$ excluding the diagonal (Oppenheim et al., 1999). The observed reduction in SNR from step 5200 to step 9800 for the KF $\mathcal{S}$ in the convolutional layer, under noisy conditions, could suggest an incremental integration of noise effects across iterations. Conversely, for the remaining factors, an increase in SNR throughout the training process is detected, which may indicate an enhancement in signal clarity. Nevertheless, the integrity of the diagonal concentration of energy remains predominantly intact, demonstrating the underlying robustness of the network's feature extraction capability against noise perturbations. Ultimately, the spectral analyses validate the hypothesis that the KFs' informational content is predominantly diagonal and resistant to the effects of off-diagonal Gaussian noise. This durability is sustained through successive iterations, maintaining the primary spectral characteristics of the KFs. Figure 9 offers a visual exposition of the Kronecker Product Factors $\mathcal{H}$

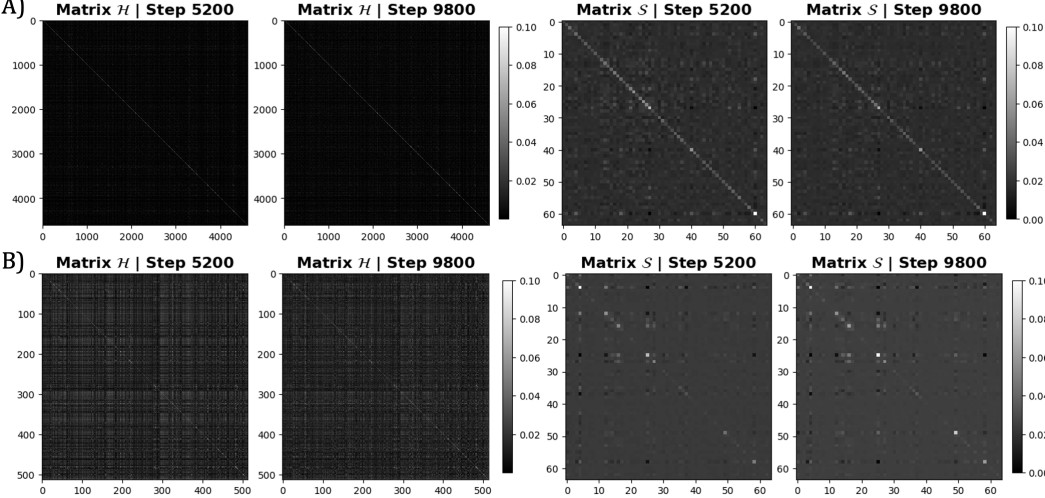

Figure 9: Visualization of KFs $\mathcal{H}$ and $\mathcal{S}$ for convolutional (A) and linear (B) layers at different iteration steps within a ResNet-18 network. For the convolutional layer (37th layer), the first two plots in (A) represent factor $\mathcal{H}$ at steps 5200 (middle of training) and 9800 (end of training), elucidating the matrix's structure at these stages. The subsequent two plots display factor $\mathcal{S}$, highlighting changes in granularity and contrast with iteration progression. Similarly, in (B) for the linear layer (41st position), we observe the structural evolution of factor $\mathcal{H}$ and $\mathcal{S}$ over the same iterations, with variations in pattern density and clarity. These visualizations collectively underscore the dynamic nature of the KFs' architecture as training advances.

and $\mathcal{S}$ at progressive iteration junctures—specifically steps 5200 and 9800 for a convolutional and a linear layers (37th and 41st of a ResNet-18 network respectively). The initial duo of plots in each (A) and (B), delineate the KF $\mathcal{H}$ at the aforementioned steps, elucidating the matrix's structure at two distinct evolutionary stages. The next duo plots in (A) and (B) represent the KF $\mathcal{S}$ at different steps of training. This visual examination, in conjunction with the preceding spectral analyses, articulates an integrated story of the developmental trajectory of the KFs. The enduring diagonal salience observed in both $\mathcal{H}$ and $\mathcal{S}$ underscores the notion that the informational energy of the KFs

is predominantly concentrated along the diagonal. This persistent feature accentuates the structural stability and the focused nature of information encoding within the network's layers.

## A.2 PROOFS

**Proposition A.1** (FIM for normalization layer). *Let $(\nu_i, \beta_i) \in \mathbb{R}^{C_i}$ be the scale and shift parameters of a normalization layer $i$. The empirical KFs for the FIM approximation are*

$$\mathcal{H}_{i-1}\Big|_{\nu_i} = \frac{1}{|\mathcal{T}_i|} \sum_{x \in \mathcal{T}_i} h_{i-1,x} h_{i-1,x}^\top, \quad \mathcal{H}_{i-1}\Big|_{\beta_i} = \mathbf{1}\mathbf{1}^\top, \quad \mathcal{S}_i = \frac{1}{|\mathcal{T}_i|} \sum_{x \in \mathcal{T}_i} s_{i,x} s_{i,x}^\top$$

*where $h_{i-1}, s_i \in \mathbb{R}^{C_i \times |\mathcal{T}_i|}$ represent the pre-normalized activations and gradients, respectively. Here, $\mathcal{T}_i$ is the set of dimensions over which normalization statistics are computed, and $C_i$ is the channels/features size.*

*Proof.* Let $(\nu_i, \beta_i) \in \mathbb{R}^{C_i}$ be the scale and shift parameters of a normalization layer $i$, with transformation

$$h_i = a_i = \nu_i \odot h_{i-1} + \beta_i$$

where $h_{i-1} \in \mathbb{R}^{C_i \times |\mathcal{T}_i|}$ contains normalized activations and $\odot$ denotes element-wise multiplication. Let $\nabla_{\nu_i} J(\theta) = \sum_x h_{i-1,x} \odot s_{i,x}$ and $\nabla_{\beta_i} J(\theta) = \sum_x s_{i,x}$ where $s_i = \nabla_{h_i} J(\theta)$.

**For $\nu_i$ parameters:**

$$\mathbb{E}[\nabla_{\nu_i}\mathcal{L}\nabla_{\nu_i}\mathcal{L}^\top] = \mathbb{E}\left[\left(\sum_x h_{i-1,x} \odot s_{i,x}\right)\left(\sum_{x'} h_{i-1,x'} \odot s_{i,x'}\right)^\top\right]$$

$$\approx \mathbb{E}\left[\sum_x (h_{i-1,x} h_{i-1,x}^\top) \otimes (s_{i,x} s_{i,x}^\top)\right] \quad \text{(K-FAC independence assumption)}$$

$$= \left(\frac{1}{|\mathcal{T}_i|} \sum_x h_{i-1,x} h_{i-1,x}^\top\right) \otimes \left(\frac{1}{|\mathcal{T}_i|} \sum_x s_{i,x} s_{i,x}^\top\right)$$

$$= \mathcal{H}_{i-1}\Big|_{\nu_i} \otimes \mathcal{S}_i$$

**For $\beta_i$ parameters:**

$$\mathbb{E}[\nabla_{\beta_i}\mathcal{L}\nabla_{\beta_i}\mathcal{L}^\top] = \mathbb{E}\left[\left(\sum_x s_{i,x}\right)\left(\sum_{x'} s_{i,x'}\right)^\top\right]$$

$$= \mathbf{1}\mathbf{1}^\top \otimes \left(\frac{1}{|\mathcal{T}_i|} \sum_x s_{i,x} s_{i,x}^\top\right) \quad \text{(Bias term factorization)}$$

$$= \mathcal{H}_{i-1}\Big|_{\beta_i} \otimes \mathcal{S}_i$$

Cross-terms between $\nu_i$ and $\beta_i$ are excluded under the diagonal block assumption. $\square$

**Proposition A.2** (Efficient FIM). *Let $\mathcal{H}_{i-1}$ and $\mathcal{S}_i$ represent the KFs for a given layer index $i$ within a neural network, where these factors exhibit semi-diagonal characteristics indicating energy concentration predominantly along the diagonal, as elaborated in Section 3.1. Define $g_i$ as the gradient obtained through backpropagation at layer $i$. Assume that $\mathcal{H}_{i-1}$ and $\mathcal{S}_i$ can be closely approximated by diagonal matrices, denoted by $\mathcal{H}_{D_{i-1}}$ and $\mathcal{S}_{D_i}$ respectively at layer $i$, such that $\mathcal{H}_{D_{i-1}} = Diag(\mathcal{H}_{i-1})$, $\mathcal{S}_{D_i} = Diag(\mathcal{S}_i)$ where $Diag(\mathcal{M})$ denote the diagonal approximation of a matrix $\mathcal{M}$, which retains only the main diagonal. Therefore, we define the Empirical FIM as*

$$\tilde{F}_{D_i} \triangleq \mathcal{H}'_{D_{i-1}} \otimes \mathcal{S}'_{D_i} + \lambda \mathbf{I}, \tag{7}$$

*where $\mathcal{M}'$ denotes the Min-Max normalization technique Patro & Sahu (2015) for $\mathcal{M} = \mathcal{H}_{D_{i-1}}$ or $\mathcal{S}_{D_i}$. The regularization parameter $\lambda$ set to $0.001$ serves as damping factors, in alignment with the principles of Tikhonov regularization, to enhance computational stability and improve the conditioning of the matrix. The foundational aspects of the K-FAC optimization approach are detailed in Martens & Grosse (2015). Then, the closed-form solution for the preconditioned gradient $\bar{g}^{(t)}$, derived from the diagonal approximation of the FIM, is given by: $\bar{g}^{(t)} = (\tilde{F}_D^{(t)})^{-1} g^{(t)}$.*

*Proof.* The justification of our approach comprises two principal components: the rationale for adopting a diagonal approximation of the KFs and the methodology for normalization and regularization of these factors.

**Part 1: Diagonalization of KFs**

The assumption of independent neuronal activity within layers is foundational to our approach. This assumption posits that the covariance matrices $\mathcal{H}$ and $\mathcal{S}$, encapsulating the second-order statistics of activations and sensitivities, respectively, are diagonal. This diagonal nature arises because independence among random variables implies a covariance of zero for any pair of distinct variables, thereby nullifying all off-diagonal elements of these covariance matrices.

Consider matrices $A$ and $B$, each being diagonal with elements $a_{ii}$ and $b_{jj}$, respectively. The Kronecker product $A \otimes B$, by definition, generates elements $a_{ii}b_{jj}$ at the corresponding $(i,j)$ positions. For diagonal $A$ and $B$, this product maintains non-zero values exclusively at diagonal positions where $i = j$, resulting in:

$$A \otimes B = \mathrm{diag}(a_{11}b_{11}, \ldots, a_{nn}b_{mm}),$$

yielding a purely diagonal matrix. Moreover, we have empirically demonstrated that the energy of the KFs is concentrated along the diagonal, as detailed in Sections 3.1 and A.1. These arguments support our initial premise.

**Part 2: Normalization and Regularization**

Let $\mathcal{M} \in \{\mathcal{H}_{D_i}, \mathcal{S}_{D_i}\}$ be a diagonal matrix with entries $m_k > 0$. The min-max normalized matrix $\mathcal{M}'$ satisfies

$$\mathcal{M}' = D^{-1}(\mathcal{M} - m_{\min}I)D^{-1}, \quad D = \mathrm{diag}(\sqrt{m_{\max} - m_{\min}})$$

where $m_{\min} = \min_k m_k$, $m_{\max} = \max_k m_k$. This affine transformation ensures that $0 \preccurlyeq \mathcal{M}' \preccurlyeq I$ where $\preccurlyeq$ denotes Loewner ordering. Combined with Tikhonov regularization, the modified FIM, $\tilde{F}_{D_i} = \mathcal{H}'_{D_{i-1}} \otimes \mathcal{S}'_{D_i} + \lambda\mathbf{I}$ admits eigenvalue bounds

$$\lambda \leq \lambda_k(\tilde{F}_{D_i}) \leq 1 + \lambda \quad \forall k$$

which guarantees numerical stability for inversion. This approach ensures that all elements are scaled uniformly, preserving their relative magnitudes and distances. Compared to other normalization methods, such as z-score normalization (Patro & Sahu, 2015), Min-Max normalization offers several advantages such as the normalization Stability, where for $\mathcal{M}' = (\mathcal{M} - m_{\min}I)/(m_{\max} - m_{\min})$ we have $\sigma(\mathcal{M}') \subseteq [0,1]$ where $\sigma(\cdot)$ denotes matrix spectrum. Moreover, the Kronecker product satisfies $\sigma(\mathcal{H}'_{D_{i-1}} \otimes \mathcal{S}'_{D_i}) \subseteq [0,1]$, thus $\lambda_{\min}(\tilde{F}_{D_i}) \geq \lambda > 0$, guaranteeing invertibility. And the relative error satisfies $\frac{\|\tilde{F}_{D_i}^{-1} - F_{D_i}^{-1}\|}{\|F_{D_i}^{-1}\|} \leq \mathcal{O}(\epsilon + \lambda)$ where $\epsilon$ measures diagonal approximation error. Therefore, the preconditioned gradient can be written has $\bar{g}^{(t)} = (\mathcal{H}'_{D_{i-1}} \otimes \mathcal{S}'_{D_i} + \lambda\mathbf{I})^{-1} g^{(t)} = (\tilde{F}_{D_i}^{(t)})^{-1} g^{(t)}$. $\qquad\square$

**Proposition A.3** (Convergence in convex optimization)**.** *For the FIM defined in Eq. (7), the updating scheme $\theta^{(t+1)} = \theta^{(t)} - \alpha(\tilde{F}_D^{(t)})^{-1}\nabla J(\theta^{(t)})$ converges. Moreover, if $\nabla J$ is Lipschitz, i.e., $\|\nabla J(\theta) - \nabla J(\theta')\|_2 \leq L\|\theta - \theta'\|$ for any $\theta$ and $\theta'$, then for the $k$-step iteration with a fixed step size $\alpha \leq 1/L$, then*

$$J(\theta^{(k)}) - J(\theta^*) \leq \frac{\|\theta^{(0)} - \theta^*\|_2^2}{2\alpha k},$$

*where $J(\theta^*)$ is the optimal value.*

*Proof.* We follow the same proof as in Yao et al. (2021). Assume that $J(\theta)$ is a strongly convex and strictly smooth function in $\mathbb{R}^d$, such that there exist positive constants $\alpha$ and $\beta$ so that $\alpha I \preceq \nabla^2 J(\theta) \preceq \beta I$ for all $w$. We can show that the update formulation $\triangle\theta^{(t)} = (\tilde{F}^{(t)})^{-1}g^{(t)}$ converges by showing that with the proper learning rate:

$$\triangle\theta^{(t)} := J(\theta^{(t+1)}) - J(\theta^{(t)}) \leq -\frac{\alpha}{2\beta^2}||g^{(t)}||^2$$

Note that when $k = 0$ or 1, the convergence rate is the same as gradient descent or Newton method, respectively. Our proof is similar to Boyd & Vandenberghe (2004) for the Newton method. We denote $\lambda(\theta^{(t)}) = (g^{(t)})^\top (\tilde{F}^{(t)})^{-1}g^{(t)})^{1/2}$. Since $J(\theta)$ is strongly convex, we have

$$J(\theta^{(t)} - \eta\triangle\theta^{(t)}) \leq J(\theta^{(t)}) - \eta(g^{(t)})^\top\triangle\theta^{(t)} + \frac{\eta^2\beta||\triangle\theta^{(t)}||^2}{2}$$

$$\leq J(\theta^{(t)}) - \eta\lambda(\theta^{(t)})^2 + \frac{\beta}{2\alpha}\eta^2\lambda(\theta^{(t)})^2.$$

The second inequality comes from the fact that

$$\lambda(\theta^{(t)})^2 = \triangle(\theta^{(t)})^\top \tilde{F}^{(t)}\triangle\theta^{(t)} \geq \alpha||\triangle\theta^{(t)}||^2.$$

Therefore, the step size $\hat{\eta} = \alpha/\beta$ will make $f$ decrease as follows,

$$J(\theta^{(t)} - \hat{\eta}\triangle\theta^{(t)}) - J(\theta^{(t)}) \leq -\frac{1}{2}\hat{\eta}\lambda(\theta^{(t)})^2.$$

Since $\alpha I \preceq \tilde{F}^{(t)} \preceq \beta I$, we have

$$\lambda(\theta^{(t)})^2 = (g^{(t)})^\top(\tilde{F}^{(t)})^{-1}g^{(t)} \geq \frac{1}{\beta}||g^{(t)}||^2.$$

Therefore,

$$J(\theta^{(t)} - \hat{\eta}\triangle\theta^{(t)}) - J(\theta^{(t)}) \leq -\frac{1}{2\beta}\hat{\eta}||g^{(t)}||^2 = -\frac{\alpha}{2\beta^2}||g^{(t)}||^2 \tag{8}$$

Since $F_D^{(t)}$ is positive definite, hence Eq. (8) holds true. For the bound on convergence rate, we refer to Ryu & Boyd (2016) for the details of the complete proof. □

**Proposition A.4** (Convergence in non-convex stochastic optimization). *Under the assumptions:*
*(i) $f$ is lower bounded and differentiable; $||\nabla J(\theta) - \nabla J(\theta')||_2 \leq L||\theta - \theta'||_2$, $||\tilde{F}_D^{(t)}||_\infty < L$, $\forall t, \theta, \theta'$.*
*(ii) Both the true and stochastic gradient are bounded, i.e. $||\nabla J(\theta^{(t)})||_2 \leq \lambda$ and $||g_t||_2 \leq \lambda$, $\forall t$ for some $\lambda > 0$.*
*(iii) Unbiased and independent noise in $g^{(t)}$, i.e. $g^{(t)} = \nabla J(\theta^{(t)}) + \zeta^{(t)}$, $\mathbb{E}[\zeta^{(t)}] = 0$, and $\zeta^{(t)} \perp \zeta^{(t)}$, $\forall i \neq j$.*

*Assume $\eta^{(t)} = \frac{\eta}{\sqrt{t}}$, $\beta^{(t)} \leq \beta \leq 1$ is non-increasing, $\frac{\tilde{F}_D^{(t-1)}[j]}{\eta^{(t-1)}} \leq \frac{\tilde{F}_D^{(t)}[j]}{\eta^{(t)}}$, $\forall t \in [T], j \in [d]$, we then have*

$$\min_{t\in[T]} \mathbb{E}[||\nabla J(\theta^{(t)})||_2^2] \leq \frac{L}{\sqrt{T}}(C_1\eta^2\lambda^2(1 + \log T) + C_2 d\eta + C_3 d\eta^2 + C_4) \tag{9}$$

*where $C_1, C_2, C_3$ are constants independent of $d$ and $T$, $C_4$ is a constant independent of $T$, the expectation is taken w.r.t all the randomness corresponding to $\{g^{(t)}\}$.*

*Proof.* Follow Chen et al. (2019), as AdaFisher is an Adam-type method with the condition $||\eta^{(t)}m^{(t)}/\tilde{F}_D^{(t)}||_2 \leq G$ for some $G$ (which can be obtained by $\eta^{(t)} < \eta$, $||g^{(t)}||_2 \leq \lambda$ and $||\tilde{F}_D^{(t)}||_2 \geq 1$), we have

$$\mathbb{E}\left[\sum_{t=1}^T \eta^{(t)}\langle\nabla J(\theta^{(t)}), \nabla J(\theta^{(t)})/\tilde{F}_D^{(t)}\rangle\right] \leq \mathbb{E}\left[C_1\sum_{t=1}^T\left\|\frac{\eta^{(t)}g^{(t)}}{\tilde{F}_D^{(t)}}\right\|_2^2 + C_2\sum_{t=1}^T\left\|\frac{\eta^{(t)}}{\tilde{F}_D^{(t)}} - \frac{\eta^{(t-1)}}{\tilde{F}_D^{(t-1)}}\right\|_1\right.$$

$$\left. + C_3\sum_{t=1}^T\left\|\frac{\eta^{(t)}}{\tilde{F}_D^{(t)}} - \frac{\eta^{(t)}}{\tilde{F}_D^{(t)}}\right\|_2^2\right] + C_4. \tag{10}$$

We first bound non-constant terms in RHS of Eq. (10). For the term with $C_1$, since $||\tilde{F}_D^{(t)}||_2 \geq 1$, we have

$$\mathbb{E}\left[\sum_{t=1}^{T}\left\|\frac{\eta^{(t)}g^{(t)}}{\tilde{F}_D^{(t)}}\right\|_2^2\right] \leq \mathbb{E}\left[\sum_{t=1}^{T}||\eta^{(t)}g^{(t)}||_2^2\right]$$

$$= \mathbb{E}\left[\sum_{t=1}^{T}\left\|\frac{\eta}{\sqrt{t}}g^{(t)}\right\|_2^2\right]$$

$$\leq \eta^2\lambda^2\sum_{t=1}^{T}\frac{1}{t} \leq \eta^2\lambda^2(1+\log T).$$

For the term with $C_2$, we have

$$\mathbb{E}\left[\sum_{t=1}^{T}\left\|\frac{\eta^{(t)}}{\tilde{F}_D^{(t)}} - \frac{\eta^{(t-1)}}{\tilde{F}_D^{(t-1)}}\right\|_1\right] = \mathbb{E}\left[\sum_{j=1}^{d}\sum_{t=2}^{T}\left(\frac{\eta^{(t-1)}}{\tilde{F}_D^{(t-1)}[j]} - \frac{\eta^{(t)}}{\tilde{F}_D^{(t)}[j]}\right)\right]$$

$$= \mathbb{E}\left[\sum_{j=1}^{d}\frac{\eta^{(1)}}{\tilde{F}_D^{(1)}[j]} - \frac{\eta^{(T)}}{\tilde{F}_D^{(T)}[j]}\right]$$

$$\leq \mathbb{E}\left[\sum_{j=1}^{d}\frac{\eta^{(1)}}{\tilde{F}_D^{(1)}[j]}\right] \leq d\eta$$

where the first equality is due to $\frac{\tilde{F}_D^{(t-1)}[j]}{\eta^{(t-1)}} \leq \frac{\tilde{F}_D^{(t)}[j]}{\eta^{(t)}}, \forall t \in [T], j \in [d]$.

For the term with $C_3$, we have

$$\mathbb{E}\left[\sum_{t=1}^{T}\left\|\frac{\eta^{(t)}}{\tilde{F}_D^{(t)}} - \frac{\eta^{(t-1)}}{\tilde{F}_D^{(t-1)}}\right\|_2^2\right] = \mathbb{E}\left[\sum_{t=1}^{T}\sum_{j=1}^{d}\left(\frac{\eta^{(t)}}{\tilde{F}_D^{(t)}[j]} - \frac{\eta^{(t-1)}}{\tilde{F}_D^{(t)}[j]}\right)^2\right]$$

$$= \mathbb{E}\left[\sum_{t=1}^{T}\sum_{j=1}^{d}\left|\frac{\eta^{(t)}}{\tilde{F}_D^{(t)}[j]} - \frac{\eta^{(t-1)}}{\tilde{F}_{D^{(t-1)}}[j]}\right| \cdot \left|\frac{\eta^{(t)}}{\tilde{F}_D^{(t)}[j]} - \frac{\eta^{(t-1)}}{\tilde{F}_D^{(t-1)}[j]}\right|\right]$$

$$\leq \mathbb{E}\left[\sum_{t=1}^{T}\sum_{j=1}^{d}\left|\frac{\eta^{(t)}}{\tilde{F}_D^{(t)}[j]} - \frac{\eta^{(t-1)}}{\tilde{F}_D^{(t-1)}[j]}\right| \cdot \left|\frac{\eta}{\sqrt{t}\tilde{F}_D^{(t)}[j]} - \frac{\eta}{\sqrt{t-1}\tilde{F}_D^{(t-1)}[j]}\right|\right]$$

$$\leq \mathbb{E}\left[\eta\sum_{t=1}^{T}\sum_{j=1}^{d}\left|\frac{\eta_t}{\tilde{F}_D^{(t)}[j]} - \frac{\eta^{(t-1)}}{\tilde{F}_D^{(t-1)}[j]}\right|\right]$$

$$= \eta\mathbb{E}\left[\sum_{t=1}^{T}\left\|\frac{\eta^{(t)}}{\tilde{F}_D^{(t)}} - \frac{\eta^{(t-1)}}{\tilde{F}_D^{(t-1)}}\right\|_1\right]$$

$$\leq d\eta^2$$

Hence

$$\mathbb{E}\left[C_1\sum_{t=1}^{T}\left\|\frac{\eta^{(t)}g^{(t)}}{\tilde{F}_D^{(t)}}\right\|_2^2 + C_2\sum_{t=1}^{T}\left\|\frac{\eta^{(t)}}{\tilde{F}_D^{(t)}} - \frac{\eta^{(t-1)}}{\tilde{F}_D^{(t-1)}}\right\|_1 + C_3\sum_{t=1}^{T}\left\|\frac{\eta^{(t)}}{\tilde{F}_D^{(t)}} - \frac{\eta^{(t-1)}}{\tilde{F}_{D^{(t-1)}}}\right\|_2^2\right] + C_4$$

$$\leq C_1\eta^2\lambda^2(1+\log T) + C_2d\eta + C_3d\eta^2 + C_4 \qquad (11)$$

Now we lower bound the LHS of Eq. (9). With the assumption $||\tilde{F}_D^{(t)}||_\infty \leq L$, we have

$$(\eta^{(t)}/\tilde{F}_D^{(t)})_j \geq \frac{\eta}{L\sqrt{t}}.$$

Thus

$$\mathbb{E}\left[\sum_{t=1}^{T}\eta^{(t)}\langle\nabla J(\theta^{(t)}), \nabla J(\theta^{(t)})/\tilde{F}_D^{(t)}\rangle\right] \geq \mathbb{E}\left[\sum_{t=1}^{T}\frac{\eta}{L\sqrt{t}}||\nabla J(\theta^{(t)})||_2^2\right] \geq \frac{\sqrt{T}}{L}\min_{t\in[T]}\mathbb{E}[||\nabla J(\theta^{(t)})||_2^2]$$

$$(12)$$

Combining Eq. (11) and (12) gives the desired result. $\qquad\square$

### A.3 COMPUTATION OF KFs

The KFs $\mathcal{H}$ and $\mathcal{S}$, which are integral to the AdaFisher optimizer, are computed following methodologies similar to those described in Grosse & Martens (2016); Martens & Grosse (2015). This section revisits the key equations used for this computation. For a given layer $i$ in a neural network, the empirical KFs are computed as follows:

- For **fully connected layers**, the KFs are:

$$\mathcal{H}_{D_{i-1}} = \mathrm{diag}(\bar{h}_{i-1}\bar{h}_{i-1}^\top), \quad \mathcal{S}_{D_i} = \mathrm{diag}(s_i s_i^\top);$$

- For **convolutional layers**, the computation accounts for the spatial positions within the layer, denoted as $\mathcal{T}$:

$$\mathcal{H}_{D_{i-1}} = \mathrm{diag}\left(\frac{[\![\bar{h}_{i-1}]\!][\![\bar{h}_{i-1}]\!]^\top}{|\mathcal{T}|}\right), \quad \mathcal{S}_{D_i} = \mathrm{diag}\left(\frac{s_i s_i^\top}{|\mathcal{T}|}\right);$$

The algorithm employs the expansion operation denoted by $[\![\cdot]\!]$ (Grosse & Martens, 2016). *This operation essentially takes the patches surrounding spatial locations, stretches them into vectors, and compiles these vectors into a matrix.*

- For **Normalization layers** (BatchNorm & LayerNorm) refer to Proposition. 3.1
- For all **other type of layers** the KFs are:

$$\mathcal{H}_{D_{i-1}} = \mathbf{I}_{P_{i-1}^{out}+1}, \quad \mathcal{S}_{D_i} = \mathbf{I}_{P_i^{out}};$$

Table 6: AdaFisher training time per epoch (s) across various numbers of GPUs on ResNet-50 ImageNet-1k.

| GPU amount | Batch Size | AdaFisher training time per epoch (s) |
|---|---|---|
| 1 | 256 | 2882 |
| 2 | 512 | 1438 |
| 3 | 768 | 963 |
| 4 | 1024 | 720 |

### A.4 DISTRIBUTED ADAFISHER

The efficacy of AdaFisher hinges on its innovative approximation of the FIM, denoted as $\tilde{F}$, which leverages KFs for computation. In a distributed setting, it is crucial to aggregate these KFs across multiple GPUs before updating the model parameters. Consider a training environment consisting of $K$ GPUs. For any given layer $i$, the KFs are computed and aggregated across all GPUs as

$$\mathcal{H}_{D_{i-1}}^{(\mathrm{SUM})} = \frac{1}{K}\sum_{k=1}^{K}\mathcal{H}_{D_{i-1}}^{(k)}, \quad \mathcal{S}_{D_i}^{(\mathrm{SUM})} = \frac{1}{K}\sum_{n=1}^{K}\mathcal{S}_{D_i}^{(k)} \tag{13}$$

The theoretical justification for this aggregation lies in the linearity of expectation and the unbiasedness of the local KF estimates. Specifically, if each $\mathcal{H}_{D_{i-1}}^{(k)}$ and $\mathcal{S}_{D_i}^{(k)}$ are unbiased estimators of their respective true factors $\mathcal{H}_{D_{i-1}}$ and $\mathcal{S}_{D_i}$ for $k = 1, \ldots, K$, then the averaged factors $\mathcal{H}_{D_{i-1}}^{(\mathrm{SUM})}$ and $\mathcal{S}_{D_i}^{(\mathrm{SUM})}$ remain unbiased estimators of $\mathcal{H}_{D_{i-1}}$ and $\mathcal{S}_{D_i}$. Consequently, using Eq. (13), the aggregated EFIM for layer $i$ can be calculated as

$$\tilde{F}_{D_i}^{\mathrm{SUM}} = \mathcal{H}_{D_{i-1}}^{\prime(\mathrm{SUM})} \otimes \mathcal{S}_{D_i}^{\prime(\mathrm{SUM})} + \lambda\mathbf{I}$$

where $\lambda$ is a regularization parameter added to ensure numerical stability. This methodology ensures that each GPU contributes to a comprehensive update of the model, enhancing both convergence and performance in large-scale distributed training environments. We assessed the distributed version of AdaFisher on ImageNet-1k, utilizing batch sizes of 512 and 1024 (refer to Table 3 and Figure 15 for details). Our findings indicate that AdaFisher scales nearly linearly with the number of GPUs, as evidenced in Table 6. There remains scope for additional low-level optimizations within the implementation to further enhance performance.

# B ABLATION STUDIES

Building on the ablative studies detailed in Section 4.4, this section extends our stability analysis to explore the impact of various learning rate schedulers and convergence efficiency, as discussed in Section B.1. Additionally, we conduct an in-depth examination of the key components of AdaFisher. This includes analyzing the effects of the EMA, the use of square root transformations, our novel approximation of the FIM, and the critical role of computing the FIM for normalization layers, all of which are detailed in Section B.2. We have consolidated the key findings of each ablation study in Table 7.

Table 7: Summary of Ablation Studies for AdaFisher Optimizer.

| Ablation Study | Component Studied | Key Findings |
|---|---|---|
| Learning rate schedulers | Impact of Cosine Annealing, StepLR, and no scheduler on AdaFisher | AdaFisher maintains stable and efficient performance across various schedulers, demonstrating its robustness and adaptability in diverse training environments. Further details are given in Section B.1. |
| Convergence Efficiency | Performance and alignment of FIM with Hessian | AdaFisher shows marked performance improvements towards the end of training, with FIM alignment to the Hessian enhancing rapid convergence and stable generalization across training and testing phases. Further details are provided in Section B.1. |
| Square Root Utilization | Effect of omitting square root in update rules | Eliminating the square root enhances AdaFisher's performance and stability, outperforming both its own version with the square root and Adam without the square root, while also improving computational efficiency. Further details are listed in Section B.2. |
| EMA of KFs | Utilization of EMA for curvature estimation | Using EMA on KFs enhances AdaFisher's curvature estimation, leveraging data from multiple mini-batches for continuous updates, demonstrating significant benefits in methods with diagonal or block-diagonal curvature approximations. Further analysis are included in Section B.2. |
| Importance of Fisher Computation for Normalization Layers | Impact of EFIM in normalization layers | Incorporating Fisher computation in normalization layers significantly improves AdaFisher's generalization and stability by enhancing parameter sensitivity and gradient variability insights, crucial for optimizing training dynamics and model convergence. Further details are given in Section B.2. |
| New Approximation of the FIM | Diagonal approximation of the FIM | Our novel method focuses on the diagonal elements of the FIM, enhancing computation efficiency without losing critical information. Validation shows our approximation closely aligns with the true Fisher, confirming its efficacy. Further details are contained in Section B.2. |

## B.1 EVALUATING STABILITY ACROSS LEARNING RATE SCHEDULERS, AND ASSESSING CONVERGENCE EFFICIENCY

**Learning rate schedulers.** This analysis evaluates the impact of different learning rate schedulers–Cosine Annealing, StepLR, and no scheduler—on the performance of AdaFisher, as depicted in Figure 10. AdaFisher exhibits remarkable robustness across these scheduling strategies. Notably, its performance remains stable and efficient, whether it is paired with the gradual adjustments of Cosine Annealing, the abrupt changes of StepLR, or even in the absence of any scheduler. This underscores AdaFisher's adaptability and effectiveness in diverse training environments.

**Convergence Efficiency.** As training progresses, AdaFisher optimizer demonstrates a significant enhancement in performance compared to its counterparts, especially evident towards the end of the training period (see Appendix D.2.4). This rapid convergence is attributed to AdaFisher's approach by incorporating the FIM. Early and mid-training, the FIM serves as an approximation to

the Hessian matrix, equivalent to the Generalized Gauss Newton Matrix (Eschenhagen et al., 2024). However, as the model approaches a local minimum, the FIM increasingly aligns precisely with the Hessian (Martens, 2020). This precise alignment accelerates convergence, markedly improving the optimizer's efficiency in the final phases of training. Additionally, AdaFisher's tendency to converge to flat local minima leads to more stable generalization when transitioning from training to testing distributions (Cha et al., 2021), contrasting sharply with other optimizers. To support these points, we analyze the training distribution of our diagonal block-Kronecker FIM during the training of ResNet18 on CIFAR10. Specifically, we examine the FIM distribution for the first (Panel A), middle (Panel B) convolutional layers and the last linear layer (Panel C), as shown in Figure 11. It is evident that for each layer, the FIM distribution with AdaFisher narrows to smaller values with fewer variations compared to that with Adam. This pattern demonstrates AdaFisher's convergence toward flatter local minima, as the Fisher Information, an approximation of the Hessian, contains crucial curvature information.

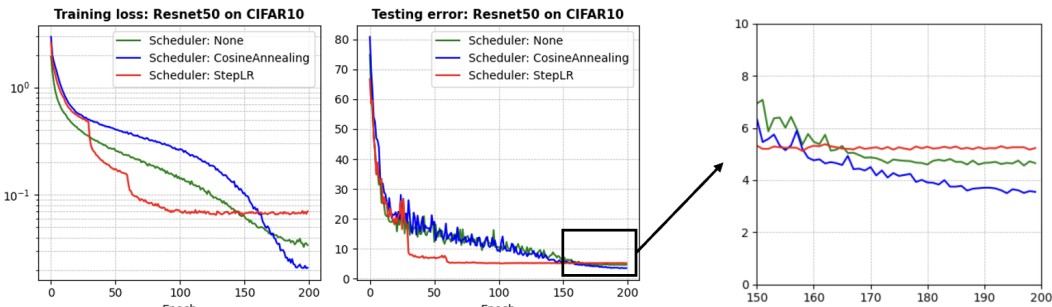

Figure 10: Performance comparison of AdaFisher using the ResNet50 on the CIFAR10 with a batch size of 256 with different learning rate schedulers.

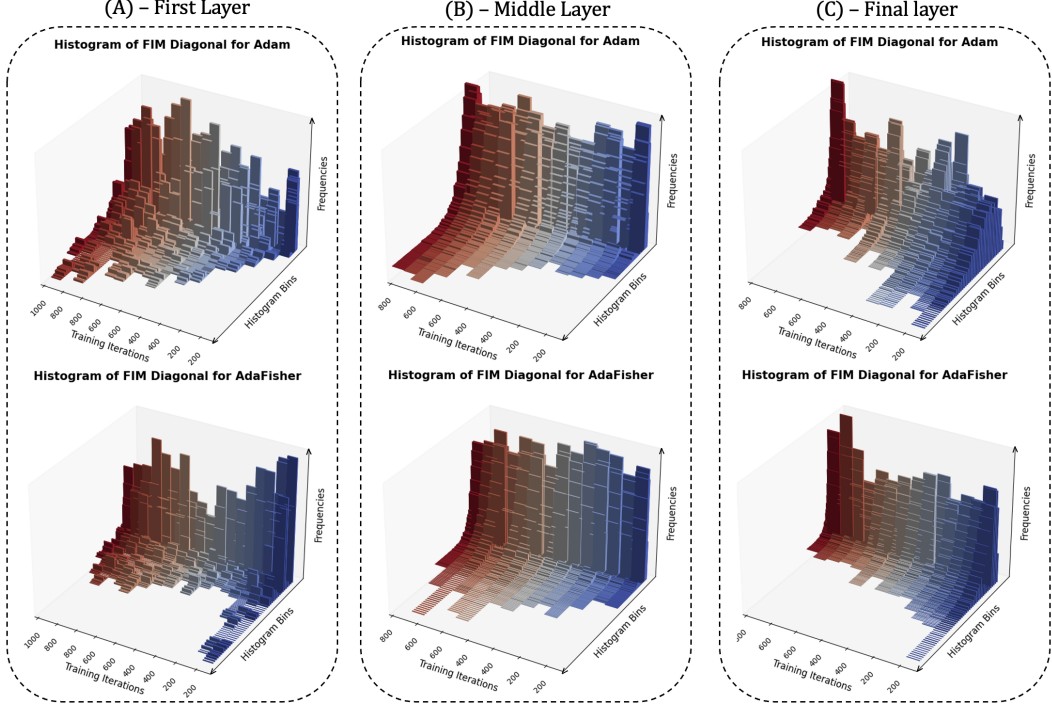

Figure 11: Comparison of FIM Diagonal Histograms during ResNet18 Training on CIFAR10 with Adam and AdaFisher over 1,000 training iterations. Panel (A) displays the FIM diagonal elements for the first convolutional layer; Panel (B) illustrates the FIM diagonal elements for the middle convolutional layer; Panel (C) shows the FIM diagonal elements for the last Linear layer.

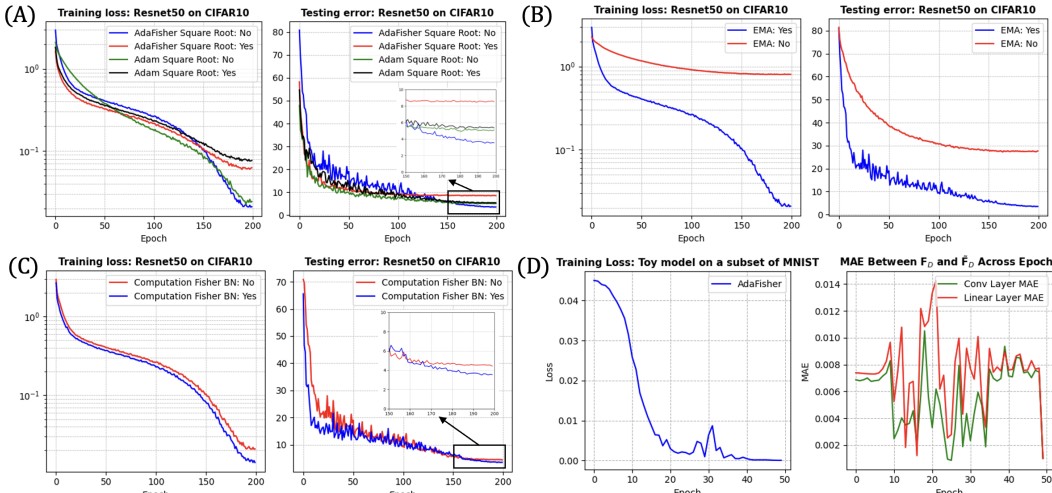

Figure 12: AdaFisher Component Analysis. (A) Comparison of MAE between the true FIM $F_D$ and our approximation $\tilde{F}_D$ across convolutional and dense layers. (B) Performance comparison of AdaFisher with and without the EMA of KFs. (C) Assessment of AdaFisher's performance with and without the computation of EFIM for Batch Normalization (BN) layers.

### B.2 COMPONENT ANALYSIS: EVALUATING THE SIGNIFICANCE OF ADAFISHER'S ELEMENTS

AdaFisher incorporates several key components, including a novel approximation of the FIM, the EMA of the KFs, the omission of the square root in the update rule, and a new EFIM formula for normalization layers. In this part, we elucidate each component and its significance within the AdaFisher optimizer.

**Square Root Utilization.** Recent studies, such as (Lin et al., 2024), have reevaluated the necessity of the square root operation in the Adam family's update rules. These studies suggest that eliminating the square root does not affect convergence and may even narrow the generalization gap compared to SGD in CNN models. Our analysis, shown in panel (A) of Figure 12, investigates this aspect by comparing the performance of AdaFisher and Adam, both with and without the square root operation. The findings reveal that removing the square root not only boosts the performance and stability of both optimizers but also significantly enhances computational efficiency. Specifically, AdaFisher without the square root not only outperforms the version with the square root but also surpasses Adam without the square root. However, Adam without the square root typically requires an additional **scaling factor** proportional to the batch size, denoted as $f \propto$ batch size, to function correctly. Without this factor, Adam, without the square root, fails to learn effectively, making direct comparisons with AdaFisher invalid.

**EMA of KFs.** As elucidated in Section 3.2, employing an EMA over the KFs facilitates a more sophisticated curvature estimation. This technique leverages data across multiple mini-batches, enabling continuous updates to the Fisher information rather than relying solely on the data from a single batch. Panel (B) of Figure 12 underscores, using ResNet-50 on CIFAR10 over 200 epochs, the benefits of using EMA on KFs, a strategy particularly advantageous in methods that utilize diagonal or block-diagonal approximations of the curvature matrix.

**Importance of Fisher Computation for Normalization Layers.** The integration of the EFIM in normalization layers, as detailed in Proposition 3.1, significantly enhances the generalization process. Panel (C) of Figure 12 illustrates the impact of incorporating Fisher computation in these layers during the training of AdaFisher with ResNet-50 on CIFAR10 over 200 epochs. In contrast, the identity matrix is employed when Fisher's computation is omitted. The superior performance of AdaFisher when incorporating Fisher computation can be attributed to the critical role normalization layers play in adjusting the input distribution for each mini-batch. This adjustment substantially enhances the neural network's learning stability (Jiang et al., 2024b). By quantifying the information each output $y$ carries about the parameters $\theta$ under the model distribution $p(y|x;\theta)$, the computa-

tion of the FIM in these layers provides valuable insights into parameter sensitivity and gradient variability. This insight is crucial for optimizing training dynamics and enhancing model convergence—areas that are often inadequately addressed by existing optimizers.

**New Approximation of the FIM.** In Proposition 3.2, we introduce a new methodology for approximating the FIM that diverges from the K-FAC optimizer. Unlike K-FAC, which utilizes the full Kronecker product, our approach focuses solely on the diagonal elements of the FIM, where, as demonstrated in Section 3.1, the energy of the KFs is predominantly concentrated. This method enables a more efficient computation of the FIM without sacrificing critical information. To validate our approach, we compare the true FIM diagonal with our approximation in convolutional and dense layers using a toy model composed of 2 convolutional layers and two linear layers on a subset of the MNIST dataset (Deng, 2012) over 50 epochs. Specifically, we calculate the true Fisher using the NNgeometry Python package (George, 2021), which facilitates the computation of the FIM, Gauss-Newton Matrix, or Neural Tangent Kernels applied to neural networks. We estimate $p(y|x)$ through Monte-Carlo sampling. During each epoch, we collected both the empirical and true Fisher information and calculated the Mean Absolute Error (MAE) between these two measures. Panel (D) of Figure 12 showcases the close approximation of AdaFisher's empirical diagonal to the true Fisher, thus validating the efficacy of our approximation method.

## C  VISUALIZATION

The convergence rate of an optimizer is crucial, serving as an indicator of its robustness against saddle points and its ability to generalize effectively. In this section, we introduce a novel methodology for visualizing the convergence behavior of optimizers through a statistical model, as depicted in Figure 1. Initially, our process employs Principal Component Analysis (PCA) for dimensionality reduction, reducing the dataset dimensions from $\mathcal{D} \in \mathbb{R}^{m \times n}$ to $\hat{\mathcal{D}} \in \mathbb{R}^{m \times 2}$, following the protocol established in F.R.S. (1901). We then apply this reduced dataset to a toy model composed of an $L$-layer multi-layer perceptron (MLP). Notably, we focus on the first weight matrix $W_1^e$ of this MLP, which resides in $\mathbb{R}^2$, where $e$ denotes the epoch number. For consistency and to ensure comparability, all layers' weights are initialized identically across different optimizers. Following the training

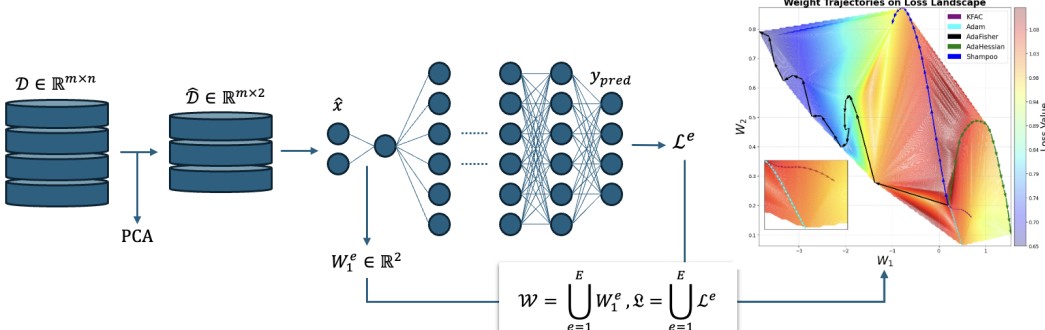

Figure 13: Pipeline for visualization of optimization paths for various algorithms on a loss surface, comparing their convergence efficiency.

phase with various optimizers where we denote a set of optimizer results $\mathcal{O}$, we analyze both the collection of first-layer weights, $\mathcal{W}$, and the evolution of the loss function, $\mathfrak{L}$ defined as:

$$\mathcal{W} = \begin{bmatrix} (W_1^1)^\top \\ (W_1^2)^\top \\ \vdots \\ (W_1^E)^\top \end{bmatrix}, \quad \mathfrak{L} = [\mathcal{L}_1^1, \mathcal{L}_1^2, \dots, \mathcal{L}_1^E]^\top$$

where $(W_1^e)^\top$ represents the weight vector at the $e$th epoch, and $\mathcal{L}_1^e$ represents the loss at the $e$th epoch, extracted from the optimization results $\mathcal{O}$. We construct a grid $(\mathbf{X}, \mathbf{Y})$ spanning the range of weight parameters, discretized into 200 linearly spaced points along each axis:

$$\mathbf{X}, \mathbf{Y} = \text{meshgrid}\left(\min(\mathcal{W}_{:,1}), \max(\mathcal{W}_{:,1}), \min(\mathcal{W}_{:,2}), \max(\mathcal{W}_{:,2}), 200\right)$$

Finally, we interpolate the loss values $\mathcal{L}$ over the grid using cubic interpolation to obtain a smooth loss surface $\mathbf{Z}$:

$$\mathbf{Z} = \text{griddata}(\mathcal{W}, \mathcal{L}, (\mathbf{X}, \mathbf{Y}), \text{method} = \texttt{cubic})$$

These elements are integral to the visualization process, which elucidates the optimizer's trajectory through the parameter space across training epochs. It is important to note that while we focus on the first layer's weight matrix for clarity, the methodology can be adapted to visualize the weights of any layer within the network. Figure 13 summarizes the pipeline.

In the experiment depicted in Figure 1, we selected the IRIS dataset (rz7, 2018), owing to its widespread recognition and compatibility with PCA application. Our model employs a 2-layer MLP architecture. We specifically attend to the weight matrix of the first layer, denoted by $W_1 \in \mathbb{R}^2$. This particular focus is informed by the empirical observation that the parameters of the first layer tend to exhibit a faster convergence rate compared to those of subsequent layers in the network. Such a phenomenon can be attributed to the more direct influence of the input features on the first layer's weights, which often results in a more pronounced and expedited learning dynamic. Given the classification nature of the task, we employed the Cross-Entropy loss function (Zhang & Sabuncu, 2018). The network was trained over 20 epochs using a suite of optimizers: Adam, AdaHessian, K-FAC, Shampoo, and AdaFisher. We standardized the learning rate across all optimizers at $1 \times 10^{-3}$ to ensure comparability of results. Examination of Figure 1 reveals that AdaFisher's convergence is markedly superior to that of its counterparts, achieving rapid convergence to the local minimum of the loss landscape concerning the first weight parameter within a few iterations. Conversely, the alternative optimizers demonstrate convergence to less optimal local minima. Note that while the results may vary due to the stochastic nature of parameter initialization, AdaFisher typically converges to a better local minimum compared to its counterparts.

# D EXPERIMENTS

## D.1 HARDWARE

In total, we had a server with 6 NVIDIA RTX 6000 Ada Generation GPUS with 48 gigabytes of VRAM and 128 gigabytes of RAM available for all experiments. All experiments described in this report were conducted on a system equipped with a single NVIDIA RTX 6000 Ada Generation GPU and 64 gigabytes of RAM, except for training AdaFisher on ImageNet-1k with batch sizes of 512 and 1024, where four GPUs were utilized.

## D.2 IMAGE CLASSIFICATION

We provide further results and detailed descriptions of our image classification experiments in this section. We conducted five trials with random initializations for the CIFAR experiments and one trial each for Tiny ImageNet and ImageNet-1k. We present the mean and standard deviation of the results for these trials.

**Note on training time.** Given that various optimizers demonstrate significantly different epoch durations, we have standardized our comparisons by restricting training to the total WCT consumed by 200 epochs using AdaFisher for both CIFAR and Tiny ImageNet experiments. Conversely, for ImageNet-1k, we report the results based on 90 WCT training epochs using Adam, as, surprisingly, AdaFisher and Adam exhibited the same duration in this experiment. The final selected number of epochs for each optimizer is detailed in Table 8. Note that we were unable to train AdaHessian on ImageNet-1k due to the significant computational resources required by this optimizer.

Table 8: Comparison of the final epoch counts for various optimizers across different datasets.

| | | | CIFAR10/100 & Tiny ImageNet | | | | ImageNet-1k | | |
|---|---|---|---|---|---|---|---|---|---|
| Optimizers | SGD | Adam/AdamW | AdaHessian | K-FAC | Shampoo | AdaFisher/AdaFisherW | Adam | K-FAC | Shampoo | AdaFisher |
| Epochs | 226 | 210 | 89 | 107 | 36 | 200 | 90 | 60 | 26 | 90 |

### D.2.1 HP TUNING

Effective HP tuning is crucial for optimizing the performance of deep learning models. In this study, we systematically explored various HPs for both CNNs and ViTs across multiple image clas-

sification tasks. The following subsections detail the tuning strategies employed for each model architecture and dataset.

**CNNs.** For all image classification tasks involving CNNs, we utilized ResNet18 as the backbone architecture and evaluated its performance on the CIFAR-10 dataset with a fixed batch size of 256 trained on 50 epochs. The HP tuning process encompassed the following components:

- **Optimizer Selection and Learning Rate Tuning:** Each optimizer was fine-tuned using ResNet18 on CIFAR-10. We performed a grid search to identify the optimal learning rate from the set $\{0.0001, 0.0003, 0.0005, 0.0009, \ldots, 0.1, 0.3, 0.5, 0.9\}$.

- **Learning Rate Scheduling:** A cosine annealing learning rate decay strategy was employed, aligning with the number of training epochs specified for each optimizer in Table 8. This approach follows the methodology proposed by Loshchilov & Hutter (2019a) and was determined to be optimal for our experimental setup.

- **Weight Decay:** We applied a uniform weight decay of $5 \times 10^{-4}$ across all optimizers for CIFAR-10 and Tiny ImageNet. An exception was made for MobileNetV3, where the weight decay was set to $1 \times 10^{-5}$. For experiments on ImageNet-1k, the weight decay was established at $1 \times 10^{-4}$.

- **Damping Parameter Tuning:**
  - **AdaFisher, K-FAC, and Shampoo:**
    * **K-FAC and AdaFisher:** The damping parameter was searched within $\{0.0001, 0.0003, 0.0005, 0.0009, 0.001, 0.003, 0.005, 0.009, 0.01, 0.03, 0.05, 0.09\}$. This range was chosen based on prior research (Martens & Grosse, 2015) and our own experiments, which indicated optimal damping values around $1 \times 10^{-3}$.
    * **Shampoo:** The damping parameter was tuned within $\{1 \times 10^{-6}, 3 \times 10^{-6}, 5 \times 10^{-6}, 9 \times 10^{-6}, 1 \times 10^{-5}, 3 \times 10^{-5}, 5 \times 10^{-5}, 9 \times 10^{-5}, 1 \times 10^{-4}, 3 \times 10^{-4}, 5 \times 10^{-4}, 9 \times 10^{-4}\}$, as optimal values typically reside around $1 \times 10^{-5}$.
  - **AdaHessian:** The Hessian power was tuned within the range $\{0.1, 0.2, \ldots, 0.9, 1.0\}$.
  - **SGD:** The momentum of SGD was tuned within the range $\{0.1, 0.2, \ldots, 0.9, 1.0\}$.
  - **AdaFisher Decay Factors:** The decay factor $\gamma$ for AdaFisher was tuned within $\{0.1, 0.2, \ldots, 0.9, 0.99\}$. The optimal value is: $\gamma = 0.8$.

- **Implementation Details:** For the Shampoo and K-FAC optimizers, we utilized the ASDL library as implemented in PyTorch provided by Osawa et al. (2023).

**ViTs.** For ViT-based image classification tasks, we employed the Tiny Swin Transformer on the CIFAR-10 dataset with a batch size of 256. The HP tuning strategy for ViTs included the following elements:

- **Weight Decay:** Weight decay values were set as indicated in the respective original publications for each model:
  - Tiny Swin: $1 \times 10^{-2}$
  - FocalNet: $5 \times 10^{-2}$
  - CCT-2/3×2: $6 \times 10^{-2}$

- **Learning Rate Tuning:** For SGD, AdaFisher, AdaHessian, K-FAC, and Shampoo optimizers, we conducted a grid search over the learning rates $\{0.3, 0.15, 0.1, 0.05, 0.03, 0.015, 0.01, 0.005, 0.003, 0.0015, 0.001\}$, as these optimizers typically operate with higher learning rates compared to Adam-based optimizers. For AdamW, the learning rates were adopted from the original publications:
  - Tiny Swin and FocalNet: $1 \times 10^{-4}$
  - CCT-2/3×2: $5.5 \times 10^{-5}$

- **Damping Parameter Tunning:** We performed the same grid search over the damping parameter for K-FAC, Shampoo and AdaFisher, the Hessian power for AdaHessian, the momentum for SGD, and the decay factors for AdaFisher as explained in the CNNs part.

This meticulous HP tuning process ensures that each optimizer is optimally configured for the respective model architectures and datasets, thereby facilitating a fair and comprehensive comparison

of their performance across different image classification tasks. The final learning rates for all optimizers and models are detailed in Table 9.

Table 9: Final selected learning rates for each optimizer, tuned using ResNet18 (for CNN) and Tiny Swin (for ViT) on CIFAR10 using a batch size of 256. We selected based on final validation top-1 accuracy.

| Architecture | SGD | Adam | AdamW | AdaHessian | K-FAC | Shampoo | AdaFisher | AdaFisherW |
|---|---|---|---|---|---|---|---|---|
| CNNs | 0.1 | 0.001 | - | 0.15 | 0.3 | 0.3 | 0.001 | - |
| ViTs | 0.01 | - | 0.0001/0.000055 | 0.01 | 0.003 | 0.003 | - | 0.001 |

### D.2.2 DATASET DETAILS

**CIFAR.** The training/test sets for Cifar10/100 dataset contain 50k/10k images, respectively. We consider a batch size of 256. For CIFAR-related experiments, we perform $32 \times 32$ random-resize cropping, random horizontal flipping and cutout (DeVries & Taylor, 2017) as data augmentations. Refer to Takahashi et al. (2020) for more details.

**Tiny ImageNet.** The training/test sets for Tiny ImageNet Le & Yang (2015) contains 100k/10k images. We perform $64 \times 64$ random-resize cropping and random horizontal flipping. The batch size is set to be 256.

**ImageNet-1k.** The training/test sets for ImageNet-1k Russakovsky et al. (2015) contains 1,281,167/150k images. We consider a batch size of 256, as we performed experiments on a single GPU instance without any GPU parallelism. We follow He et al. (2016) and perform random resized cropping to $224 \times 244$ and random horizontal flipping on the train set and $256 \times 256$ resizing with $224 \times 224$ center cropping on the test set.

Table 10: Final selected learning rates for each optimizer with ImageNet-1k pretrained weights, tuned using ResNet50 on CIFAR10 using a batch size of 256. We tuned by completing a full WCT epoch training cycle and selected based on final validation top-1 accuracy.

| SGD | Adam | AdaHessian | K-FAC | Shampoo | AdaFisher |
|---|---|---|---|---|---|
| 0.01 | 0.0001 | 0.15 | 0.3 | 0.03 | 0.001 |

Table 11: Final selected epoch counts for various optimizers across transfer learning task.

| SGD | Adam/AdamW | AdaHessian | K-FAC | Shampoo | AdaFisher/AdaFisherW |
|---|---|---|---|---|---|
| 58 | 55 | 22 | 27 | 18 | 50 |

### D.2.3 TRANSFER LEARNING

For transfer learning, weights are initialized to the values provided by the publicly available checkpoints by PyTorch, except the first convolutional for the ResNet architecture and last dense layers for all networks, which change size to accommodate the new kernel size and number of classes, respectively, that are randomly initialized. We train all models with weight decay $1e^{-4}$ as suggested in Wightman et al. (2021), except for MobileNetV3, where weight decay is set to be $1e^{-5}$. Moreover, we did a grid search for each optimizer for selecting the best learning rate of the range $\{0.3, 0.15, 0.1, 0.03, 0.015, 0.01, \ldots, 1e - 5\}$ where we tabulate the selected learning rate for each optimizer in Table 10. We use a batch size of 256 and cosine learning rate decay. We use the same augmentation policy (without Cutout) as in the previous experiments. The results were obtained using the WCT technique over 50 training epochs of AdaFisher, with the final epoch count detailed in Table 11. All other parameters remained unchanged.

Table 12: Performance of various networks and optimizers on Tiny ImageNet using batch size 256. Reported using wall clock time of 200 AdaFisher training epochs as the cutoff.

| Network | Adam | AdaHessian | K-FAC | Shampoo | AdaFisher |
|---|---|---|---|---|---|
| ResNet50 | 53.06 | 50.21 | 50.05 | 53.53 | **57.41** |
| Big Swin | 48.11 | — | 8.89 | 4.11 | **48.86** |

#### D.2.4 RESULTS

Table 12 displays the results for the Tiny ImageNet dataset using ResNet50 and Big Swin networks, with visualizations provided in Figure 14. AdaFisher and AdaFisherW consistently outperform current SOTA optimizers. Notably, Figure 14 illustrates that although AdaFisher converges slower than K-FAC during ResNet50 training, it achieves superior generalization. This is evidenced by lower testing errors, suggesting that AdaFisher tends to converge to a flatter local minimum, enabling smoother transitions between training and testing datasets with minimal generalization loss. For further explanation, see Cha et al. (2021). Note that due to AdaHessian's high memory consumption, we were unable to train it on Big Swin. Table 13 presents the performance of various networks on CIFAR10/100 datasets using different optimizers, both with and without the cutout augmentation technique. AdaFisher and AdaFisherW consistently outperform their counterparts in both scenarios, demonstrating stable training and robustness to the augmentation techniques. The training losses and test errors for the CIFAR experiments, both with and without cutout, are visually represented in Figures 16, 17, 18, and 19. Moreover, Figure 20 shows the training and validation error of transfer learning task. Finally, Figure 15 illustrates the training and validation error of the distributed version of AdaFisher on ImageNet-1k across various batch sizes. AdaFisher not only outperforms its counterparts with smaller batch sizes (256), but it also continues to achieve superior generalization as batch sizes increase. Furthermore, these results reinforce the stability analysis concerning batch sizes presented in Section 4.4, extending it to a more challenging dataset.

Table 13: Performance metrics (mean, std) of different networks and optimizers on CIFAR10 and CIFAR100 using batch size 256 (a) without Cutout and (b) with Cutout. Reported using WCT of 200 AdaFisher training epochs as the cutoff.

(a) Without Cutout

| Network | CIFAR10 | | | | | | CIFAR100 | | | | | |
|---|---|---|---|---|---|---|---|---|---|---|---|---|
| | SGD | Adam | AdaHessian | K-FAC | Shampoo | AdaFisher | SGD | Adam | AdaHessian | K-FAC | Shampoo | AdaFisher |
| ResNet18 | $94.89_{0.1}$ | $93.64_{0.1}$ | $94.05_{0.1}$ | $94.04_{0.2}$ | $94.52_{0.1}$ | $\mathbf{95.02_{0.1}}$ | $76.42_{0.1}$ | $72.71_{0.2}$ | $73.64_{0.2}$ | $74.79_{0.2}$ | $76.53_{0.1}$ | $\mathbf{77.10_{0.2}}$ |
| ResNet50 | $95.07_{0.2}$ | $93.89_{02}$ | $94.26_{0.1}$ | $94.25_{0.1}$ | $94.92_{0.1}$ | $\mathbf{95.42_{0.2}}$ | $77.50_{0.2}$ | $73.12_{0.7}$ | $75.29_{0.3}$ | $75.49_{0.2}$ | $77.81_{0.2}$ | $\mathbf{78.91_{0.9}}$ |
| ResNet101 | $94.77_{0.1}$ | $93.14_{0.1}$ | $94.73_{0.9}$ | $94.23_{0.1}$ | $94.22_{0.1}$ | $\mathbf{95.51_{0.1}}$ | $78.76_{0.2}$ | $73.23_{0.4}$ | $72.19_{0.2}$ | $75.46_{0.3}$ | $78.82_{0.1}$ | $\mathbf{79.74_{0.3}}$ |
| DenseNet121 | $95.11_{0.1}$ | $93.74_{0.2}$ | $94.54_{0.1}$ | $94.97_{0.1}$ | $94.99_{0.1}$ | $\mathbf{95.29_{0.1}}$ | $78.61_{0.2}$ | $75.38_{0.3}$ | $72.54_{0.9}$ | $77.09_{0.3}$ | $78.70_{0.3}$ | $\mathbf{79.03_{0.2}}$ |
| MobileNetV3 | $92.13_{0.2}$ | $91.95_{0.1}$ | $91.4_{3.1}$ | $91.92_{0.1}$ | $91.91_{0.2}$ | $\mathbf{92.89_{0.1}}$ | $73.81_{0.2}$ | $65.64_{0.2}$ | $60.78_{3.6}$ | $69.87_{0.3}$ | $68.01_{0.2}$ | $\mathbf{73.15_{0.2}}$ |
| Tiny Swin | $80.08_{0.2}$ | $87.47_{0.2}$ | $78.34_{0.2}$ | $66.84_{0.3}$ | $68.44_{0.2}$ | $\mathbf{89.08_{0.1}}$ | $57.43_{0.3}$ | $62.20_{0.2}$ | $54.12_{0.3}$ | $36.12_{0.3}$ | $33.75_{0.3}$ | $\mathbf{66.47_{0.2}}$ |
| FocalNet | $80.87_{0.2}$ | $85.65_{0.1}$ | $71.03_{0.3}$ | $42.92_{0.2}$ | $41.49_{0.2}$ | $\mathbf{86.92_{0.1}}$ | $45.66_{0.3}$ | $52.88_{0.3}$ | $38.05_{0.3}$ | $11.23_{0.3}$ | $11.06_{0.3}$ | $\mathbf{52.9_{0.1}}$ |
| CCT-2/3×2 | $73.12_{0.2}$ | $83.95_{0.1}$ | — | $34.63_{1.1}$ | $35.1_{0.8}$ | $\mathbf{84.63_{0.3}}$ | $52.12_{1.2}$ | $60.14_{1.1}$ | — | $8.06_{0.6}$ | $9.76_{0.3}$ | $\mathbf{60.63_{0.6}}$ |

*Note that Adam and AdaFisher were used for all CNN architectures, while AdamW and AdaFisherW were applied for all ViT experiments.

(b) With Cutout

| Network | CIFAR10 | | | | | | CIFAR100 | | | | | |
|---|---|---|---|---|---|---|---|---|---|---|---|---|
| | SGD | Adam | AdaHessian | K-FAC | Shampoo | AdaFisher | SGD | Adam | AdaHessian | K-FAC | Shampoo | AdaFisher |
| ResNet18 | $95.64_{0.1}$ | $94.85_{0.1}$ | $95.44_{0.1}$ | $95.17_{0.2}$ | $94.08_{0.2}$ | $\mathbf{96.25_{0.2}}$ | $76.56_{0.2}$ | $75.74_{0.1}$ | $71.79_{0.2}$ | $76.03_{0.3}$ | $76.78_{0.2}$ | $\mathbf{77.28_{0.2}}$ |
| ResNet50 | $95.71_{0.1}$ | $94.45_{0.2}$ | $95.54_{0.1}$ | $95.66_{0.1}$ | $94.59_{0.1}$ | $\mathbf{96.34_{0.2}}$ | $78.01_{0.1}$ | $74.65_{0.5}$ | $75.81_{0.3}$ | $77.40_{0.4}$ | $78.07_{0.4}$ | $\mathbf{79.77_{0.4}}$ |
| ResNet101 | $95.98_{0.2}$ | $94.57_{0.1}$ | $95.29_{0.6}$ | $96.01_{0.1}$ | $94.63_{0.1}$ | $\mathbf{96.39_{0.1}}$ | $78.89_{0.2}$ | $75.56_{0.3}$ | $73.38_{0.2}$ | $77.01_{0.4}$ | $78.83_{0.2}$ | $\mathbf{80.65_{0.4}}$ |
| DenseNet121 | $96.09_{0.1}$ | $94.86_{0.1}$ | $96.11_{0.1}$ | $96.12_{0.1}$ | $95.66_{0.1}$ | $\mathbf{96.72_{0.1}}$ | $80.13_{0.4}$ | $75.87_{0.4}$ | $74.80_{0.9}$ | $79.79_{0.2}$ | $80.24_{0.3}$ | $\mathbf{81.36_{0.3}}$ |
| MobileNetV3 | $94.43_{0.2}$ | $93.32_{0.1}$ | $92.86_{3.1}$ | $94.34_{0.1}$ | $93.81_{0.2}$ | $\mathbf{95.28_{0.1}}$ | $73.89_{0.3}$ | $70.62_{0.3}$ | $56.58_{4.5}$ | $73.75_{0.3}$ | $70.85_{0.3}$ | $\mathbf{77.56_{0.1}}$ |
| Tiny Swin | $82.34_{0.2}$ | $87.37_{0.6}$ | $84.15_{0.2}$ | $64.79_{0.5}$ | $63.91_{0.4}$ | $\mathbf{88.74_{0.4}}$ | $54.89_{0.4}$ | $60.21_{0.4}$ | $56.86_{0.5}$ | $34.45_{0.4}$ | $30.39_{1.2}$ | $\mathbf{66.05_{0.5}}$ |
| FocalNet | $82.03_{0.2}$ | $86.23_{0.1}$ | $64.18_{0.2}$ | $38.94_{0.8}$ | $37.96_{0.7}$ | $\mathbf{87.90_{0.1}}$ | $47.76_{03}$ | $52.71_{0.5}$ | $32.33_{0.3}$ | $9.98_{0.6}$ | $9.18_{0.1}$ | $\mathbf{53.69_{0.3}}$ |
| CCT-2/3×2 | $78.76_{0.3}$ | $83.89_{0.4}$ | — | $33.08_{2.3}$ | $35.16_{0.4}$ | $\mathbf{84.94_{0.3}}$ | $54.05_{0.4}$ | $59.78_{0.5}$ | — | $7.17_{0.2}$ | $8.60_{0.1}$ | $\mathbf{62.91_{0.5}}$ |

*Note that Adam and AdaFisher were used for all CNN architectures, while AdamW and AdaFisherW were applied for all ViT experiments.

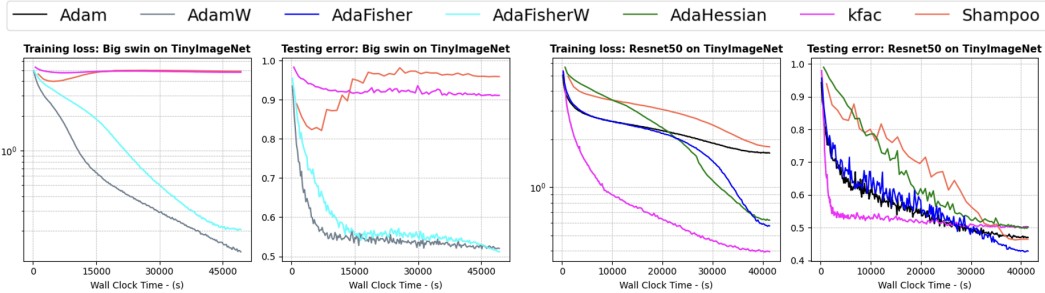

Figure 14: WCT training loss and testing error curves of several optimizers on Tiny ImageNet dataset, ResNet-50 and Big Swin with batch size of 256. AdaFisher consistently achieves lower test error as compared to Adam, AdaHessian, K-FAC and Shampoo. The final accuracy results are reported in Table 12.

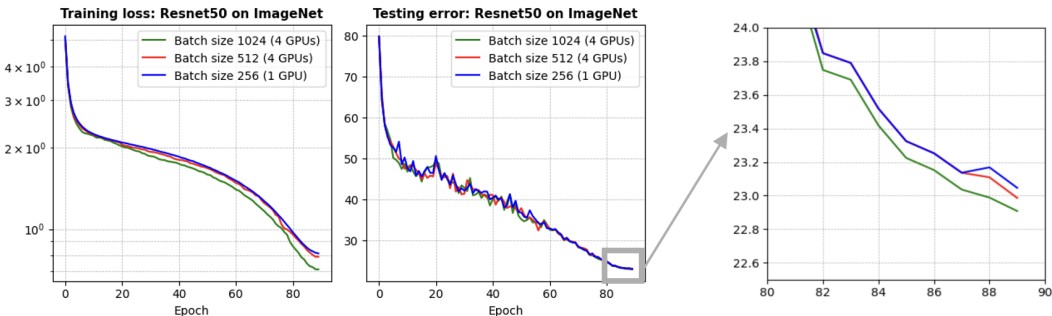

Figure 15: Performance of distributed AdaFisher using ResNet50 on ImageNet-1k with different batch sizes for 90 epochs. The final accuracy results are reported in Table 3.

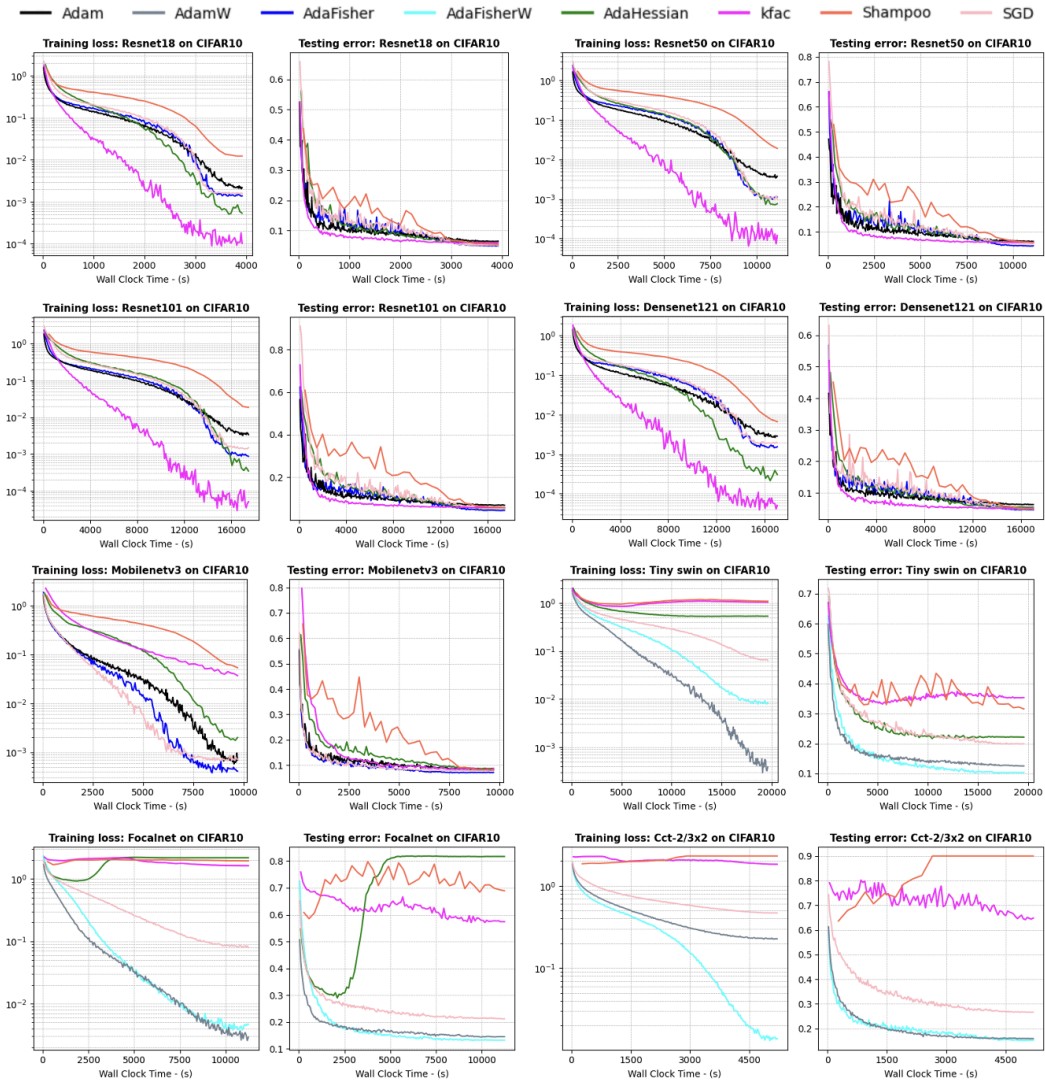

Figure 16: WCT training loss, test error, for CNNs and ViTs on CIFAR10 experiments, without Cutout. A batch size of 256 was used, and all networks were tuned using ResNet18 applied on CIFAR10. The final accuracy results are reported in Table 13 (a).

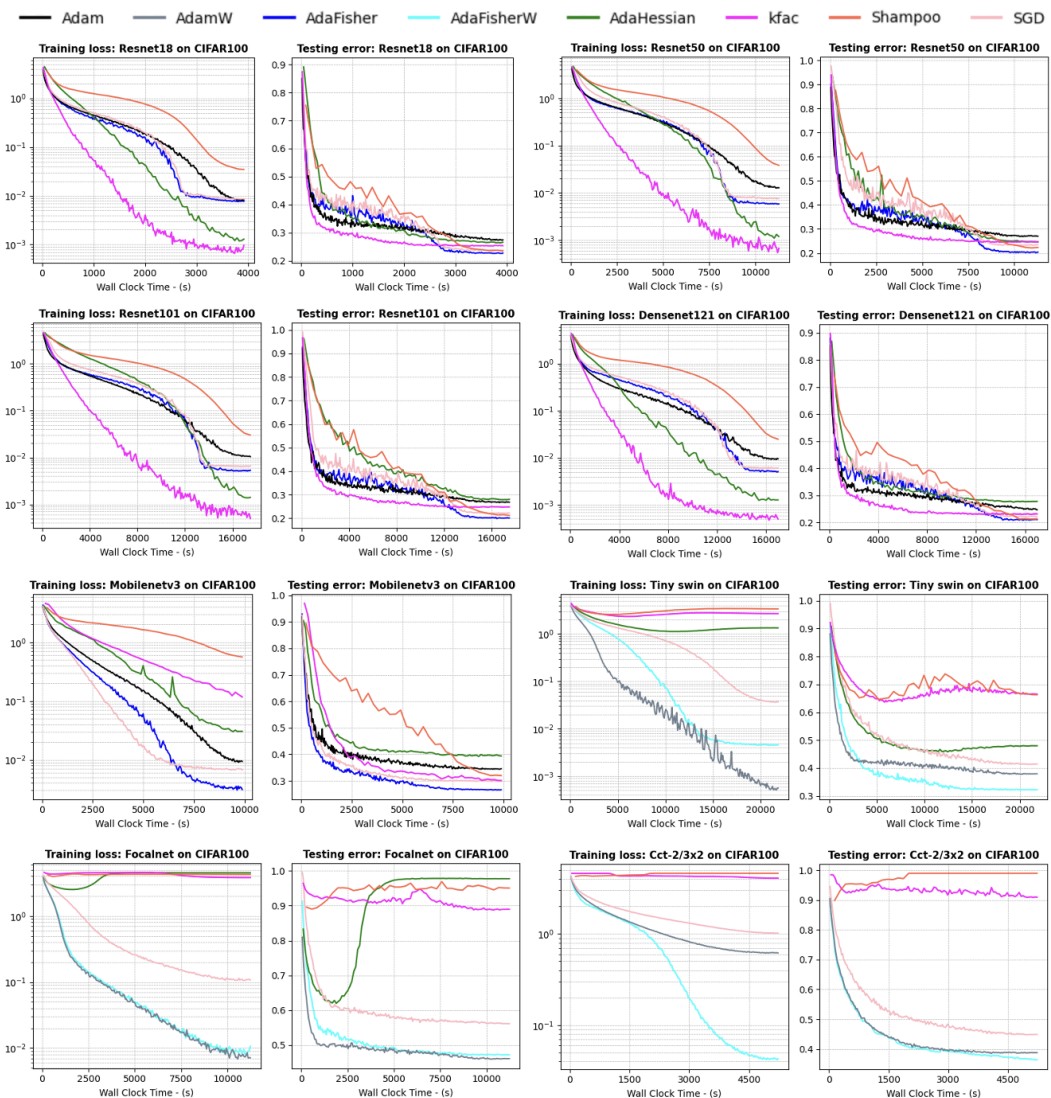

Figure 17: WCT training loss, test error, for CNNs and ViTs on CIFAR100 experiments, without Cutout. A batch size of 256 was used, and all networks were tuned using ResNet18 applied on CIFAR10. The final accuracy results are reported in Table 13 (a).

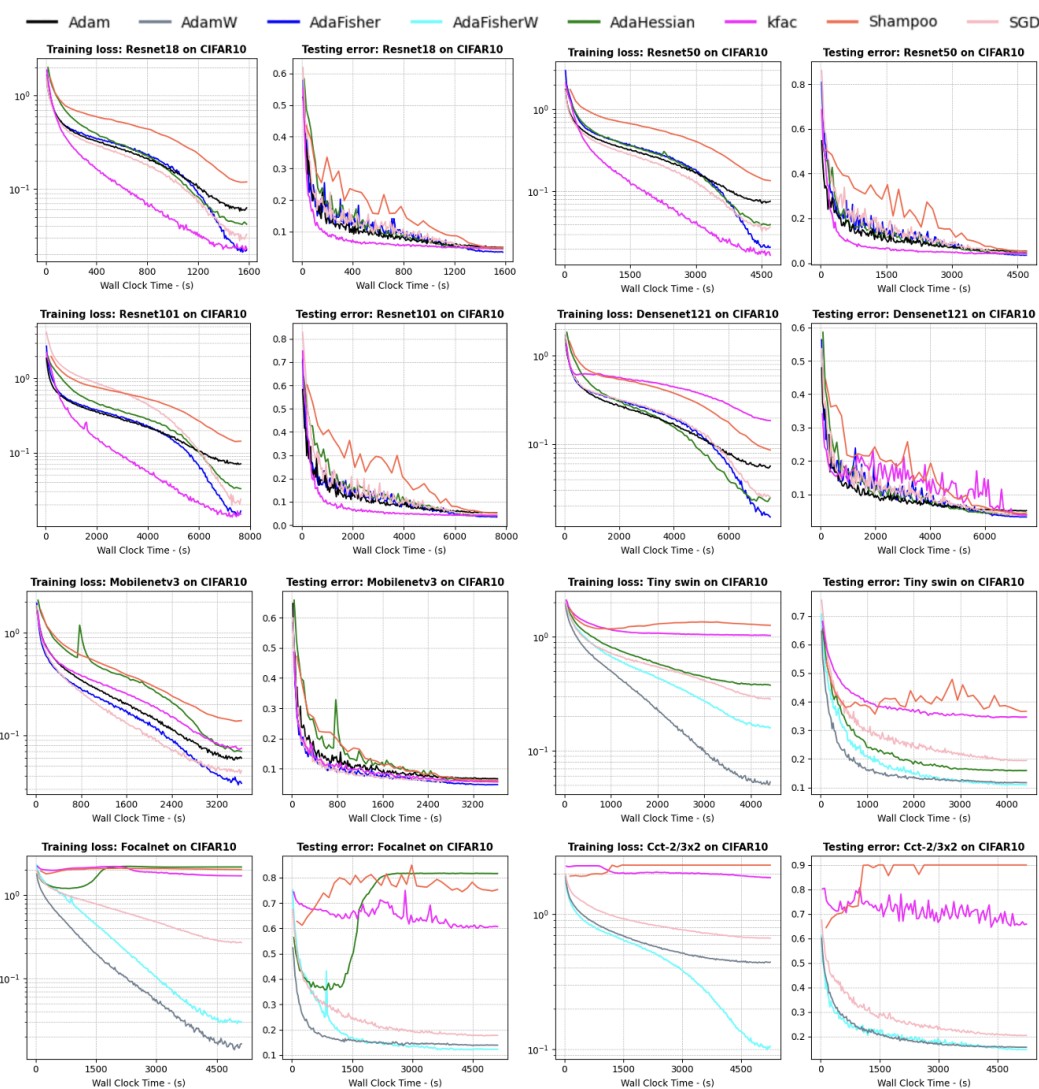

Figure 18: WCT training loss, test error, for CNNs and ViTs on CIFAR10 experiments, with Cutout. A batch size of 256 was used, and all networks were tuned using ResNet18 applied on CIFAR10. The final accuracy results are reported in Table 13 (b).

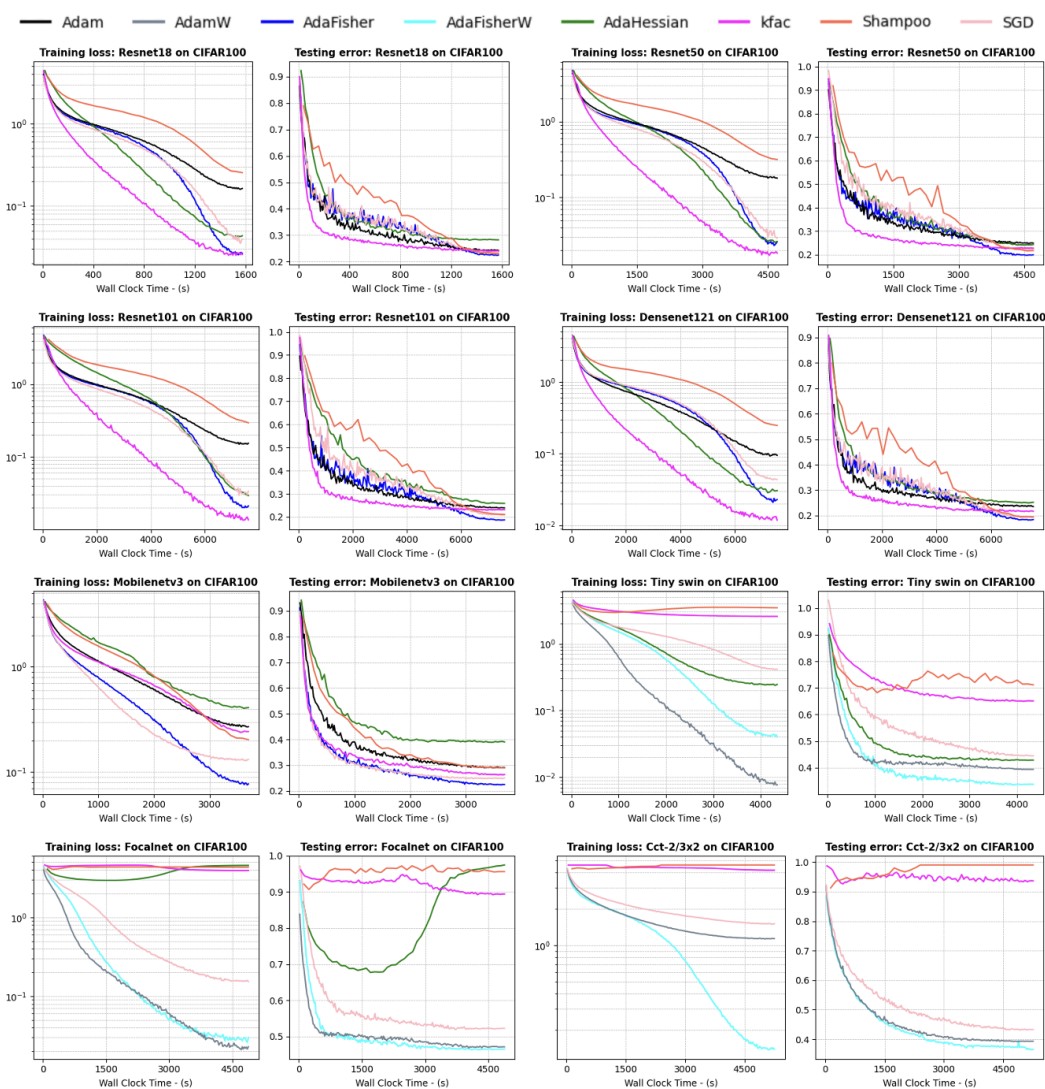

Figure 19: WCT training loss, test error, for CNNs and ViTs on CIFAR100 experiments, with Cutout. A batch size of 256 was used, and all networks were tuned using ResNet18 applied on CIFAR10. The final accuracy results are reported in Table 13 (b).

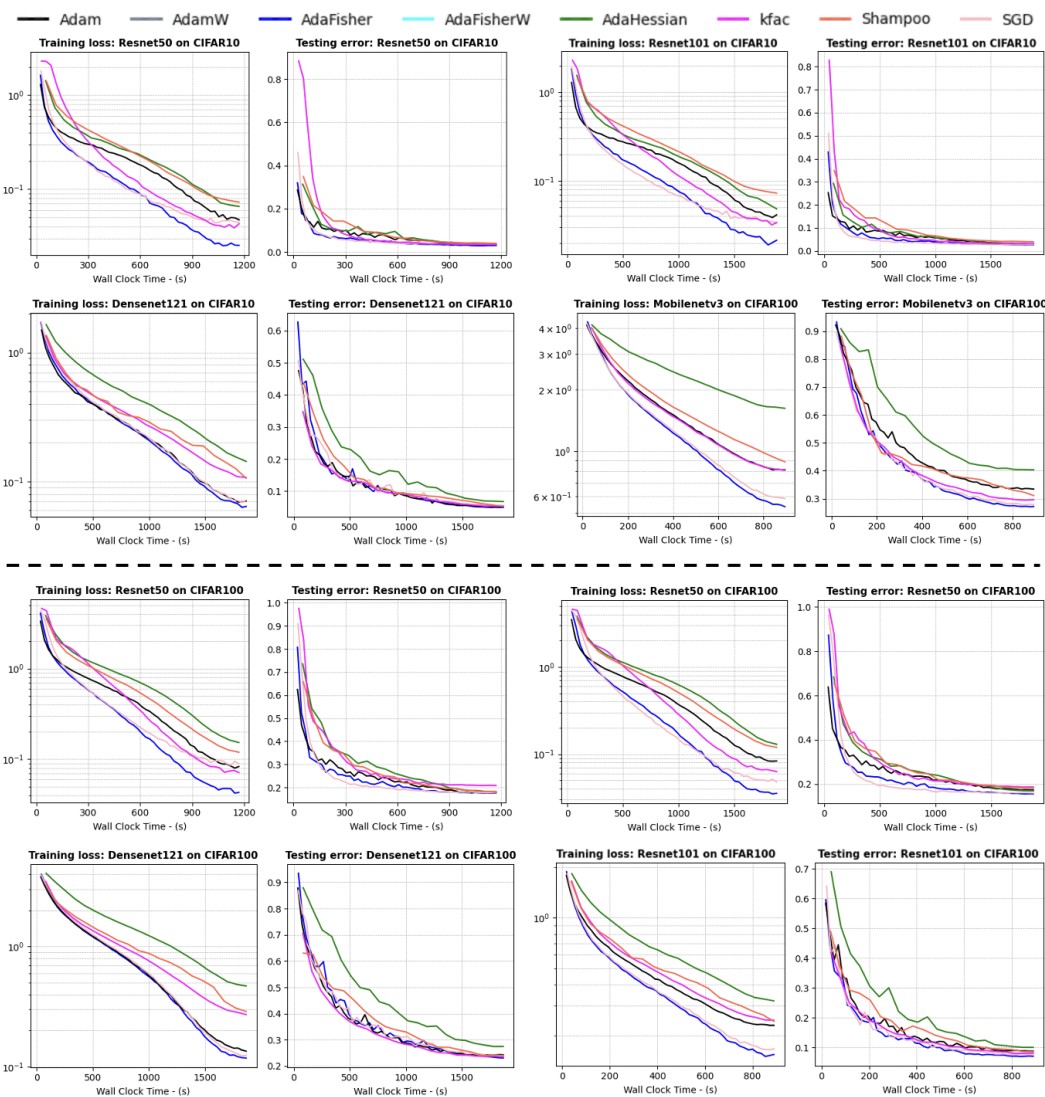

Figure 20: WCT training loss and test error for CNNs on CIFAR-10/100 experiments with pretrained weights on ImageNet-1k. A batch size of 256 was used, and all networks were tuned using ResNet50 applied on CIFAR10. The final accuracy results are reported in Table 4.

### D.2.5 COMPARISON WITH OTHER RELEVANT METHODS

In this section, we compare AdaFisher with six baseline optimizers for image classification: SGD, Adam/AdamW, AdaHessian, KFAC, and Shampoo. These baselines were selected because they either represent the current state of the art or utilize second-order gradients, making them suitable comparisons for evaluating second-order optimizers. However, other optimizers, such as AdaFactor Shazeer & Stern (2018) and EVA Zhang et al. (2023), are also relevant in this context. AdaFactor is an enhanced Adam memory-efficient optimizer that approximates second-order moments using row and column factorizations, reducing memory consumption for large-scale models. EVA is a second-order optimizer designed to leverage the FIM with efficient matrix inversion techniques. Therefore, we experimentally compare AdaFisher against the optimizer baselines, including Eva and AdaFactor. Regarding the HPs for EVA, we used the optimal values reported in its original paper and trained the model for 119 epochs using the WCT technique. For AdaFactor, we fine-tuned the learning rate as described in Appendix D.2.1, identifying 0.001 as the optimal value, and trained the model for 216 epochs. Figure 21 illustrates the performance comparison on two distinct models: ResNet-18 with CIFAR-100 and MobileNetV3 with CIFAR10. The same data augmentation techniques were applied across all experiments, as detailed in Appendix D.2.2. The best test accuracies achieved are summarized in Table 14. AdaFisher demonstrates superior performance compared to the new optimizer baselines, outperforming both EVA and AdaFactor.

Table 14: Performance comparison of AdaFisher and other optimizers using ResNet-18 (CIFAR100) and MobileNet-V3 (CIFAR10). Reported using WCT of 200 AdaFisher training epochs as the cutoff.

| Network—Optimizer | SGD | Adam | AdaFactor | AdaHessian | K-FAC | Eva | Shampoo | AdaFisher |
|---|---|---|---|---|---|---|---|---|
| MobileNet-V3 | 94.43 | 93.32 | 93.21 | 92.86 | 94.34 | 94.41 | 93.81 | **95.28** |
| ResNet-18 | 76.56 | 75.74 | 69.45 | 71.79 | 76.03 | 76.69 | 76.78 | **77.28** |

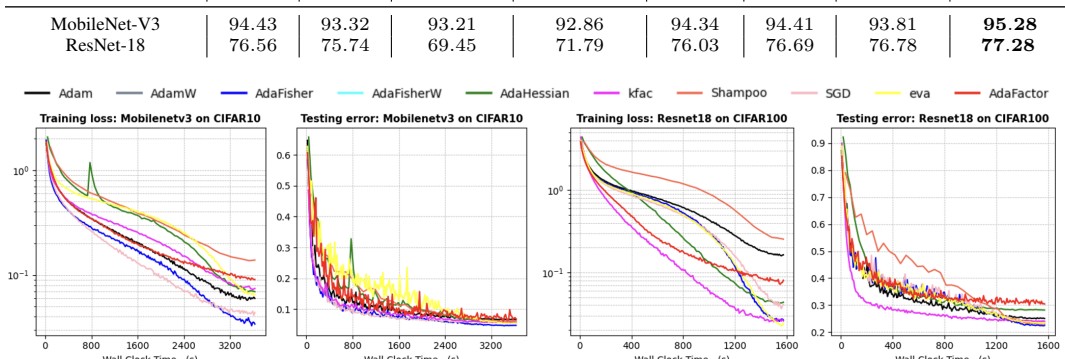

Figure 21: WCT training loss, test error, for ResNet-18 on CIFAR100 and MobileNet-V3 on CIFAR10. A batch size of 256 was used. The final accuracy and training time results are summarized in Table 14.

Table 15: Performance comparison of AdaFisher and other optimizers using (a) ResNet-18 and (b) MobileNet-V3 on CIFAR100 for 200 epochs.

(a) ResNet-18

| Optimizer | SGD | Adam | AdaFactor | AdaHessian | K-FAC | Eva | Shampoo | AdaFisher |
|---|---|---|---|---|---|---|---|---|
| Test Acc | 76.52 | 75.71 | 69.78 | 76.86 | 76.96 | 77.08 | **77.35** | 77.28 |
| Training Time (min) | 20.03 | 23.33 | 21.67 | 96.67 | 46.46 | 43.18 | 216.67 | 26.58 |

(b) MobileNet-V3

| Optimizer | SGD | Adam | AdaFactor | AdaHessian | K-FAC | Eva | Shampoo | AdaFisher |
|---|---|---|---|---|---|---|---|---|
| Test Acc | 73.42 | 70.53 | 71.08 | 62.36 | 75.16 | 75.48 | 70.65 | **77.56** |
| Training Time (min) | 50.03 | 56.63 | 54.22 | 206.28 | 116.86 | 96.78 | 487.21 | 60.12 |

(c) ResNet-50

| Optimizer | SGD | Adam | AdaFactor | AdaHessian | K-FAC | Eva | Shampoo | AdaFisher |
|---|---|---|---|---|---|---|---|---|
| Test Acc | 76.12 | 73.03 | 70.78 | 76.18 | 77.66 | 78.01 | 78.89 | **78.91** |
| Training Time (min) | 70.13 | 76.67 | 73.32 | 502.28 | 149.36 | 138.58 | 583.11 | 83.02 |

### D.2.6 COMPARISON WITH CONSISTENT EPOCH COUNTS

We evaluated AdaFisher and its counterparts, including two prominent optimizers, Eva and Adafactor, over 200 epochs on ResNet-18, ResNet-50 and MobileNet-V3 using the CIFAR100 dataset.

Figure 22 illustrates the training loss and test error trends over epochs, along with the best test error achieved as a function of training time per epoch for all optimizers across both models. Table 15 summarizes the highest test accuracy and total training time for each method on both network architectures. Notably, while Shampoo achieved marginally better test accuracy than AdaFisher on ResNet-18, it required approximately eight times longer training time. Conversely, AdaFisher outperformed all baseline optimizers, including Shampoo, in the MobileNet-V3 and ResNet-50 experiments, achieving superior test accuracy while maintaining high efficiency comparable to first-order optimizers.

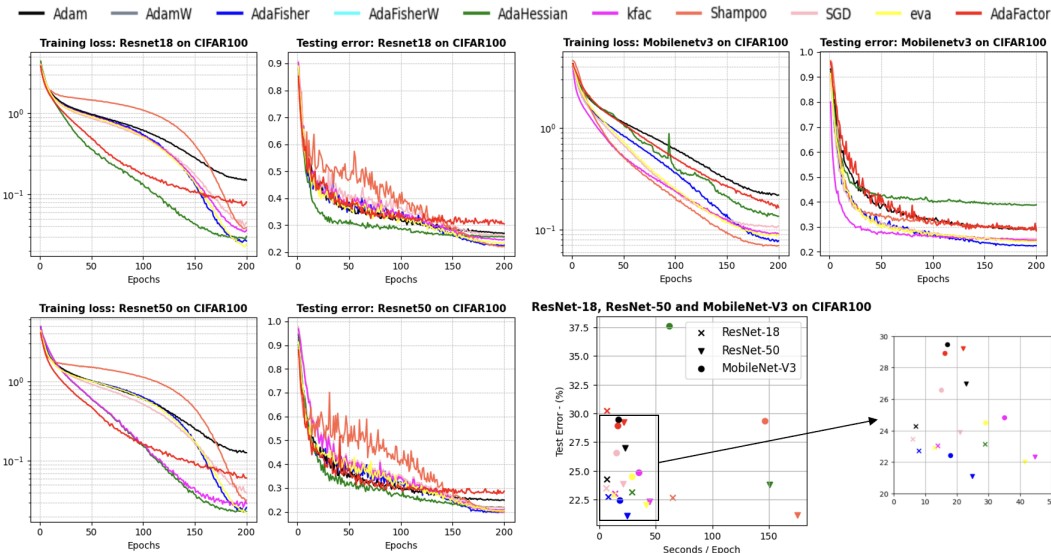

Figure 22: Performance comparison of AdaFisher and other well-finetuned optimizers at their best performances using ResNet-18 and MobileNet-V3 on CIFAR-100 for 200 epochs. A batch size of 256 was used. The final accuracy and training time results are summarized in Table 15.

### D.2.7  COMPARISON OF TRAINING SPEED AND MEMORY UTILIZATION

As discussed in Section 4.4, AdaFisher emerges as a balanced trade-off between time complexity and performance. Similarly, its memory footprint is comparable to that of Adam, showcasing efficient VRAM utilization. We extend our stability analysis to the CIFAR10 dataset to provide a dataset-independent evaluation of performance metrics, as depicted in panel (A) of Figure 23. Additionally, we analyze the memory usage for different batch sizes using the ResNet-50 model on the CIFAR-10/100, presented in panel (B) of Figure 23. The analysis reveals that AdaFisher while maintaining high accuracy levels, uses memory comparably to Adam, especially evident in larger batch sizes. This suggests that AdaFisher can achieve competitive performance without excessive VRAM consumption, making it an optimal choice for scenarios with memory constraints.

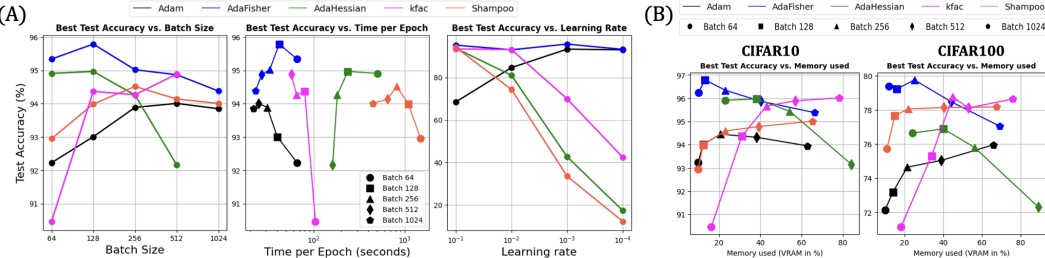

Figure 23: (A) Performance comparison of AdaFisher and other optimizers across various batch sizes, epoch times and learning rates (with a batch size of 256), evaluated using the ResNet50 on the CIFAR-10. (B) Performance comparison of AdaFisher and other optimizers regarding the memory used, assessed using ResNet50 and CIFAR10/100 across different batch sizes. This figure highlights how AdaFisher competes closely with Adam in terms of memory efficiency and performance.

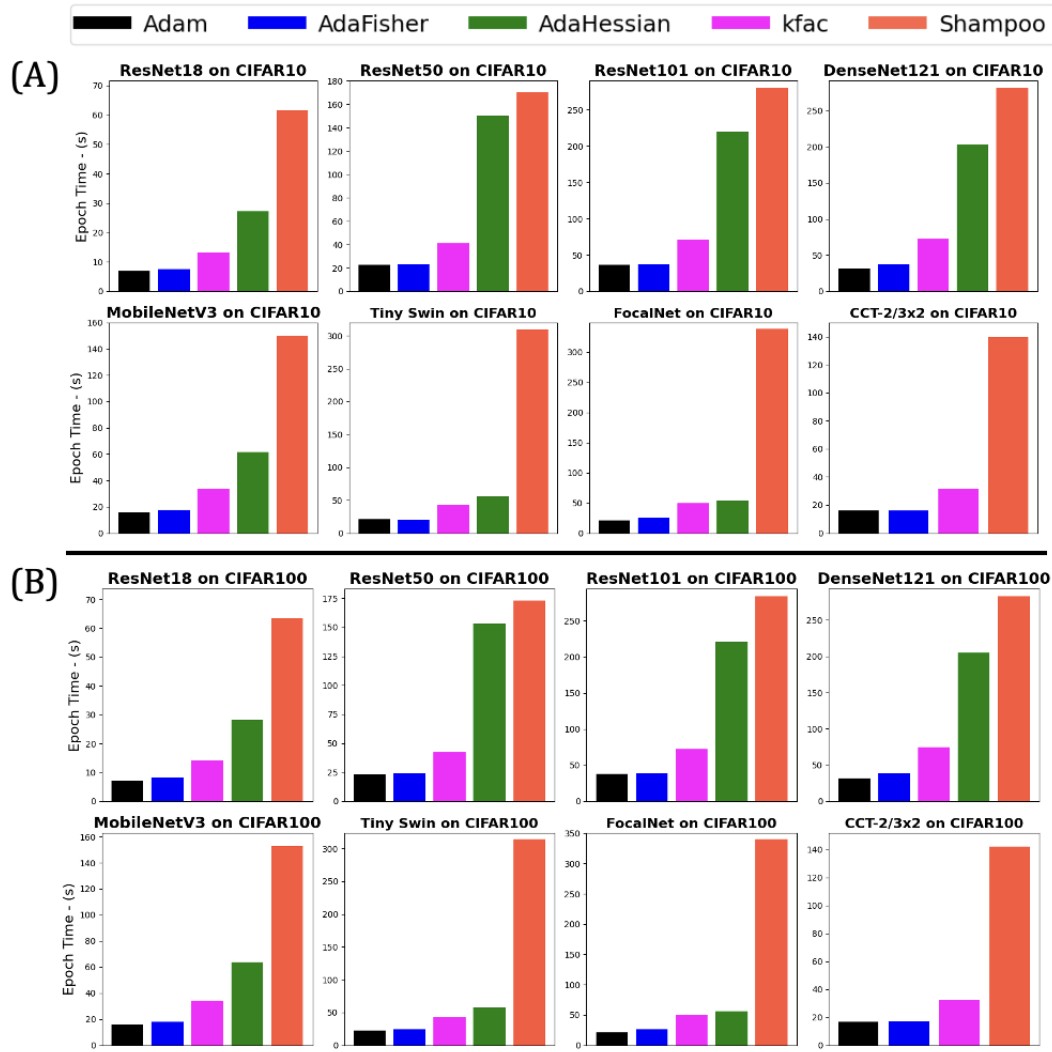

Figure 24: Epoch times for various networks on CIFAR10 (A) and CIFAR100 (B) using Adam, AdaFisher, K-FAC, AdaHessian and Shampoo.

**Epoch Times.** Continuing our analysis of the time complexity for each optimizer, we present in Figure 24 the epoch times for various network architectures and datasets. Specifically, we compare the epoch times of Adam, AdaFisher, K-FAC, AdaHessian, and Shampoo optimizers on CIFAR10 and CIFAR100 datasets. As depicted in Figure 24 panel (A), AdaFisher demonstrates a comparable training time to Adam across multiple network architectures on the CIFAR10 dataset. This indicates that AdaFisher achieves efficient optimization without incurring significant additional computational costs. Similarly, in Figure 24 panel (B), we observe that the epoch times for AdaFisher remain close to those of Adam on the CIFAR100 dataset. While K-FAC and AdaHessian exhibit increased training times, Shampoo shows the highest epoch times across all tested networks. This further highlights the efficiency of AdaFisher as an optimizer, combining the advantages of advanced optimization techniques with practical training times.

## D.3 LANGUAGE MODELING

### D.3.1 DATASET DETAILS

The Wikitext-2 dataset, derived from high-quality Wikipedia articles, contains over two million words and is structured into training, validation, and test sets. It is widely used for benchmarking language models in natural language processing, especially assessing perplexity to evaluate pre-

dictive performance. This dataset offers a balance between computational efficiency and linguistic complexity, making it ideal for practical language model training and evaluation.

### D.3.2 NETWORK DETAILS

**Network.** We utilize a streamlined GPT-1 architecture, which incorporates four self-attention layers, a reduction from the original twelve. This configuration retains core modeling capabilities while reducing complexity, encompassing a total of 28,351,488 learnable parameters.
**Embeddings & Parameter Sharing.** To expedite training, we employ pretrained embeddings from OpenAI's GPT, leveraging the benefits of parameter sharing for enhanced efficiency and faster convergence.

### D.3.3 HPS

The model underwent training for 50 WCT epochs using AdaFisher on the WikiText-2 and PTB datasets, with the final epoch counts for each optimizer detailed in Table 16. For AdamW, we follow

Table 16: Final selected epoch counts for various optimizers across language modeling task

| AdamW | AdaHessian | Shampoo | AdaFisherW |
|---|---|---|---|
| 55 | 18 | 12 | 50 |

the learning rate setting in ElNokrashy et al. (2022). For the other optimizers, we select the learning rate by doing a grid search of $\{0.3, 0.15, 0.1, 0.05, 0.03, 0.015, 0.01, \ldots, 1e^{-5}\}$. We tabulate the learning rate that we use in Table 17. The batch size was configured to 32, and the weight decay was established at $0.1$. Despite optimizing the configuration of HPs, Shampoo failed to converge, and K-FAC could not be trained at all.

Table 17: Final selected learning rates for each optimizer, tuned using GPT1 on WikiText-2 and PTB using a batch size of 32. We selected based on final validation PPL.

| AdamW | AdaHessian | Shampoo | AdaFisherW |
|---|---|---|---|
| $5e^{-5}$ | 0.015 | 0.003 | $1e-4$ |

### D.3.4 RESULTS

Figure 25 displays the training loss and testing error curves, clearly showing that AdaFisher surpasses both Adam and AdaHessian in performance on the WikiText-2 and PTB datasets.

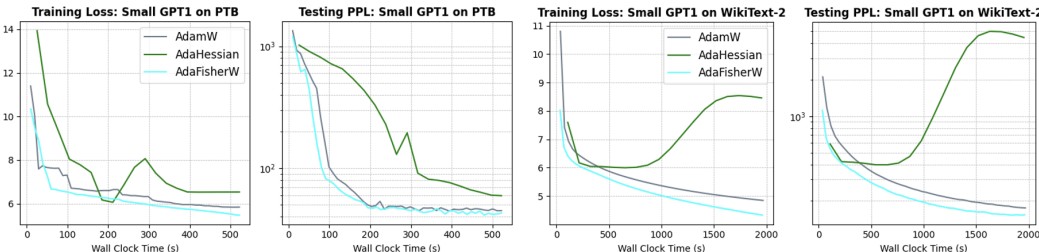

Figure 25: Training Loss and Test Perplexity of Small GPT-1 Model on WikiText-2 and PTB Datasets. Experiments were conducted using a batch size of 32 and optimal settings for all optimizers.

