# OpenReview forum: "AdaFisher: Adaptive Second Order Optimization via Fisher Information"
_ICLR.cc/2025/Conference — ICLR 2025 Poster_

### Official Review · Reviewer_K3HY · 2024-10-28

**Soundness:** 3
**Presentation:** 3
**Contribution:** 3
**Rating:** 8
**Confidence:** 3

**Summary:**

The authors introduce **AdaFisher**, a novel adaptive second-order optimizer that uses a block-diagonal approximation of the Fisher Information Matrix (FIM) to improve both convergence speed and computational efficiency in second-order optimization. The paper aims to address the limitations of current second-order methods, such as high computational costs and poor generalization, by proposing a diagonal block-Kronecker approximation of the FIM. The authors demonstrate that AdaFisher outperforms several SOTA optimizers on tasks such as image classification and language modeling.

**Strengths:**

1. **Novel methodology:** The introduction of a diagonal block-Kronecker approximation applicable to various layers offers an interesting balance between computational efficiency and the use of curvature information typically employed in second-order optimization.

2. **Empirical results:** The paper provides comprehensive experimental evidence showing that AdaFisher outperforms baseline methods (Adam, K-FAC, AdaHessian, etc.) on benchmark datasets such as CIFAR-10, CIFAR-100, ImageNet, and WikiText-2 across different network architectures.

3. **Stability:** AdaFisher shows strong stability across varying learning rates and batch sizes, reducing the need for extensive hyperparameter tuning, which is a common challenge when training deep models.

4. **Theoretical contribution:** The paper presents a rigorous theoretical convergence analysis for both convex and non-convex cases, asserting a convergence rate of  $O(\log T / \sqrt{T})$ , similar to Adam-type methods. The derivation of the update rules using the diagonal approximation of the FIM is clearly explained.

**Weaknesses:**

1. The paper considers nonconvex optimization, but in reality DNNs are **nonsmooth**, due to operations such as ReLU and max-pooling. It would be valuable to discuss this limitation in Section 3.4.

2. **Hyperparameter tuning:** It is unclear how many epochs were used for hyperparameter tuning, especially in grid search experiments (Appendix D). Providing more details on the tuning process would enhance the reproducibility of the results.

3. **Comparision with AdamW:** AdamW has become the most widely used optimizer due to its improved handling of weight decay. It would have been interesting to compare AdamW directly with AdaFisherW. (in particular, for image classification).

**Questions:**

1.	**Figure 1:**  I try reproducing Figure 1 using your code. In my experiments, AdaFisher did not consistently converge to similar local minima across multiple runs. Could you discuss the sensitivity to initialization or provide further insights?

2.	**Algorithm 1:** How did you choose the values for $\gamma_1$ and $\gamma_2$? You mentioned that “the decay factors $\gamma_1$ and $\gamma_2$ for AdaFisher were tuned within $\(\{0.1, 0.2, \dots, 0.9, 0.99\}\)$, but you did not specify which values were  used. Providing these details would help practitioners apply AdaFisher effectively.

3.	**Related work:** The paper does not mention some other recent second-order optimizers (e.g., Sophia, INNA, etc.), which could be relevant in the context of adaptive second-order methods. Including these in the discussion would provide a more comprehensive overview of related work.

4.	**Table 3:** I was surprised by the Top-1 accuracy of Adam for ResNet50 on ImageNet. Could you clarify which hyperparameters were used for Adam in this experiment?

---

> ### Author Response · Authors · 2024-11-20
> **Rebuttal (part 1/2)**
>
> Dear Reviewer K3HY,
>
> Thank you for your positive feedback and for highlighting the strengths of our work. We appreciate your recognition of the novel diagonal block-Kronecker approximation for balancing efficiency and curvature use, AdaFisher’s strong empirical performance across benchmarks, its stability in varying hyperparameters, and the rigorous theoretical convergence analysis. Your insights validate our contributions and encourage further improvements.
>
> In response to the comments in the Weaknesses and Questions sections:
>
> **[R-K3HY-C1] Regarding the non-convex optimization**: Thank you for raising this important point. We acknowledge that many DNN architectures introduce nonsmoothness due to operations like ReLU and max pooling. While our theoretical analysis primarily focuses on regret bounds for convex and nonconvex optimization, nonsmoothness presents additional challenges that are not explicitly addressed in the current version of our work. That said, nonsmoothness is common in modern deep learning, and optimizers like AdaFisher are designed to handle such challenges in practice, even if theoretical guarantees are harder to establish. For example, subgradient methods have been shown to work well with nonsmooth objectives, and many nonsmooth functions encountered in DNNs (e.g., ReLU) are piecewise linear, which can still be effectively optimized using standard techniques [1]. As the reviewer suggested, we have included a discussion in Section 3.4 in the revised manuscript (highlighted in blue text) to clarify the impact of nonsmoothness and its implications for our theoretical framework. Additionally, we emphasize that our empirical results demonstrate the robustness and effectiveness of AdaFisher, even in the presence of nonsmooth operations in DNNs.
>
> **[R-K3HY-C2] Hyperparameter Tuning**: We thank the reviewer for highlighting the missing information on the number of epochs in Appendix D. We used 50 epochs for each fine-tuning experiment and have now clarified this in detail in the revised manuscript to enhance reproducibility and transparency.
>
> **[R-K3HY-C3] Comparison with AdamW**:  We appreciate the reviewer’s suggestion to clarify the comparison with AdamW. This is a typo—Adam/AdamW and AdaFisher/AdaFisherW are interchangeable when it comes to CNNs and ViTs. Hence, we compared AdaFisherW and AdamW across all ViTs and transformer-based models. In Table 2, we now specify that all ViT experiments (including Tiny Swin, FocalNet, and CCT-2/3x2) used AdamW and AdaFisherW, given its well-known performance with transformer architectures, while Adam is more effective in CNN-based architectures [2]. This detail has been added in the revised manuscript. Additionally, Figures 16–19 in Section D.2.4 illustrate that AdamW is indeed represented in the ViT experiments, showing the corresponding loss and test error.
>
> **[R-K3HY-C4] Regarding Figure 1**: We thank the reviewer for highlighting the variability in convergence across multiple runs. Sensitivity to initialization is a known phenomenon that impacts all optimizers, including AdaFisher, as parameter initialization plays a critical role in shaping training dynamics and the resulting local minima [3,4]. To address this, we have added the following sentence to the revised manuscript in Appendix C: “_Note that while the results may vary due to the stochastic nature of parameter initialization, AdaFisher typically converges to a better local minimum compared to its counterparts._” This statement reflects the observed behavior of AdaFisher in our experiments.
>
> **[R-K3HY-C5] Regarding Algorithm 1**: We thank the reviewer for their question regarding the choice of values for $\gamma_1$ and $\gamma_2$. As detailed in Appendix D.2.1 of the revised manuscript, the current best-performing values for these decay factors in AdaFisher are $\gamma_1 = 0.92$ and $\gamma_2 = 0.08$, which naturally sum to $1$. As also raised by Reviewer jye4, **[R-jye4-C2]**, we acknowledge that defining both $\gamma_1$ and $\gamma_2$ as independent parameters is not strictly necessary as $\gamma_2$ is derived directly as $1 - \gamma_1$. For transparency and potential flexibility in future applications, we initially presented them as separate parameters to allow for scenarios where independent tuning might prove beneficial.
> In light of this, we have revised the manuscript to clarify that $\gamma$ (previously $\gamma_1$) alone is sufficient, with $\gamma_2$ being implicitly defined.
>
> **[R-K3HY-C6] Related Work**: We thank the reviewer for highlighting the omission of recent second-order optimizers, such as Sophia and INNAProp, from the related work section. We have revised this section to include these and other relevant optimizers, providing a more comprehensive overview of advancements in adaptive second-order methods.

---

> > ### Author Response · Authors · 2024-11-20
> > **Rebuttal (part 2/2)**
> >
> > **[R-K3HY-C7] Regarding Table 3**: For Adam, we used 90 training epochs with a cosine annealing learning rate scheduler, a learning rate of 0.001, a batch size of 256, betas=(0.9, 0.999), and a weight decay of 1e-4. More details on data augmentation can be found in Appendix D.2.2. Under these settings, Adam achieved a Top-1 accuracy of 67.78% on ResNet-50. This result may reflect Adam’s sensitivity to parameter initialization and its tendency to converge to sharp minima, which can affect generalization despite the use of weight decay and a learning rate scheduler [5,6]. To ensure fairness, all optimizers were trained with identical data augmentation strategies and training durations.
> >
> > **[R-K3HY-Concluding Remarks]** We thank the reviewer for their time and effort in reviewing our work and we hope the reviewer would kindly consider a fresh evaluation of our work given the main clarifying points outlined above.
> >
> > **References**
> >
> > [1] Boris Hanin. “Which neural net architectures give rise to exploding and vanishing gradients?” In Proceedings of the 32nd International Conference on Neural Information Processing Systems (NIPS'18). Curran Associates Inc., Red Hook, NY, USA, 580–589.
> >
> > [2] Xiao, Tete, et al. "Early convolutions help transformers see better." Advances in neural information processing systems 34 (2021): 30392-30400.
> >
> > [3] Skorski, Maciej, Alessandro Temperoni, and Martin Theobald. "Revisiting weight initialization of deep neural networks." Asian Conference on Machine Learning. PMLR, 2021.
> >
> > [4] Huang, Xiao Shi, et al. "Improving transformer optimization through better initialization." International Conference on Machine Learning. PMLR, 2020.
> >
> > [5] Keskar, Nitish Shirish, et al. "On large-batch training for deep learning: Generalization gap and sharp minima." arXiv preprint arXiv:1609.04836 (2016).
> >
> > [6] Foret, Pierre, et al. "Sharpness-aware minimization for efficiently improving generalization." arXiv preprint arXiv:2010.01412(2020).

---

> > > ### Comment · Reviewer_K3HY · 2024-11-20
> > > **Reply to the authors**
> > >
> > > Overall, I am satisfied with the answer of the authors. I still have a few questions.
> > >
> > > **Regarding Table 3:** For Adam, we used 90 training epochs with a cosine annealing learning rate scheduler, a learning rate of 0.001, a batch size of 256, betas=(0.9, 0.999), and a weight decay of 1e-4. More details on data augmentation can be found in Appendix D.2.2. Under these settings, Adam achieved a Top-1 accuracy of 67.78% on ResNet-50. This result may reflect Adam’s sensitivity to parameter initialization and its tendency to converge to sharp minima, which can affect generalization despite the use of weight decay and a learning rate scheduler [5,6]. To ensure fairness, all optimizers were trained with identical data augmentation strategies and training durations.
> > >
> > > Thank you for your response. To be honest, I’m a bit curious about the significant gap between Adam and AdaFisher in your results. Why did you choose these specific hyperparameters for Adam? How many runs or seeds did you perform for each optimizer?
> > >
> > > Additionally, it’s known that AdamW often outperforms Adam on ImageNet with ResNet-50. You might consider adding the performance of AdamW using reference [1].
> > >
> > > [1] Chen et al. " Symbolic Discovery of Optimization Algorithms" Advances in Neural Information Processing Systems 36 (NeurIPS 2023).

---

> > > > ### Author Response · Authors · 2024-11-20
> > > > **Reply to Reviewer K3HY**
> > > >
> > > > Thank you for your follow-up and thoughtful questions. We have addressed your questions in the following sections:
> > > >
> > > > **[R-K3HY-C8] Gap between Adam and AdaFisher**: For the hyperparameters of Adam, we used the same configuration that we fine-tuned on ResNet-18 with CIFAR-10, as detailed in Appendix D.2.1, combined with a Cosine Annealing learning rate scheduler. These hyperparameters were chosen to ensure consistency in our experimental setup. For the ImageNet experiments, we conducted a single run for each optimizer using the same random seed to maintain experimental consistency and facilitate direct comparability. While we acknowledge that multiple runs with different seeds would provide a more robust estimate of variability, our goal for the ImageNet experiments was to highlight the relative performance difference between optimizers under identical conditions. Due to the time constraints of the rebuttal period, we are unable to conduct additional runs on ImageNet at this moment. But we will perform multiple runs for all optimizers, including AdamW as per your suggestion, and include the results in the updated version of the manuscript for final consideration.
> > > >
> > > > **[R-K3HY-C9] AdamW with ResNet-50 on ImageNet**: Thank you for your suggestion. We have included the performance of AdamW on ImageNet using ResNet-50 with a batch size of 1024 in Table 3 of the revised manuscript. The top-1 accuracy achieved with AdamW is 76.34, which is lower than the accuracy obtained with AdaFisher under the same batch size, 77.09. This addition provides a direct comparison with recent experiments and further highlights the performance of AdaFisher. We appreciate your feedback and hope this addition offers greater clarity and context to our results.

---

> > > > > ### Comment · Reviewer_K3HY · 2024-11-26
> > > > >
> > > > > I appreciate the authors’ clarifications, especially regarding the empirical results, and I have decided to increase my score.

---

> > > > > > ### Author Response · Authors · 2024-11-26
> > > > > > **Official Comment by Authors**
> > > > > >
> > > > > > We sincerely appreciate your time and effort in reconsidering the score. If you have any additional questions, please do not hesitate to reach out—we would be happy to provide further clarification.
> > > > > >
> > > > > > Best regards,
> > > > > >
> > > > > > The Authors

---

### Official Review · Reviewer_jye4 · 2024-11-01

**Soundness:** 3
**Presentation:** 3
**Contribution:** 2
**Rating:** 6
**Confidence:** 4

**Summary:**

This  paper  proposes  a new  second-order  optimizer,  AdaFisher,  which  ensures  better  convergence  while  maintaining  computational  efficiency.  Based  on  the  K-FAC optimizer,  AdaFisher  discovers  that  the  Kronecker  factor  is  diagonally  dominant  and  the  Fisher  information  matrix can be approximated  by  using  a diagonal block Kronecker approximation.  On image  classification  and  language  modeling  tasks,  AdaFisher  achieves  better  results  than  other  second-order  optimization  methods when applying Wall-Clock-Time method (training  for  the  same  amount  of  time).

**Strengths:**

The  paper  innovatively  discovers  that  the  Kronecker  factor  is  diagonally  dominant  and  proposes  a diagonal  block-Kronecker  approximation  for  the  FIM. The  resulting  AdaFisher  optimizer  shows  good  performance.  And the  paper  is  well-organized  and  easy to follow.

**Weaknesses:**

Please  refer  to  Questions.

**Questions:**

- In Section  2,  it  is  stated  that  $F_i = \cal{H_{i-1}} \otimes \cal{S_{i}}$, but  in  the  algorithm  implementation,  it  becomes  $F_i = \cal{H_{i}} \otimes \cal{S_{i}}$. Why  is  this  the  case?
- Why are two parameters, $\gamma_1$ and $\gamma_2$, introduced for calculating the exponential moving averages of $\cal{H}$ and $\cal{S}$ in the AdaFisher optimizer? Would it not suffice to introduce only one parameter?
- It is known that the SGD optimizer performs better than the Adam optimizer in image classification tasks. Why is there no comparison with SGD?
- The experimental section compares performance under the same computation time. Could you provide a performance comparison under the same number of epochs? I suspect that second-order optimizers like AdaHessian may not have converged yet. When all optimizers have converged, does AdaFisher still outperform other second-order optimizers?
- The AdaFisher optimizer introduces two parameters, $\gamma_1$ and $\gamma_2$. I see that the authors conducted extensive searches for these parameters. Could you provide the values of $\gamma_1$ and $\gamma_2$ used in different tasks for the AdaFisher optimizer? If these parameters also require extensive searching to achieve good performance, then the AdaFisher optimizer might be very limited.

I'm willing to improve my score if you address my concerns.

---

> ### Author Response · Authors · 2024-11-20
> **Rebuttal (part 1/2)**
>
> Dear Reviewer jye4,
>
> We sincerely thank the reviewer for their thorough evaluation and insightful questions regarding our work to be innovative, providing competitive results and well-organized. Their constructive feedback has helped us alleviate the quality of our manuscript and clarify key points about the AdaFisher optimizer. To address the reviewer’s concerns comprehensively, we have conducted additional experiments and provided detailed explanations for each query. For the ease of the reviewer, all new content and revisions in the manuscript are highlighted in blue text.
>
> Below, we provide responses to each reviewer’s questions.
>
> **[R-jye4-C1] Pseudo-code implementation**: We thank the reviewer for pointing out the typo in the indexing of the Kronecker factor $\mathcal{H}$. This has been corrected in the updated pseudo-code to ensure consistency. As for clarity, we recall that  $\mathcal{H}\_{i-1}$ is the output of layer $i−1$ which also corresponds to the input activation of layer $i$. Specifically, $\mathcal{H}\_0$ corresponds to the batched input data ($x$) for the first layer ($i=1$). This was purely a typographical error, and the updated manuscript reflects the correct implementation. Thank you for bringing this to our attention.
>
> **[R-jye4-C2] Gamma parameters**: We appreciate the reviewer’s keen observation regarding the non-necessity of using two separate parameters for the EMA of the Kronecker factors $\mathcal{H}$ and $\mathcal{S}$ in the AdaFisher optimizer. Upon the reviewer’s suggestion, we acknowledge that introducing both $\gamma_1$ and $\gamma_2$ is redundant, as $\gamma_2 = 1 - \gamma_1$ in our experiments. Specifically, we used $\gamma_1 = 0.92$ and $\gamma_2 = 0.08$, which naturally sum to $1$. However, for transparency and clarity in implementation, we initially opted to define them as separate parameters to allow for potential flexibility in future extensions or scenarios where independent tuning may provide benefits. In light of the reviewer’s comment, we revised the manuscript to reflect that only one parameter, $\gamma$, is strictly necessary.
>
> **[R-jye4-C3] SGD Experiments**: We thank the reviewer for emphasizing the importance of including SGD experiments. In response, we have added SGD results for image classification tasks in the revised manuscript. We fine-tuned the learning rate and the momentum hyperparameters as described in Appendix D.2.1. Prior work [1] highlights that while SGD often outperforms Adam on CNN architectures, it underperforms on Transformers [2]. Nevertheless, as shown in Table 2, AdaFisher consistently surpasses SGD, even in CNN based architectures, demonstrating its superior effectiveness across these settings.

---

> ### Author Response · Authors · 2024-11-20
> **Rebuttal (part 2/2)**
>
> **[R-jye4-C4] Comparison with consistent epoch counts**:  We thank the reviewer for the insightful question. In our initial experiments, we evaluated the performance of the optimizers using the Wall-Clock-Time (WCT) technique, as it ensures a fair comparison by allocating the same computational budget for each optimizer. This approach aligns with widely accepted practices for evaluating optimizers under realistic time constraints [3,4]. To address the reviewer’s concern, we have additionally compared all optimizer baselines, including EVA and AdaFactor (as suggested by Reviewer jcNA), under the same number of epochs. Specifically, we conducted experiments on ResNet-18, ResNet-50 and MobileNet-V3 on CIFAR100 for 200 epochs. The results, now included in the revised manuscript under Appendix D.2.6, demonstrate that while Shampoo achieved slightly better performance than AdaFisher on the ResNet-18 network, its training time was approximately eight times longer than that of AdaFisher. Conversely, AdaFisher outperformed all other optimizers, including Shampoo, in the MobileNet-V3 and ResNet-50 experiments, delivering superior results with significantly better efficiency compared to the other second order optimizers. Additionally, we have summarized these findings in the following table 1 which provides a comprehensive overview of the highest test accuracies and training efficiencies for all methods discussed.
>
> **Table 1:** Performance comparison of AdaFisher and other optimizers using (a) ResNet-18, (b) MobileNet-V3, and (c) ResNet-50 on CIFAR100 for 200 epochs.
>
> ### (a) ResNet-18
> |Optimizer|SGD|Adam|AdaFactor|AdaHessian|K-FAC|Eva|Shampoo|AdaFisher|
> |---------|---|----|---------|----------|-----|---|-------|---------|
> |Test Acc|76.52|75.71|69.78|76.86|76.96|77.08|**77.35**|77.28|
> |Tot Time (min)|20.03|23.33|21.67|96.67|46.46|43.18|216.67|26.58|
>
> ### (b) MobileNet-V3
> |Optimizer|SGD|Adam|AdaFactor|AdaHessian|K-FAC|Eva|Shampoo|AdaFisher|
> |---------|---|----|---------|----------|-----|---|-------|---------|
> |Test Acc|73.42|70.53|71.08|62.36|75.16|75.48|70.65|**77.56**|
> |Tot Time (min)|50.03|56.63|54.22|206.28|116.86|96.78|487.21|60.12|
>
> ### (c) ResNet-50
> |Optimizer|SGD|Adam|AdaFactor|AdaHessian|K-FAC|Eva|Shampoo|AdaFisher|
> |---------|---|----|---------|----------|-----|---|-------|---------|
> |Test Acc|76.12|73.03|70.78|76.18|77.66|78.01|78.89|**78.91**|
> |Tot Time (min)|70.13|76.67|73.32|502.28|149.36|138.58|583.11|83.02|
>
> **[R-jye4-C5] Gamma Fine Tuning**: We thank the reviewer for raising this point. As clarified in our previous response, **[R-jye4-C2]**, only one parameter ($\gamma$, previously $\gamma_1$) is necessary for the AdaFisher optimizer, with $\gamma_2$ implicitly defined as $1 - \gamma_1$. For all experiments in our work, $\gamma$ was fixed at $0.92$, and no extensive tuning was required. This consistency across tasks demonstrates that AdaFisher is not only effective but also user-friendly, as it avoids the need for expensive hyperparameter tuning. This simplicity makes AdaFisher a practical and efficient optimizer for a wide range of applications.
>
> **[R-jye4-Concluding Remarks]** We thank the reviewer for their time and effort in reviewing our work and we hope the reviewer would kindly consider a fresh evaluation of our work given the main clarifying points outlined above.
>
> **References**
>
> [1] Zou, Difan, et al. "Understanding the generalization of adam in learning neural networks with proper regularization." arXiv preprint arXiv:2108.11371 (2021).
>
> [2] Zhang, Yushun, et al. "Why transformers need adam: A hessian perspective." arXiv preprint arXiv:2402.16788 (2024).
>
> [3] Kaddour, Jean, et al. "No train no gain: Revisiting efficient training algorithms for transformer-based language models." Advances in Neural Information Processing Systems 36 (2024).
>
> [4] Ye, Huigen, Hua Xu, and Hongyan Wang. "Light-MILPopt: Solving Large-scale Mixed Integer Linear Programs with Lightweight Optimizer and Small-scale Training Dataset." The Twelfth International Conference on Learning Representations.

---

> ### Comment · Reviewer_jye4 · 2024-11-21
> **Some confusion**
>
> Thanks for the clarification, but I still have some questions.
> - **Regarding $\gamma$ parameter**. In your original manuscript, $ H_i = \gamma_1 H_{i-1} + (1-\gamma_2) \hat{H}_{i} $.
> If $\gamma_1 = 1 - \gamma_2$, we have $ H_i = \gamma_1 H_{i-1} + \gamma_1 \hat{H}_{i} $. But  in your revised manuscript, I see $H_i = \gamma H_{i-1} + (1-\gamma) \hat{H}_{i}$. So I guess you used $\gamma_1 = \gamma_2$, right? Moreover, in your revised manuscript, you mentioned that `AdaFisher Decay Factors: The decay factor γ for AdaFisher was tuned within {0.1, 0.2, . . . , 0.9, 0.99}. The optimal value is: γ = 0.92.`. However, I noticed that 0.92 is not within the range you specified.
> - **Confusion about SGD's performance.** Regarding the performance of SGD on the ResNet18 model with the CIFAR-100 dataset, I saw that the value in Table 2 of Reference [1] is 78.0 and in Reference [2] is 79.49. However, in your experiment, it is only 76. Reference [1] use a batch size of 128 and train 200 epochs; Reference [2] use a batch size of 128 and train 400 epochs. I am not sure if the difference between the author's results and others' results is caused by the batch size.
> Could you run an experiment of SGD with a batch size of 128 in your benchmark?
>
> [1] Liu, Y.; Mai, S.; Cheng, M.; Chen, X.; Hsieh, C.-J.; You, Y. Random Sharpness-Aware Minimization. In NeurIPS 2022; 2022.
>
> [2] Yue, Y.; Jiang, J.; Ye, Z.; Gao, N.; Liu, Y.; Zhang, K. Sharpness-Aware Minimization Revisited: Weighted Sharpness as a Regularization Term. In KDD 2023; 2023.

---

> > ### Author Response · Authors · 2024-11-21
> > **Reply to Reviewer jye4**
> >
> > Thank you for your follow-up and clarification questions. We have provided detailed responses to your queries in the sections below:
> >
> > **[R-jye4-C6] Gamma Parameter**: Thank you for pointing this out. You are absolutely correct that $\gamma_1 = \gamma_2$. In fact, in our previous version: $\mathcal{H}\_{i} = \gamma_1 \mathcal{H}\_{i-1} + (1-\gamma_2)\hat{\mathcal{H}}\_{i}$, where $\gamma_1 = 0.92$ and $1-\gamma_2 = 0.08$. Therefore, $\gamma_1 = \gamma_2$ and leads to the final version:  $\mathcal{H}\_{i} = \gamma \mathcal{H}\_{i-1} + (1-\gamma)\hat{\mathcal{H}}\_{i}$. We apologize for the confusion. Secondly, to provide clarity regarding the grid search for $\gamma$, we confirm that the values used in our experiments were $[0.1, 0.2, \dots, 0.9, 0.91, 0.92, \dots, 0.99]$. We acknowledge that this was not explicitly stated in the manuscript, and we apologize for the oversight. Notably, the optimal value, $\gamma = 0.92$, lies within this extended range. We have revised the manuscript to explicitly detail the complete grid used for tuning, ensuring transparency and avoiding potential confusion.
> >
> > **[R-jye4-C7] SGD experiment**: Thank you for your observation regarding SGD’s performance in our experiments. To address this, we conducted an experiment on our benchmark with ResNet-18 on CIFAR-100 using a batch size of 128 for all optimizers, including SGD. The best test accuracy achieved with SGD in our setup was 77.83% (as shown in Table 2 below), which is closely aligned with the result reported in [1]. However, reproducing the exact experimental setup of [1] proved challenging due to limited details on certain hyperparameters, such as the specific learning rate schedule, momentum, weight decay, and learning rate decay. While the authors state that their setup follows the SAM paper [2], we found no explicit experiments involving ResNet-18 in [2] to confirm this. Additionally, there are ambiguities in the data augmentation strategies reported in [1]. For example, Section 4.1.4 mentions Mixup and RandAugment, whereas Section 4.3 refers to “basic augmentation” for image preprocessing. These inconsistencies could contribute to the observed differences in results, as augmentation strategies often play a significant role in improving model performance.
> >
> > **Table 2**: Performance comparison of AdaFisher and other optimizers using  ResNet-18 on CIFAR100 for 200 epochs. A batch size of **128** was used for all optimizers.
> > |Optimizer|SGD|Adam|AdaFactor|AdaHessian|K-FAC|Eva|Shampoo|AdaFisher|
> > |---------|---|----|---------|----------|-----|---|-------|---------|
> > |Test Acc|77.83|76.01|70.43|77.65|78.35|78.40|78.46|78.48|
> > | Tot Time (min) | 40.06 | 46.66 | 43.34 | 193.34 | 92.92 | 86.36 | 433.34 | 53.16 |
> >
> > We hope these clarifications help provide a better understanding of our results. Once again, we appreciate your constructive suggestions and look forward to further discussion.
> >
> > **References**
> >
> > [1] Liu, Y.; Mai, S.; Cheng, M.; Chen, X.; Hsieh, C.-J.; You, Y. Random Sharpness-Aware Minimization. In NeurIPS 2022; 2022.
> >
> > [2] Foret, Pierre, et al. "Sharpness-aware minimization for efficiently improving generalization." arXiv preprint arXiv:2010.01412(2020).

---

> > > ### Comment · Reviewer_jye4 · 2024-11-24
> > > **Reply to the author**
> > >
> > > Thanks for the clarifications. I'm willing to increase my score.

---

> > > > ### Author Response · Authors · 2024-11-24
> > > > **Official Comment by Authors**
> > > >
> > > > We sincerely appreciate your time and effort in reconsidering the score. If you have any additional questions, please do not hesitate to reach out—we would be happy to provide further clarification.
> > > >
> > > > Best regards,
> > > >
> > > > The Authors

---

### Official Review · Reviewer_5WtX · 2024-11-02

**Soundness:** 2
**Presentation:** 2
**Contribution:** 2
**Rating:** 5
**Confidence:** 5

**Summary:**

The authors use the Fisher information matrix as a precondition matrix to obtain a second order optimizer. They also provided some analysis and numerical results for the method.

My concern is that this method is too similar to a previous method "Kronecker-Factored Second-Order Optimizers Perform First-Order Descent on Neurons" by Frederik Benzing.

**Strengths:**

Using the Fisher imformation matrix as a precondition is really a good point.

**Weaknesses:**

It seems that the author didn't realize a previous work "Kronecker-Factored Second-Order Optimizers Perform First-Order Descent on Neurons" by Frederik Benzing, which is very similar to this work.

**Questions:**

1. The authors should specify the difference of the current work to the previous work "Kronecker-Factored Second-Order Optimizers Perform First-Order Descent on Neurons" by Frederik Benzing.

2. Empirically, the high-order optimizers are computationally more expensive than first order optimizers for each step and also have poorer generalizability. It is somehow strange that the numerical results in this work didn't show how the method accelerate the training process clearly but mainly about the strength of the method in generalization. The authors should explain why the method can help in enhance the generalization significantly. They should also include results obtained with SGD, since usually the generalization ability  of SGD is better than ADAM.

---

> ### Author Response · Authors · 2024-11-20
> **Rebuttal (part 1/2)**
>
> Dear Reviewer 5WtX,
>
> We thank the reviewer for their feedback, highlighting the strength of the paper of using a novel FIM for preconditioning the gradient and mentioning the related work. Please note, in addition to our official comments in this section, we have modified parts in the revised manuscript in blue color.
>
> Below are the responses to the comments in Weakness and Question sections:
>
> **[R-5WtX-C1] Comparison between FOOF and AdaFisher**: Please note we have cited the paper suggested by the reviewer “Kronecker-Factored Second-Order Optimizers Perform First-Order Descent on Neurons” from NeurIPS2021-OPT Workshop by Frederik Benzing in Section 5 (Related Work) from the revised manuscript. Moreover, we provide the following evidence that Fast First-Order Optimizer (FOOF), the optimizer introduced by Benzing, and AdaFisher are fundamentally different.
>
> FOOF is a first-order optimizer that updates weights based on neuron outputs, utilizing a preconditioning matrix of the form: $\mathcal{H}\_{i-1} \otimes \mathbf{I}$ for a given layer $i$. This approach requires constructing and inverting only the Kronecker factor $\mathcal{H}\_{i-1}$, which captures limited curvature information inherent to its first-order nature. In other words, it does not incorporate curvature information from the FIM, which is essential for capturing the complex geometry of the loss surface in neural networks. In contrast, AdaFisher is a second-order optimizer leveraging a novel approximation of the FIM,  $\tilde{F}\_{D_i} = \mathcal{H}\_{D_{i-1}} \otimes \mathcal{S}\_{D_i}$, to adaptively modulate learning dynamics. This formulation enables AdaFisher to integrate richer curvature information compared to FOOF while only using block diagonal Kronecker factors (please refer to section 3.1). Furthermore, AdaFisher introduces a novel computation of the FIM for normalization layers, enhancing the training performance on architectures where such layers are prevalent.
>
> These distinctions underline AdaFisher’s second-order nature and advanced adaptivity, setting it apart from FOOF’s first-order approach. Furthermore, the scope of the experiments in Benzing’s manuscript is also limited as compared to our work. Only some preliminary experiments are shown using MLPs on MNIST and Fashion-MNIST reporting only training loss as compared to our work which uses extensive experiments on SOTA models and CIFAR10/100, TinyImageNet and Imagenet datasets.
>
> **[R-5WtX-C2] Computational Cost and Acceleration**: It is true that second order optimizers are generally more computationally expensive than first-order methods like SGD for each iteration. However, our method, AdaFisher, is designed to mitigate this by employing a diagonal block Kronecker approximation of the FIM, significantly reducing the overhead compared to full-matrix or Kronecker-sum-based methods such as K-FAC or Shampoo. We recall that the main objective of our paper is to bridge the gap between enhanced convergence capabilities and computational efficiency in second-order optimization framework for training DNNs.
>
> **[R-5WtX-C3] Enhanced Generalization**: AdaFisher improves generalization by leveraging the FIM to better capture the loss landscape’s curvature and parameter sensitivity. As shown in Figure 11, AdaFisher’s FIM distribution narrows to smaller values with less variation compared to Adam, indicating convergence toward flatter minima, which are strongly associated with better generalization [1,2]. The diagonal block Kronecker approximation ensures AdaFisher captures key curvature information while avoiding overfitting to noise (please refer to Section 3.1), which is a common issue in some second-order methods. This combination allows AdaFisher to achieve both improved generalization and computational efficiency. We refer the reviewer to Appendix B.1 (Convergence Efficiency) for more information.

---

> > ### Author Response · Authors · 2024-11-20
> > **Rebuttal (part 2/2)**
> >
> > **[R-5WtX-C4] SGD Experiments**: We thank the reviewer for emphasizing the importance of including SGD experiments. In response, we have added SGD results for image classification tasks in the revised manuscript. We fine-tuned the learning rate and the momentum hyperparameters as described in Appendix D.2.1. Prior work [3] highlights that while SGD often outperforms Adam on CNN architectures, it underperforms on Transformers [4]. Nevertheless, as shown in Table 2, AdaFisher consistently surpasses SGD, demonstrating its superior effectiveness across these settings.
> >
> > **[R-5WtX-Concluding Remarks]** We thank the reviewer for their valuable feedback and great questions. We hope that our rebuttal fully addresses all the important salient points raised by the reviewer and we kindly ask the reviewer to potentially upgrade their score if the reviewer is satisfied with our responses. We are also more than happy to answer any further questions that arise. Please do let us know.
> >
> > **References**
> >
> > [1] Zhang, Jian, et al. "Exploring Flat Minima for Domain Generalization with Large Learning Rates." IEEE Transactions on Knowledge and Data Engineering (2024).
> >
> > [2] Foret, Pierre, et al. "Sharpness-aware minimization for efficiently improving generalization." arXiv preprint arXiv:2010.01412(2020).
> >
> > [3] Zou, Difan, et al. "Understanding the generalization of adam in learning neural networks with proper regularization." arXiv preprint arXiv:2108.11371 (2021).
> >
> > [4] Zhang, Yushun, et al. "Why transformers need adam: A hessian perspective." arXiv preprint arXiv:2402.16788 (2024).

---

> > > ### Author Response · Authors · 2024-11-22
> > > **Gentle Reminder of the Discussion Deadline**
> > >
> > > Dear Reviewer 5WtX,
> > >
> > > Thank you once again for your time and review of our manuscript. We understand that you have a busy schedule, and we kindly wish to remind you that the discussion deadline is approaching. If there are any remaining concerns or suggestions on how we can further improve our manuscript, we would greatly appreciate your feedback. Additionally, if you feel that we have successfully addressed the points raised in your initial review, we would be grateful if you could consider revisiting the score assigned to the paper.
> > >
> > > Sincerely,
> > >
> > > The Authors

---

> > > > ### Author Response · Authors · 2024-11-24
> > > > **Final Gentle Reminder - Discussion Period Ending Soon**
> > > >
> > > > Dear Reviewer 5WtX,
> > > >
> > > > We wanted to reach out one final time regarding our manuscript, as the discussion period is drawing to a close very soon. We greatly value your expertise and initial feedback, and we hope our rebuttal has adequately addressed your concerns.
> > > > If you have any remaining questions or if there are points that require further clarification, we would be most grateful to hear from you before the discussion period ends. Alternatively, if you feel satisfied with our responses, we would appreciate your consideration in updating the manuscript's evaluation.
> > > >
> > > > Best regards,
> > > >
> > > > The Authors

---

### Official Review · Reviewer_JcNA · 2024-11-04

**Soundness:** 3
**Presentation:** 3
**Contribution:** 3
**Rating:** 6
**Confidence:** 4

**Summary:**

The paper proposes a new idea: using Kronecker factored preconditioners, with diagonal factors. The Kronecker product approximates empirical Fisher information matrix. The method demonstrates better performance compared to diagonal approaches such as Adam.

**Strengths:**

1. Kronecker factored preconditioner with diagonal factors hasn't been used with empirical Fisher matrix before.
2. Thorough experimentation.

**Weaknesses:**

1. The preconditioner still requires layer inputs and gradients backpropaged through the layer, which is not always feasible for large training systems.
2. The Adafactor already uses Kronecker factored preconditioner with diagonal factors. There is no comparison against Adafactor.
3. sub-optimal regret bound $O(\log(T)\sqrt{T})$, compared to Shampoo - which is optimal $O(\sqrt{T})$.
4. There are low rank approaches with similar complexity as proposed methods  such as EVA [1]. There should be comparison against EVA.

If 2, 3, and 4 are addressed, I am willing to increase my score.

[1] Zhang, Lin, Shaohuai Shi, and Bo Li. "Eva: Practical Second-order Optimization with Kronecker-vectorized Approximation." The Eleventh International Conference on Learning Representations. 2022.

**Questions:**

Does your shampoo baseline use grafting?
CASPR is another paper which uses Kronecker sum based approach, what are your thoughts on Kronecker sum based combination to approximate empirical Fisher information matrix.

---

> ### Author Response · Authors · 2024-11-20
> **Rebuttal (part 1/2)**
>
> Dear Reviewer jcNA,
>
> We sincerely thank the reviewer for their valuable feedback, which highlights the novelty of our manuscript through the introduction of a new approximation of the FIM and the demonstration of AdaFisher’s performance via thorough experimentation. Your insightful comments and questions have been instrumental in improving the clarity and quality of our work.
>
> Below are the responses to the comments in Weakness and Question sections:
>
> **[R-jcNA-C1] Feasibility of AdaFisher for large training systems**: Please note adopting AdaFisher for training large models (e.g. LLMs) does not require significantly more computational resources than Adam, as AdaFisher’s computational overhead is comparable to Adam’s (please refer to Figure 24 in Appendix D.2.7 for memory usage and Figure 6 in Section 4.4 for time comparisons). We recognize that our computational resources are limited compared to the extensive infrastructure available to large corporations like Meta and Google. Despite our constraints, we are committed to achieving the highest possible standards in our work. We are hopeful that access to additional resources in the future will enable us to further expand the scope and impact of our research.
>
> **[R-jcNA-C2] AdaFactor and EVA comparison**: The reviewer highlights two important and relevant optimizer methods: AdaFactor and EVA. In response, we have included a detailed comparison of AdaFisher against these baselines in the revised manuscript (please refer to Appendix D.2.5 and D.2.6). Our results demonstrate that while EVA offers improved efficiency and slightly better performance compared to K-FAC, AdaFisher consistently achieves superior results under WCT (Appendix D.2.5) and with the same amount of training epochs (Appendix D.2.6). Moreover, while AdaFactor is slightly more efficient than both Adam and AdaFisher, AdaFisher achieves significantly better performance, underscoring its effectiveness. Additionally, we have summarized these findings in the following tables (Table 1 shows WCT results, and Table 2 illustrates performances with the same amount of epochs), which provide a comprehensive overview of the highest test accuracies and training efficiencies for all methods discussed.
>
> **Table 1:** Performance comparison of AdaFisher and other optimizers using ResNet-18 (CIFAR100) and MobileNet-V3 (CIFAR10). Results are reported using WCT of 200 AdaFisher training epochs as the cutoff.
> |Network-Optimizer|SGD|Adam|AdaFactor|AdaHessian|K-FAC|Eva|Shampoo|AdaFisher|
> |-----------------|---|----|---------|----------|-----|---|-------|---------|
> |MobileNet-V3|94.43|93.32|93.21|92.86|94.34|94.41|93.81|**95.28**|
> |ResNet-18|76.56|75.74|69.45|71.79|76.03|76.69|76.78|**77.28**|
>
> **Table 2:** Performance comparison of AdaFisher and other optimizers using (a) ResNet-18, (b) MobileNet-V3, and (c) ResNet-50 on CIFAR100 for 200 epochs.
> ### (a) ResNet-18
> |Optimizer|SGD|Adam|AdaFactor|AdaHessian|K-FAC|Eva|Shampoo|AdaFisher|
> |---------|---|----|---------|----------|-----|---|-------|---------|
> |Test Acc|76.52|75.71|69.78|76.86|76.96|77.08|**77.35**|77.28|
> |Training Time (min)|20.03|23.33|21.67|96.67|46.46|43.18|216.67|26.58|
>
> ### (b) MobileNet-V3
> |Optimizer|SGD|Adam|AdaFactor|AdaHessian|K-FAC|Eva|Shampoo|AdaFisher|
> |---------|---|----|---------|----------|-----|---|-------|---------|
> |Test Acc|73.42|70.53|71.08|62.36|75.16|75.48|70.65|**77.56**|
> |Training Time (min)|50.03|56.63|54.22|206.28|116.86|96.78|487.21|60.12|
>
> ### (c) ResNet-50
> |Optimizer|SGD|Adam|AdaFactor|AdaHessian|K-FAC|Eva|Shampoo|AdaFisher|
> |---------|---|----|---------|----------|-----|---|-------|---------|
> |Test Acc|76.12|73.03|70.78|76.18|77.66|78.01|78.89|**78.91**|
> |Training Time (min)|70.13|76.67|73.32|502.28|149.36|138.58|583.11|83.02|

---

> > ### Author Response · Authors · 2024-11-20
> > **Rebuttal (part 2/2)**
> >
> > **[R-jcNA-C3] Regret Bound**: Please note Shampoo is based on SGD framework while AdaFisher on Adam’s framework, a two different frameworks yielding different regret bounds (please refer to Table 1). While Shampoo theoretically achieves a lower regret bound, this advantage is accompanied by highly increased computational complexity. Specifically, Shampoo requires the computation, storage, and inversion of preconditioner matrices, which is resource-intensive, especially for large-scale models. This results in longer training times per iteration, as illustrated in Figure 6 (Panel D) and Table 15 in Appendix D.2.6. Conversely, AdaFisher employs a novel approximation to the FIM based on the diagonal block-Kronecker and reduces the computational demands (i.e. storing and inverting diagonal matrices). This efficiency allows AdaFisher to perform more optimization steps within the same wall-clock time (WCT), potentially offsetting its theoretical disadvantage in regret bound. As illustrated in Tables 2,3 and Figures 5, 16-19, this results in much faster convergence for AdaFisher within a given wall-clock budget, a critical factor in large-scale machine learning tasks. Thus, while regret bounds are a valuable theoretical measure, WCT offers a more practical metric when evaluating optimizers’ effectiveness in real-world applications.
> >
> > **[R-jcNA-C4] Question on using grafting for Shampoo/CASPR**: In our implementation, the Shampoo baseline does not utilize grafting. We adhered to the original formulation of Shampoo to ensure consistency with its standard implementation. While grafting techniques, such as those used in AdaGrad or Adam, may enhance optimization in certain scenarios, our comparison focuses on evaluating the performance of the original Shampoo algorithm to maintain consistency with its standard implementation.
> >
> > The reviewer raises a relevant point about the use of Kronecker sum-based approaches, such as the CASPR optimizer, for approximating the empirical FIM. While CASPR combines axis preconditioners through a Kronecker-sum-based approximation, AdaFisher takes a distinct approach by employing a diagonal block Kronecker approximation of the FIM. This difference is critical: Kronecker sum-based methods, like CASPR, involve higher computational complexity due to the need for combining and inverting multiple matrix components. In contrast, AdaFisher captures essential curvature information directly through diagonal blocks, offering a computationally efficient and practical alternative without sacrificing approximation fidelity (please refer to Section 3.1). This streamlined design ensures AdaFisher avoids the increased complexity associated with Kronecker sum-based methods while maintaining strong performance across tasks.
> >
> > **[R-jcNA-Concluding Remarks]** We would like to take the opportunity and thank the reviewer for their helpful comments to improve the quality and clarity of our paper. We have tried addressing all the points raised by the reviewer to the best of our ability given the limited time we had. Finally, we highly appreciate the reviewer for reconsidering their rating of the paper. We welcome further questions to address that may arise during the rebuttal.

---

> > > ### Comment · Reviewer_JcNA · 2024-11-20
> > > **Increased score**
> > >
> > > Authors have addressed my concerns with their experiments on comparisons with EVA and Adafactor, which is cruicial for the paper. CASPR requires more matrix multiplications, but in case of diagonal blocks, the matmul won't be needed, thus complexity isn't even an issue.

---

> > > > ### Author Response · Authors · 2024-11-20
> > > >
> > > > We would like to thank you for your consideration and re-evaluation of the score. Please let us know if you have any further questions and we would be happy to address them.
> > > > Sincerely,
> > > > Authors

---

### Author Response · Authors · 2024-11-20
**Dear Reviewers (rebuttal)**

We would like to thank all reviewers for their time and valuable feedback when reviewing our paper. We appreciate their constructive criticisms that aided us in formulating our responses which will serve to improve the overall quality of our paper. We have received all comments and addressed them all in the revised manuscript as well the official comment section for each reviewer. Furthermore, we encourage all the reviewers to read official comments from our reply to the other reviewers to follow up more on the detailed questions. For the ease of the reviewers, we have highlighted the revised manuscript in blue for ease of visual navigation.

Below, we summarize the strengths of our paper based on the reviewers' evaluations.

### Novel Methodology
- **`jcNA`**: *"Kronecker factored preconditioner with diagonal factors hasn't been used with empirical Fisher matrix before."*
- **`jye4`**: *"The paper innovatively discovers that the Kronecker factor is diagonally dominant and proposes a diagonal block-Kronecker approximation for the FIM."*
- **`K3HY`**: *"The introduction of a diagonal block-Kronecker approximation applicable to various layers offers an interesting balance between computational efficiency and the use of curvature information typically employed in second-order optimization."*

### Theoretical Contributions
- **`K3HY`**: *"The paper presents a rigorous theoretical convergence analysis for both convex and non-convex cases, asserting a convergence rate of [..], similar to Adam-type methods. The derivation of the update rules using the diagonal approximation of the FIM is clearly explained."*

### Empirical Performance of the Proposed Method
- **`K3HY`**: *"Comprehensive experimental evidence showing AdaFisher outperforms baseline methods."*
- **`jcNA`**: *"The method demonstrates better performance compared to diagonal approaches such as Adam."*
- **`jye4`**: *"On image classification and language modeling tasks, AdaFisher achieves better results than other second-order optimization methods when applying Wall-Clock-Time method (training for the same amount of time)."*
- **`jye4`**: *"The resulting AdaFisher optimizer shows good performance."*
- **`K3HY`**: *"Comprehensive experimental evidence showing that AdaFisher outperforms baseline methods (Adam, K-FAC, AdaHessian, etc.) on benchmark datasets such as CIFAR-10, CIFAR-100, ImageNet, and WikiText-2 across different network architectures."*
- **`jcNA`**: *"Thorough experimentation."*

### Stability
- **`K3HY`**: *"AdaFisher shows strong stability across varying learning rates and batch sizes, reducing the need for extensive hyperparameter tuning, which is a common challenge when training deep models."*

### Presentation
- **`jye4`**: *"The paper is well-organized and easy to follow."*

---

---

### Meta-Review · Area_Chair_aZ5A · 2024-12-18

**Metareview:**

This paper proposes AdaFisher, a new adaptive second-order optimization method, which offers a strong balance between convergence efficiency and computational feasibility. The authors introduce a novel diagonal block-Kronecker approximation to the Fisher Information Matrix (FIM) for preconditioning gradients, making second-order methods more practical for deep neural network training. One of the key strengths is the robustness of AdaFisher across different tasks, including image classification and language modeling, as well as its ability to outperform state-of-the-art optimizers like Adam and K-FAC in terms of accuracy and convergence speed. The paper also provides a solid theoretical foundation, demonstrating convergence analysis for both convex and non-convex cases, which is a valuable contribution to the field of second-order optimization.

The empirical results are thorough and demonstrate clear improvements over baseline optimizers. The experiments are well-structured, showing that AdaFisher achieves better generalization and stability across a range of hyperparameters, which reduces the effort needed for tuning. Notably, the method is shown to perform well under both wall-clock time comparisons and fixed-epoch evaluations. Additionally, the authors have addressed concerns during the discussion phase effectively, clarifying the novelty compared to related works and providing further comparisons with optimizers like EVA and AdaFactor. Overall, this work presents meaningful advancements in optimization strategies for deep learning.

**Additional Comments On Reviewer Discussion:**

During the reviewer discussion, the authors effectively addressed the key concerns raised by reviewers, particularly those related to comparisons with existing methods, computational efficiency, and experimental clarity. The authors provided detailed rebuttals, clarifying the distinctions between AdaFisher and similar optimizers like EVA, AdaFactor, and Shampoo, while also adding new experimental results to strengthen their claims. They acknowledged and corrected minor issues related to hyperparameter tuning and initialization sensitivity.

---

### Decision · Program_Chairs · 2025-01-22

Accept (Poster)